# Dietary intake, nutritional status, and health outcomes among vegan, vegetarian, and omnivorous Czech families

Marina Heniková [1,2,9], Anna Ouřadová[1,9], Eliška Selinger[1,3,4], Filip Tichánek[5], Petra Polakovičová [5,6], Dana Hrnčířová[2], Pavel Dlouhý[2], Martin Světnička[7], Eva El-Lababidi[7], Jana Potočková[1], Tilman Kühn [8], Monika Cahová[5,10] & Jan Gojda [1,10] ✉

## Abstract

**Background:** Vegan diets are gaining popularity in the general population because of their perceived environmental and health benefits. However, concerns remain regarding potential nutrient deficiencies, particularly during critical growth periods. We aimed to compare growth, cardiovascular health, bone turnover, iodine, and overall micronutrient status among families adhering to vegan, vegetarian, and omnivorous dietary habits.
**Methods:** A cross-sectional study was conducted among 95 Czech families (47 vegan, 23 vegetarian, and 25 omnivore), comprising 187 adults and 142 children. Clinical examination, fasting blood, and 3-day prospective diet records were collected to compare growth, cardiovascular health, bone turnover, iodine, and overall micronutrient status among dietary groups and across ages. We used robust mixed-effect models, adjusted for confounders and accounting for family clustering, for group comparison and elastic net logistic regression.
**Results:** No significant differences in children's growth characteristics between the dietary groups are found. Vegan children have the best cardiometabolic indices (low-density lipoprotein and total cholesterol) observed as well as in adults. Comparable indices of bone turnover among groups are observed, although vitamin D levels are generally highest and urinary phosphate levels lowest in vegan groups. While vegan children show lower urinary iodine, it is not associated with differences in thyroid-stimulating hormone levels compared to other groups. Mixed-effects models demonstrate familial clustering of height, uric acid, high-density lipoprotein, parathormone, and vitamins B12 and D in children and selenium, zinc, iodine, vitamin B12, and folate in adults.
**Conclusions:** Our results show that dietary habits significantly predict nutritional biomarkers, with familial influences contributing to interindividual variability. While vegans have better cardiometabolic profiles, low iodine status could be of concern.

## Plain language summary

Vegan and vegetarian diets are increasingly popular, but concerns remain about whether they provide all essential nutrients for children's growth and health. This study compared Czech families following vegan, vegetarian, or omnivorous diets, where children had been on the respective diet since birth. We examined 95 families, including 142 children, through physical exams, dietary records, and blood and urine samples. Children across all diet groups had similar growth and bone health. Vegan families showed the most favorable cholesterol and cardiovascular health indices, but also had lower iodine levels. Family patterns influenced several vitamin and mineral levels. Overall, plant-based diets supported normal child growth and provided cardiometabolic benefits, although iodine intake may require attention.

The global trends to reduce the environmental burden of food production and tackle the pandemic of non-communicable diseases are being followed by tendencies both in consumers and policy-makers aimed at reducing the consumption of foods of animal origin[1]. The growing trend towards plant-based diets is increasingly evident in many regions of the former Eastern European bloc, including the Czech Republic. This trend is particularly more pronounced among younger demographic groups. According to national surveys, in 2022, 3% of Czech consumers identified themselves as vegan, 7% as vegetarian, and 25% as flexitarian, i.e., voluntarily reducing their intake of animal-based foods[2]. The numbers that have more than doubled since 2019[3]. As the eating habits are shared among the households, a growing number of children on these diets could be expected.

---

While vegan and vegetarian diets were shown to be associated with favorable health effects and reduced risk of various non-communicable diseases[4–7], when self-directed without guidance, they may also carry the potential risks of both specific nutrient deficiencies (such as vitamins B12 and D, and the minerals iodine, zinc, calcium, iron, and selenium) and inadequate caloric and protein intake[8,9]. This may be of particular concern in critical developmental stages such as childhood[10,11], where vegan diets were related to specific concerns of growth[12,13], bone health[14], and iodine deficiency[15]. However, the existing findings need to be interpreted with caution due to the often lower certainty of evidence[6,7].

Families usually share dietary patterns, so dietary choices made within a household have a collective effect on all members, including children[16]. More importantly, dietary habits formed in childhood can significantly shape health outcomes later in life, highlighting the importance of establishing healthy eating habits early in life[16,17]. Investigating plant-based diets within family units, encompassing both children and parents, is integral to understanding their impact on health outcomes.

The KOMPAS study (Cohort prospective study of emerging nutritional factors among families) is a prospective cohort single-center study that aims to investigate the health effects of different dietary patterns within family units, tracking these effects from childhood into adulthood[18]. The aim of the current study is to compare baseline data of families with distinct dietary habits (VN: vegan, VG: vegetarian, and OM: omnivore) to describe (1) differences in growth, anthropometric characteristics, (2) differences in health-related outcomes (cardiovascular and bone health, risk of iodine and iron deficiency, risk of micronutrient deficiencies), (3) differences in dietary intake of critical nutrients, and (4) explore the interrelations between observed variables, namely the predictive potential of a dietary group and family on the observable variations in these variables.

## Methods
### Design and the study population
An overview of the study design is depicted in Fig. 1[18]. For this single-center observational study conducted in the University Hospital Kralovske Vinohrady, Czech Republic, Prague, participants were recruited through advertisements on social media and through the clinical practice of cooperating pediatricians (July 2021- April 2022). Inclusion criteria were defined as a family consisting of two adults and at least one child under 7 years of age. All family members should follow the same dietary pattern (self-identified vegan, vegetarian, or omnivore) and be willing to undergo the study procedures, including the collection of biological samples. For the purpose of our study, a vegan diet was defined as a diet that excludes meat and meat products, fish, milk and dairy products, and eggs. The vegetarian diet was defined by the exclusion of meat and meat products, and fish, but the consumption of eggs or dairy products. The omnivore diet was defined as a diet without strict dietary restrictions on foods of animal origin. Exclusion criteria were as follows: different diets followed by different family members, only children older than 7 years, inability or unwillingness to undergo a complete clinical examination, and sampling. Reported history of diseases associated with malabsorption (pancreatitis, celiac disease, etc.). No other restrictions.

Each subject underwent a clinical visit: (1) anthropometric examination (weight, height, waist, and hip circumference; maximum handgrip strength (hydraulic dynamometer Saehan SH5001, Saehan Corp.) and bioimpedance body composition analysis (Nutriguard-M, Data Input GmbH) in adults only); growth was assessed by converting height and weight into percentiles using standard percentile charts validated for use in the Czech Republic[19]. (2) detailed medical history report (self-reported), and (3) 12 h fasting (only in children ≥3 years and adults) venous blood and spot urine sampling. All laboratory parameters were analyzed in ISO-certified laboratories. Individual analytic methods are summarized in Supplementary Data 1.

### Nutritional assessment
A 3-day (two weekdays and one weekend day) weighted dietary record method was used to evaluate the dietary intake of foods. Electronic kitchen scales were provided for the participants. Breast milk intake was recorded by mothers in terms of breastfeeding duration in minutes [19]. Participants were trained in diet recording, and after completion, the records were checked and verified by a dietitian. For children, diet was recorded by parents or carers; only unweighted estimates were generally available in nursery facilities. Nutrient and energy intake data were analyzed using the Nutrixo nutritional software, based on validated food composition databases (ArcaiSoft, Czech Republic), and averaged to produce daily intakes. For products not listed in any of the databases, the dietitian recorded nutrient content from the product packaging. As a 3-day dietary record may not capture irregularities and seasonal variations in supplement use, supplements were recorded in parallel and were not included in the quantitative nutrient intake estimates.

### Statistics and reproducibility
All analyses were performed in R (v4.4.3)[20]. For descriptive analyses, clinical characteristics across diet groups were summarized separately for children <3 years, children ≥3 years, and adults, reporting medians (25th and 75th percentiles) for continuous variables and counts (%) for categorical variables. The rationale for separating the children into two groups was based on differences in daily allowances, reference values, and breastfeeding prevalence. For inferential analyses, children were analysed as one group. Specific statistical outcomes were: (1) do clinical outcomes vary significantly among different dietary groups, (2) which factors (e.g., sex, age, breastfeeding status for children, or supplementation when applicable), are most significantly related to the clinical outcomes, besides the dietary group, (3) are these factors correlated ("clustered") within the same family, and (4) can the clinical characteristics effectively discriminate between different diet groups?

Differences in clinical outcomes were assessed using robust linear mixed-effects models with 'robustlmm' R package, adjusting for key prespecified variables (age and sex; for children, also breastfeeding-related variables). Additional covariates were included when relevant, namely supplementation status and, for anthropometric outcomes in children, birth weight. A random effect (intercept only) for family was included to account for within-family correlations. To assess the importance of each variable, we also fitted standard linear mixed-effects models with 'lme4' R package and compared models with and without specific variable groups using the Akaike Information Criterion (AIC), which reflects how much a given variable contributes to outcome prediction. When necessary, outcomes were log2-transformed, and observations with missing outcomes were excluded (assuming missingness unrelated to diet). Thirteen missing values in partial breastfeeding were imputed based on age.

To evaluate whether clinical characteristics could predict diet group, we applied Elastic Net logistic regression using 'glmnet' R package, comparing (i) a baseline model (age, sex, and, for children, breastfeeding status), (ii) a reduced model (adding clinical variables not strongly influenced by supplementation), and (iii) a full model (including all clinical characteristics). Missing predictor values were imputed using single stochastic imputation with 'mice' R package. Out-of-sample predictive accuracy was assessed via area under the ROC (Receiver Operating Characteristic) curve (area under curve, AUC) using 'pROC' R package, with 500 cluster bootstrap resamples that kept entire families together in training or testing sets. Differences in AUC, along with 95% confidence intervals, were used to determine whether more complex models provided significant improvements in predictive performance.

Potential sources of bias were addressed by recruiting families across all diet groups and by modeling family-level clustering explicitly (random intercepts in LMMs, cluster bootstrap in validation of predictive models). Missing data were handled differently according to context: in inferential LMMs, 13 missing values on partial breastfeeding were singly imputed based on age, whereas in predictive models, missing predictors were stochastically imputed using the "mice" R package. Robust estimation was used

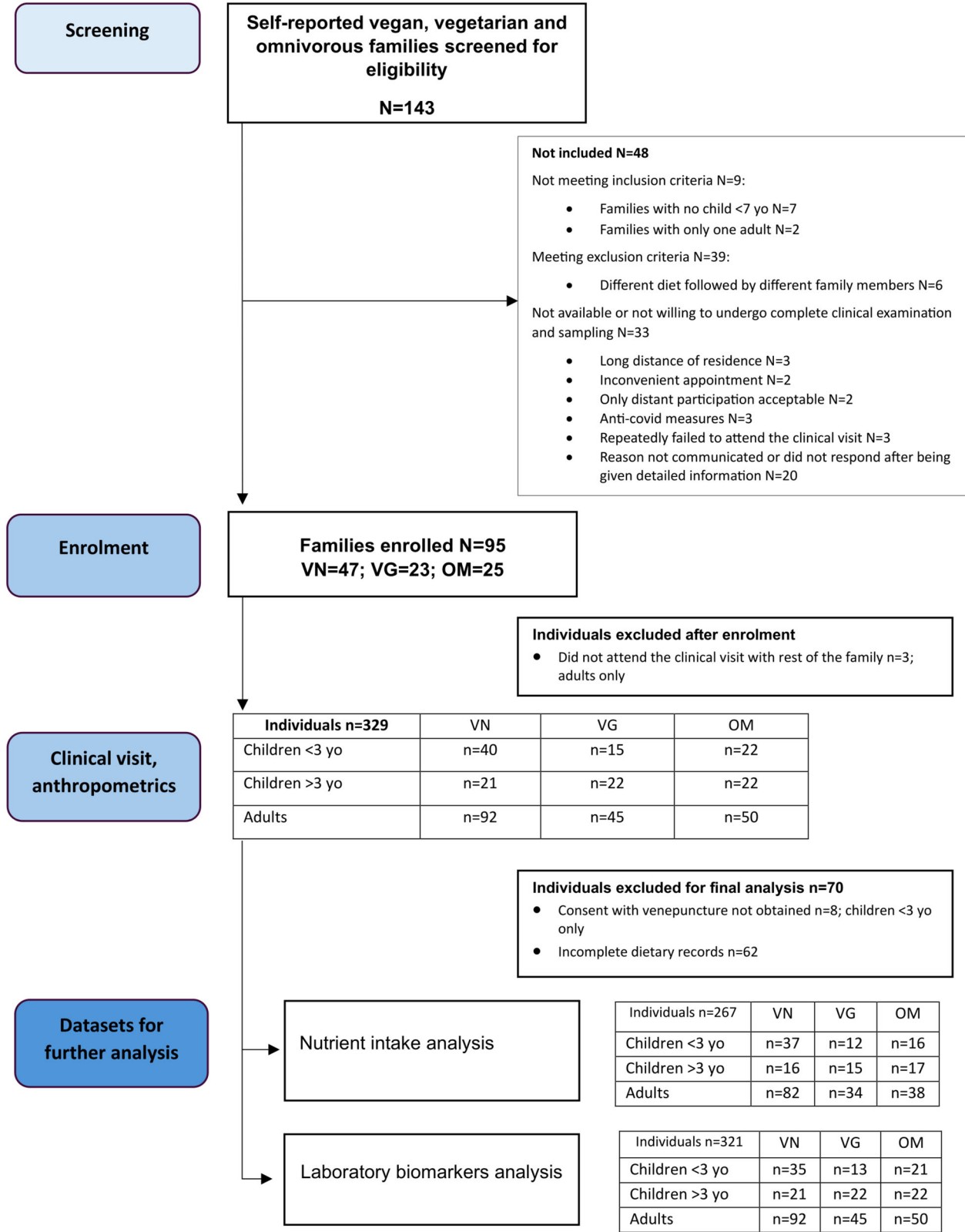

**Fig. 1 | Flow diagram of the study.** Vegan (VN), vegetarian (VG), and omnivorous (OM) families and individuals. *N* number of families; *n* number of individual subjects.

to reduce the impact of outliers, and analyses were adjusted for key pre-specified covariates. Voluntary participation may still limit the representativeness of the sample.

Detailed descriptions of the modeling approaches and assumptions are provided in the Supplementary Materials and online statistical report https://filip-tichanek.github.io/kompas_clinical/. We used a significance level of $\alpha = 0.05$ ($P < 0.05$) for the robust linear mixed-effects models. P-values were not corrected for multiple comparisons to maximize sensitivity for a potential risk associated with the vegan/vegetarian diet (omitting a true risk was considered more serious than allowing a few false positives).

**Table 1 | Groups' characteristics**

| | | Vegan | Vegetarian | Omnivore |
|---|---|---|---|---|
| n | Adults | 92 | 45 | 50 |
| | Children <3 yo | 40 | 15 | 22 |
| | Children >3 yo | 21 | 22 | 22 |
| Age (years) | Adults | 33.9 (31.5; 37.2) | 36.7 (33.4; 38.2) | 35.8 (32.8; 40.0) |
| | Children <3 yo | 1.4 (0.9; 2.1) | 1.5 (0.7; 2.0) | 1.5 (1.0; 2.2) |
| | Children >3 yo | 4.7 (3.6; 5.8) | 5.3 (4.5; 6.8) | 5.5 (4.3; 6.3) |
| Sex F/M | Adults | 47/45 | 23/22 | 25/25 |
| | Children <3 yo | 21/19 | 9/6 | 11/11 |
| | Children >3 yo | 10/11 | 9/13 | 13/9 |
| BreastFeed_full (month) | Children <3 yo | 6.0 (5.0; 6.0) | 6.0 (5.8; 6.0) | 6.0 (6.0; 6.0) |
| | Children >3yo | 6.0 (6.0; 6.0) | 7.0 (6.0; 7.8) | 6.0 (5.3; 6.0) |
| BreastFeed_total (month) | Children <3 yo | 12 (8; 17) | 18 (9; 21) | 10 (9; 14) |
| | Children >3 yo | 18 (13; 30) | 18 (16; 22) | 10 (8; 14) |
| Birth weight (g) | Children <3 yo | 3425 (2919; 3705) | 3325 (2993; 3645) | 3290 (3019; 3575) |
| | Children >3 yo | 3500 (3060; 3650) | 3400 (3220; 3555) | 3430 (3088; 3743) |
| Allergies | Adults | 12 (13%) | 12 (27%) | 17 (34%) |
| | Children <3 yo | 1 (2.5%) | 1 (6.7%) | 5 (23%) |
| | Children >3 yo | 2 (9.5%) | 4 (18%) | 4 (18%) |
| Fractures | Adults | 36 (40%) | 23 (51%) | 27 (54%) |
| | Children <3 yo | 0 | 0 | 1 |
| | Children >3 yo | 1 (4.2%) | 0 | 3 (13%) |

Group characteristics of vegans (VN = 153), vegetarians (VG = 82), and omnivores (OM = 94) across age strata. Data are shown as medians (25th; 75th percentile) when appropriate. BreastFeed_full, duration of exclusive breastfeeding; BreastFed_total, duration of exclusive and partial breastfeeding. The history of allergies and fractures is reported as the total number of subjects among groups across age strata. Source data are available online[21].

However, FDR (False Discovery Rate) - corrected P-values and confidence intervals for the diet group differences can be found in the online statistical report, along with all relevant R codes[21].

Further details of the study protocol are available in the **Supplementary Methods** and the protocol published online[18].

### Ethics approval and consent to participate
The study was reviewed and approved by the relevant Institutional Review Board of University Hospital Kralovske Vinohrady, 22/06/2020, under no. EK-VP/39/0/2020. Written informed consent was signed before enrollment in the protocol for each study participant. For children, parental consent was sought. The study was performed under the guidance of the Helsinki Declaration.

## Results
### Description of the Study Groups
For the study overview, see the Study flow-chart Fig. 1 and Table 1.

A total of 95 families (VN = 47, VG = 23, OM = 25) consisting of 187 adults, 65 children ≥3 years of age and 77 children <3 years of age were enrolled (October 2021-October 2022) and examined in a cross-sectional setting. Numbers of families with more than one child were: VN = 13, VG = 9, and OM = 15. Adult vegans were on the diet on average ≈for 7.4 years, whereas vegetarians ≈12 years, and all children were on each respective diet from birth. Detailed medical history of the participants can be found in the Supplementary Table 1, history of fractures in detail in Supplementary Table 2. and demographic characteristics in Supplementary Table 3.

### Anthropometrics and clinical outcomes
Description of anthropometric and clinical characteristics among three diet groups and across three age strata: (i) children <3 years old, (ii) children ≥ 3 years old, and (iii) adults are shown in Tables 2–4. For subsequent modeling,

all children were combined into a single dataset to increase statistical power, and robust mixed-effects models (rLME) were applied separately for children and adults to estimate adjusted differences between diet groups (Tables 5, 6), accounting also for age, sex, breastfeeding status (children only), and, when relevant, supplementation status or birth weight (children, anthropometric outcomes), as well as within-family clustering (by inclusion of a random intercept for family). The relative importance of diet, covariates, and within-family clustering was evaluated using AIC (Fig. 2).

### Clinical outcomes in children
The descriptive results of children groups are summarized in Tables 2, 3. In the following section, we comment on the findings based on the outcomes of the rLME model, where children were analyzed as one group (summarized in Table 5).

We found no significant differences in *anthropometric characteristics* in children. Altogether, seven children (VN = 4; VG = 2; OM = 2) below the third height percentile and four VN children below the third weight percentile were identified.

Among the *cardiovascular health outcomes*, we found lower C-LDL in VN and lower TC and C-LDL in VN and VG when compared to OM. Serum C-LDL showed a trend with vegans having the lowest levels.

*Exploring bone health*. Vegan and vegetarian children had higher serum vitamin D concentrations than the OM group. Forty-one children had 25(OH)D < 75 nmol/l; (VN = 12, VG = 9, OM = 20). We found a lower phosphate-creatinine ratio in VN when compared to other groups.

The mean *urinary iodine* was significantly lower in VN compared to the other dietary groups. Twenty children had UIC < 100 µg/l (VN = 16, VG = 3, OM = 1), though their mean TSH was 2,25 mU/l (none above the reference range), i.e., in the normal range. Out of the sixteen vegan children, two had UIC < 20 µg/l.

In VN children, we found indicators of better saturation with *vitamin B12*, i.e., lower concentration of methylmalonic acid and homocysteine

**Table 2 | Clinical variables in children <3 years old among dietary groups**

| | Vegan | Vegetarian | Omnivore |
|---|---|---|---|
| Anthropometrics | | | |
| Body weight (percentile) | 34 (14; 68) | 48 (30; 71) | 63 (34; 78) |
| Body height (percentile) | 38 (15; 64) | 51 (31; 68) | 43 (33; 74) |
| BMI (percentile) | 56 (35; 70) | 56 (42; 73) | 55 (45; 78) |
| Weight to height ratio (percentile) | 53 (40; 66) | 61 (41; 70) | 58 (46; 77) |
| Glucose and lipid metabolism | | | |
| FPG (mmol/l) | 4.4 (4.2; 4.6) | 4.7 (4.3; 4.9) | 4.5 (4.3; 4.6) |
| TC (mmol/l) | 3.9 (3.5; 4.0) | 3.9 (3.4; 4.3) | 4.0 (3.1; 4.4) |
| C-HDL (mmol/) | 1.3 (1.0; 1.4) | 1.1 (1.0; 1.3) | 1.1 (0.8; 1.4) |
| C-LDL (mmol/l) | 2.0 (1.7; 2.2) | 2.3 (1.9; 2.6) | 2.0 (1.5; 2.8) |
| TG (mmol/l) | 1.1 (0.8; 1.5) | 0.8 (0.7; 1.2) | 1.1 (0.9; 1.5) |
| Biogenic elements | | | |
| Magnesium (mmol/l) | 0.88 (0.84; 0.92) | 0.89 (0.85; 0.97) | 0.87 (0.82; 0.90) |
| Selenium (mmol/l) | 0.75 (0.54; 0.94) | 0.73 (0.64; 0.86) | 0.75 (0.68; 0.95) |
| Zinc (mmol/l) | 10.7 (9.8; 11.8) | 10.8 (10.5; 11.2) | 11.7 (10.7; 13.4) |
| Iron (µmol/l) | 11.1 (8.1; 16.8) | 10.2 (5.4; 12.5) | 9.6 (8.3; 14.6) |
| Iron metabolism | | | |
| TIBC (µmol/l) | 71 (66; 77) | 73 (69; 75) | 75 (72; 79) |
| Ferritin (µg/l) | 12 (9; 16) | 11 (9; 14) | 15 (9; 21) |
| Transferrin (g/l) | 2.8 (2.6; 3.0) | 2.9 (2.7; 3.0) | 3.0 (2.9; 3.2) |
| Transferrin saturation (%) | 16 (11; 24) | 14 (7; 18) | 13 (12; 18) |
| sTfR Index | 1.5 (1.3; 1.9) | 1.6 (1.4; 1.8) | 1.6 (1.3; 2.1) |
| STFR (mg/l) | 1.7 (1.5; 2.0) | 1.65 (1.4; 1.9) | 1.8 (1.7; 1.9) |
| Hemoglobin (g/l) | 119 (115; 121) | 117 (110; 121) | 116 (112; 123) |
| MCV (fl) | 79.7 (76.8; 80.8) | 77.6 (75.8; 80.7) | 76.8 (70.9; 78.8) |
| Bone health | | | |
| Calcium (mmol/l) | 2.63 (2.55; 2.68) | 2.64 (2.60; 2.72) | 2.65 (2.54; 2.72) |
| Phosphorus (mmol/l) | 1.72 (1.63; 1.83) | 1.79 (1.69; 1.83) | 1.70 (1.66; 1.80) |
| uCa (mmol/l) | 1.0 (0.3; 2.1) | 1.1 (0.6; 2.6) | 0.6 (0.5; 1.9) |
| uCCR (mg/mg) | 0.4 (0.2; 0.8) | 0.5 (0.1; 0.9) | 0.1 (0.1; 0.8) |
| uPCR (mg/mg) | 3.1 (1.7; 4.6) | 3.5 (2.2; 4.2) | 6.4 (3.1; 7.4) |
| PTH (pmol/l) | 3.2 (2.1; 4.0) | 2.6 (1.8; 3.2) | 2.5 (1.4; 3.8) |
| CTx (ug/l) | 1.2 (1.1; 1.4) | 1.3 (0.9; 1.4) | 1.2 (1.0; 1.3) |
| P1NP (ug/l) | 988 (670; 1,201) | 986 (861; 1,201) | 920 (585; 1,201) |
| Vitamin D (nmol/l) | 104 (83; 118) | 98 (81; 129) | 82 (67; 98) |
| Iodine and one-carbon metabolism | | | |
| UIC (µg/l) | 98 (66; 168) | 190 (134; 271) | 169 (135; 231) |
| uICR (µg/µg) | 402 (263; 786) | 563 (325; 1;089) | 614 (338; 788) |
| Active B12 (pmol/l) | 138 (91; 184) | 59 (46; 85) | 84 (72; 93) |
| Homocysteine (µmol/l) | 7.1 (6.0; 8.9) | 9.0 (7.2; 12.1) | 9.3 (7.4; 12.0) |
| MMA (nmol/l) | 152 (125; 187) | 305 (190; 628) | 280 (248; 386) |
| Folate (µg/l) | 18.1 (15.7; 22.4) | 18.2 (16.9; 21.1) | 14.4 (12.7; 17.2) |

**Table 2 (continued) | Clinical variables in children <3 years old among dietary groups**

| | Vegan | Vegetarian | Omnivore |
|---|---|---|---|
| Others | | | |
| IGF-1 (µg/l) | 64 (45; 89) | 63 (51; 83) | 70 (48; 107) |
| Urea (mmol/l) | 3.6 (2.7; 4.4) | 3.2 (2.7; 3.9) | 4.3 (3.3; 5.6) |
| Creatinine (µmol/l) | 19 (18; 23) | 20 (19; 23) | 21 (19; 23) |
| Uric acid (µmol/l) | 218 (187; 255) | 223 (175; 241) | 222 (191; 267) |

Medians (25th; 75th percentile) of clinical variables in vegan (VN), vegetarian (VG), and omnivorous (OM) children <3 years. Active B12, Holotranscobalamine; *BMI* Body mass index, *CTx* beta cross laps, *FPG* Fasting plasma glucose, *C-HDL* HDL cholesterol, *DBP* diastolic blood pressure, *IGF-1* insulin-like growth factor 1, *C-LDL* LDL cholesterol, *MCV* Mean corpuscular volume *MMA* Methylmalonic acid, *perc* percentile, *SBP* systolic blood pressure, *sTfR Index* Soluble transferrin receptor/log Ferritin Index, *STFR* Soluble transferrin receptor, *PTH* Parathormone, *P1NP* Procollagen type I aminoterminal propeptide, *UIC* Urine Iodine Concentration, *TC* total cholesterol, *TIBC* total iron binding capacity, *TG* Triglycerides, *uCa* urine calcium concentration, *uCCR* urinary calcium creatinine ratio, *uICR* urinary iodine creatinine ratio, urinary phosphate creatinine ratio. *IGF-1* insulin-like growth factor – 1; yo, years old. Source data are available online[21].

compared with both OM and VN groups. Nineteen children with active B12 levels >169,2 pmol/l (VN = 15, VG = 4) were identified. On the contrary, vitamin B12 < 27,4 pmol/l was identified in only two children (OM = 1, VN = 1).

## Clinical outcomes in adults

The descriptive results of adults are summarized in Table 4 and the outcomes of the rLME model in Table 6.

We found comparable *anthropometric* characteristics in adults in all three groups. The rLME model revealed significantly lower mean *total and LDL cholesterol* serum concentrations in vegans compared with OM and VG. There were forty-five subjects with total and LDL cholesterol above the upper reference limit of the institutional laboratory (VN = 15, VG = 13, OM = 17).

We found comparable mean values of markers of *bone health*: bone turnover and calcium metabolism. Mean values of vitamin D were in the normal reference range. Vegans had higher mean serum concentrations of vitamin D than VG and OM. We found seventy-six subjects with low vitamin D status across groups (i.e., 25(OH)D < 75 nmol/l; OM = 35, VG = 25, VN = 44), and we identified six vegans with PTH above the upper reference limit. Differences in the zinc levels showed a gradient (VN < VG < OM), though the significance was reached only between VN and OM and VG and OM, not between VN and VG. Twelve subjects had zinc bellow the reference range (i.e., zinc <9,8 mmol/l; OM = 2, VG = 1, VN = 9),

We found lower *iron stores* (serum ferritin and transferrin-to-ferritin ratio) in vegans and vegetarians when compared to OM. Sixty-five subjects had low iron stores (i.e., ferritin <22 (µg/l); OM = 12, VG = 20, VN = 33). There were no indices of disturbance of systemic iron homeostasis, as transferrin saturation or hemoglobin levels were comparable among groups. The mean folate serum levels were significantly higher in both VN and VG, compared with OM. B12 levels were comparable across groups. One vegan participant had vitamin B12 < 27,4 pmol/l and we identified subjects above the upper reference limit in active vitamin B12 (OM = 2, VG = 1, VN = 2). We found one male vegan with hemoglobin below the reference Hb<130 g/l, and in the whole sample, Hb levels were positively related to ferritin (R = 0.4, p < 0,0001).

## Family clustering and covariates' importance

Besides diet, clinical characteristics are influenced by other factors, namely those clustering within families. The statistical tool used in the previous analysis, rLME, allows us to assess the relative importance of covariates and quantify the extent to which these characteristics cluster within families. To evaluate the importance of each variable in a model, we employed the Akaike Information Criterion (AIC), which estimates a variable's contribution to the model's out-of-sample predictive accuracy. A larger

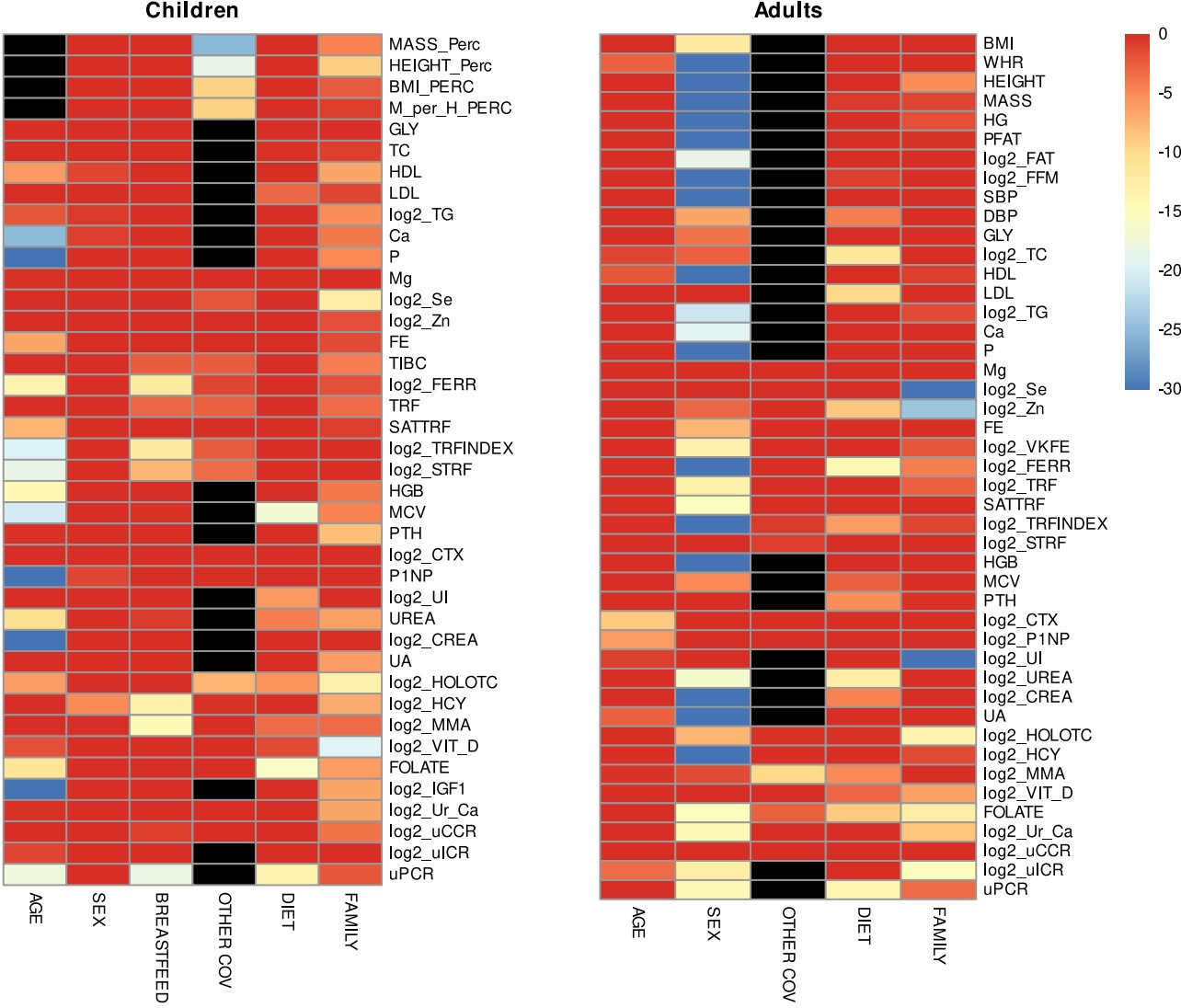

**Fig. 2 | Decrease of the Akaike information criterion (AIC) after the addition of the given variable to mixed-effect models.** Children <3 years old, >3 years old, and adults across dietary groups. The change of AIC indicates out-of-sample predictive accuracy of the models and was bounded to -30 (large importance of a covariate) and 0 (no to negligible importance). Black indicates that the variable was not included in the given model. Source data are available online[21].

decrease in AIC following the inclusion of a variable indicates a larger contribution to the model's predictive capability. The results are summarized in Fig. 2. When the AIC showed the large importance of family, we also reported an adjusted intraclass correlation coefficient (ICC), indicating how much of the total variability in the outcome is due to the grouping structure, i.e. family, after accounting for other variables.

In children, the most important covariate was age. It strongly affected serum concentrations of IGF-1, calcium, phosphorus, P1NP, and creatinine (AIC > −20). Age was also a substantial covariate for serum levels of folate, urea, ferritin, hemoglobin, uPCR, and soluble transferrin receptor, AIC = < -10;−20 >. HDL cholesterol, iron, and transferrin saturation were only mildly affected by age, AIC = < -5;−10 >. Sex had negligible effects. Breastfeeding was the most important factor for homocysteine serum concentration (AIC > −20), and it also affected serum MMA, ferritin, uPCR, and transferrin saturation, AIC = < −10;−20 >. Surprisingly, we did not find an effect of supplementation on relevant minerals or vitamins (denoted by the collective term other effects). Clustering within the family was found for most variables, but the most prominent was this factor for height (ICC = 57%), HDL cholesterol (ICC = 59%), B12 (ICC = 60%), PTH (ICC = 59%), uric acid (ICC = 56%), and particularly vitamin D (ICC = 67%).

In adults, the most important covariate was sex, as it had a strong (AIC > −20) or moderate (AIC = < −10;−20 >) effect on 30 variables. In addition to anthropometrics, sex was also crucial for blood pressure, HDL, TG, Ca, and P levels, variables reflecting iron metabolism, hemoglobin, urea, creatinine, uric acid, and homocysteine. Age was an insignificant covariate for all variables, but it mildly affected CTx (AIC = −8.7) and P1NP (AIC = −6.3). We did not identify the importance of vitamins and mineral supplements (denoted by the collective term other effects). Diet was the main determinant of total and LDL cholesterol and PTH, but it also contributed to other variables. Family clustering was substantial for circulating selenium (ICC = 73%), zinc (ICC = 41%), urinary iodine (ICC = 58%), B12 (ICC = 44%), and folate (ICC = 35%), altogether likely reflecting family-specific diet habits.

**Dietary intake**

Dietary intake of the main macro- and micronutrients of interest from foods, excluding supplementation, is summarized in Fig. 3 and Supplementary Tables 4–6.

The differences among groups were negligible in *children < 3 years old*. In this age group, the nutrient intake was similar across all groups, with the exception of saturated fats, where lower intakes were found in VN and VG

## Table 3 | Clinical variables in children ≥ 3 years old among dietary groups

| | Vegan | Vegetarian | Omnivore |
|---|---|---|---|
| Anthropometrics | | | |
| Body weight (percentile) | 40 (27; 67) | 62 (28; 65) | 52 (29; 73) |
| Body height (percentile) | 42 (11; 61) | 57 (38; 74) | 41 (19; 76) |
| BMI (percentile) | 43 (21; 68) | 46 (32; 58) | 46 (35; 63) |
| Weight to height ratio (percentile) | 43 (17; 70) | 47 (32; 57) | 47 (30; 63) |
| Glucose and lipid metabolism | | | |
| FPG (mmol/l) | 4.3 (4.2; 4.6) | 4.3 (3.8; 4.7) | 4.3 (4.0; 4.7) |
| TC (mmol/l) | 3.4 (3.2; 3.9) | 3.7 (3.2; 4.3) | 4.1 (3.6; 4.3) |
| C-HDL (mmol/) | 1.2 (1.1; 1.5) | 1.3 (1.1; 1.4) | 1.4 (1.1; 1.7) |
| C-LDL (mmol/l) | 1.7 (1.6; 2.0) | 2.1 (1.8; 2.5) | 2.2 (1.8; 2.3) |
| TG (mmol/l) | 0.9 (0.7; 1.1) | 0.6 (0.6; 0.9) | 0.9 (0.7; 1.2) |
| Biogenic elements | | | |
| Magnesium (mmol/l) | 0.86 (0.81; 0.89) | 0.83 (0.80; 0.87) | 0.82 (0.79; 0.85) |
| Selenium (mmol/l) | 0.77 (0.59; 0.86) | 0.68 (0.60; 0.91) | 0.83 (0.73; 0.92) |
| Zinc (mmol/l) | 10.6 (9.6; 13.1) | 11.0 (10.0; 13.0) | 11.8 (10.8; 13.1) |
| Iron (µmol/l) | 17 (11; 19) | 13 (12; 19) | 15 (9; 18) |
| Iron metabolism | | | |
| TIBC (µmol/l) | 67 (62; 74) | 73 (63; 77) | 72 (66; 78) |
| Ferritin (µg/l) | 16 (12; 22) | 19 (15; 26) | 22 (17; 29) |
| Transferrin (g/l) | 2.7 (2.5; 3.0) | 2.9 (2.5; 3.1) | 2.9 (2.6; 3.1) |
| Transferrin saturation (%) | 25 (17; 30) | 20 (14; 27) | 20 (15; 25) |
| sTfR Index | 1.2 (1.1; 1.4) | 1.3 (1.0; 1.4) | 1.1 (1.0; 1.4) |
| STFR (mg/l) | 1.5 (1.4; 1.7) | 1.6 (1.4; 1.7) | 1.5 (1.4; 1.6) |
| Hemoglobin (g/l) | 124 (117; 128) | 129 (123; 135) | 127 (122; 132) |
| MCV (fl) | 82.7 (80.1; 84.2) | 78.7 (76.9; 80.3) | 80.3 (78.5; 82.1) |
| Bone health | | | |
| Calcium (mmol/l) | 2.49 (2.47; 2.52) | 2.50 (2.45; 2.58) | 2.50 (2.44; 2.55) |
| Phosphorus (mmol/l) | 1.58 (1.43; 1.67) | 1.54 (1.44; 1.63) | 1.65 (1.49; 1.77) |
| uCa (mmol/l) | 0.6 (0.3; 1.3) | 1.8 (0.4; 3.5) | 1.4 (0.5; 2.4) |
| uCCR (mg/mg) | 0.2 (0.1; 0.4) | 0.3 (0.1; 0.8) | 0.3 (0.1; 0.5) |
| uPCR (mg/mg) | 1.7 (1.4; 3.0) | 3.4 (1.6; 4.8) | 3.3 (2.6; 5.4) |
| PTH (pmol/l) | 2.7 (2.3; 3.5) | 2.8 (1.9; 3.3) | 2.6 (1.8; 3.1) |
| CTx (µg/l) | 1.3 (1.1; 1.4) | 1.2 (1.1; 1.4) | 1.2 (0.9; 1.5) |
| P1NP (µg/l) | 572 (519; 644) | 492 (442; 649) | 454 (402; 509) |
| Vitamin D (nmol/l) | 94 (77; 104) | 85 (70; 97) | 70 (59; 85) |
| Iodine and One-Carbon matbolism | | | |
| UIC (µg/l) | 110 (76; 158) | 126 (85; 179) | 173 (141; 255) |
| uICR (µg/mg) | 339 (107; 404) | 169 (136; 366) | 259 (189; 440) |
| Active B12 (pmol/l) | 143 (102; 228) | 95 (81; 135) | 118 (97; 132) |
| Homocysteine (µmol/l) | 7.7 (6.4; 8.5) | 8.0 (7.0; 10.0) | 8.5 (8.0; 9.9) |
| MMA (nmol/l) | 155 (136; 202) | 170 (144; 257) | 204 (170; 236) |
| Folate (µg/l) | 17.2 (13.1; 21.0) | 17.5 (15.0; 18.8) | 11.1 (9.3; 13.7) |
| Others | | | |
| IGF-1 (µg/l) | 105 (82; 137) | 154 (122; 182) | 156 (107; 192) |

## Table 3 (continued) | Clinical variables in children ≥ 3 years old among dietary groups

| | Vegan | Vegetarian | Omnivore |
|---|---|---|---|
| Urea (mmol/l) | 4.1 (3.9; 4.8) | 4.7 (4.3; 5.8) | 4.5 (4.1; 5.4) |
| Creatinine (µmol/l) | 30 (27; 33) | 31 (27; 36) | 33 (28; 37) |
| Uric acid (µmol/l) | 226 (202; 261) | 236 (221; 249) | 222 (209; 243) |

Medians (25th; 75th percentile) of clinical variables in vegan (VN), vegetarian (VG), and omnivorous (OM) children >3 years. Active B12, Holotranscobalamine, *BMI* Body mass index, *CTx* beta cross laps, *FPG* Fasting plasma glucose, *C-HDL* HDL cholesterol, *DBP* diastolic blood pressure, *IGF-1* insulin-like growth factor 1, *C-LDL* LDL cholesterol, *MCV* Mean corpuscular volume, *MMA* Methylmalonic acid, *perc* percentile, *SBP* systolic blood pressure, *sTfR Index* Soluble transferrin receptor/log Ferritin Index, *STFR* Soluble transferrin receptor, *PTH* Parathormone, *P1NP* Procollagen type I aminoterminal propeptide, *UIC* Urine Iodine Concentration, *TC* total cholesterol, *TIBC* total iron binding capacity, *TG* Triglycerides, *uCa* urine calcium concentration, *uCCR* urinary calcium creatinine ratio, *uICR* urinary iodine creatinine ratio, urinary phosphate creatinine ratio; *uPCR* urinary phosphate creatinine ratio. Source data are available online[21].

when compared to OM. Similarly, in the age stratum of (*children ≥ 3 years old*, the total energy, carbohydrate, and fat intake was comparable among groups. Both groups adhering to plant-based diets (VN and VG) had significantly higher intake of fiber and consumed less cholesterol and saturated fats (VN«VG) compared with the OM group. The protein intake was lower in VN compared to OM. Micronutrient intake was comparable among all groups except selenium, whose intake was lower in both VN and VG.

Among adults, all groups had comparable total energy, sugar, protein, and fat intake. As expected, VN participants had a lower intake of saturated fats and cholesterol (VN < VG < OM) and a higher intake of fiber (VN > VG > OM) compared to the OM group. Carbohydrate intake was higher in VN only (VN vs OM). Concerning micronutrients, the VN group had higher intakes of magnesium, zinc, and iron than both the VG and OM groups. VN and VG groups had lower intakes of iodine and selenium than OM.

### Supplementation habits
Micronutrients were supplemented by many study participants in the form of dietary supplements, but the exact dose is generally very difficult to quantify. The diet record may not reflect year-round supplementation and may underreport overall intake in irregularly supplementing persons. Therefore, we used a qualitative approach in surveying individual nutrient use among the study participants. The results are summarized in Supplementary Table 7. The groups differed significantly in supplementation habits, namely in supplementing B12, vitamin D, and n-3 fatty acids. A high proportion of VN and VG supplemented vitamin B12 across all age strata; OM did not supplement B12 for any participants, except one. VN also supplemented n-3 fatty acids significantly more than OM. For all groups, there was a high proportion of individuals who supplemented vitamin D; however, the supplementation was significantly more prevalent in VN and VG when compared to OM, with the exception of children ≥3 years old where significance was reached in VG vs OM only.

### Clinical variables as diet predictors
To explore how dietary patterns influence clinical characteristics, we employed elastic net logistic regression to determine whether the clinical characteristics could effectively discriminate between different diet groups. This approach provides insight into the extent to which diet shapes health outcomes, offering a predictive perspective on the role of diet in determining clinical profiles.

For both adults and children, we first fitted a *baseline model* incorporating basic subject characteristics (age, sex, and, for children, breastfeeding status) as predictors. We then expanded the analysis with a *complete model*, incorporating all clinical characteristics. The predictive capacity of each set of clinical variables was estimated as the difference between the discriminative capacity of complex and baseline models, expressed as a

## Table 4 | Clinical variables in adults among dietary groups

|  | Vegan | Vegetarian | Omnivore |
|---|---|---|---|
| Anthropometrics |  |  |  |
| BMI (kg/m$^2$) | 22.6 (20.8; 25.5) | 22.9 (21.5; 26.1) | 24.5 (22.4; 26.6) |
| Waist-to-hip ratio | 0.78 (0.74; 0.82) | 0.78 (0.73; 0.85) | 0.79 (0.75; 0.85) |
| Hand grip (kg) | 40 (31; 52) | 43 (32; 51) | 38 (30; 54) |
| Body fat (%) | 21 (16; 26) | 23 (16; 29) | 21 (18; 29) |
| Body fat (kg) | 14 (10; 18) | 17 (11; 23) | 17 (12; 21) |
| Free Fat Mass (kg) | 53 (45; 65) | 56 (49; 64) | 54 (48; 68) |
| Blood pressure |  |  |  |
| SBP (mmHg) | 121 (113; 134) | 121 (111; 134) | 125 (116; 139) |
| DBP (mmHg) | 75 (70; 80) | 79 (73; 86) | 79 (70; 83) |
| Glucose and lipid metabolism |  |  |  |
| FPG (mmol/l) | 4.5 (4.1; 4.7) | 4.4 (4.2; 4.7) | 4.5 (4.3; 4.8) |
| TC (mmol/l) | 4.2 (3.7; 4.7) | 4.3 (3.8; 5.0) | 4.9 (4.3; 5.4) |
| C-HDL (mmol/) | 1.4 (1.2; 1.7) | 1.4 (1.2; 1.7) | 1.5 (1.2; 1.7) |
| C-LDL (mmol/l) | 2.2 (1.9; 2.8) | 2.4 (2.0; 3.1) | 2.8 (2.4; 3.3) |
| TG (mmol/l) | 0.7 (0.6; 1.0) | 0.8 (0.7; 1.0) | 0.9 (0.6; 1.1) |
| Biogenic elements |  |  |  |
| Magnesium (mmol/l) | 0.81 (0.77; 0.85) | 0.81 (0.78; 0.83) | 0.79 (0.77; 0.87) |
| Selenium (mmol/l) | 0.97 (0.75; 1.18) | 0.97 (0.67; 1.18) | 1.00 (0.92; 1.17) |
| Zinc (mmol/l) | 12.1 (10.8; 13.7) | 13.1 (11.7; 14.1) | 14.1 (12.9; 15.6) |
| Iron (µmol/l) | 19 (15; 25) | 15 (12; 25) | 20 (16; 24) |
| Iron metabolism |  |  |  |
| TIBC (µmol/l) | 68 (65; 76) | 70 (62; 77) | 68 (62; 74) |
| Ferritin (µg/l) | 26 (14; 35) | 25 (16; 50) | 38 (23; 82) |
| Transferrin (g/l) | 2.7 (2.6; 3.0) | 2.7 (2.5; 3.1) | 2.7 (2.5; 2.9) |
| Transferrin saturation (%) | 29 (21; 36) | 22 (17; 39) | 28 (24; 36) |
| sTfR Index | 0.8 (0.7; 1.0) | 0.9 (0.6; 1.2) | 0.7 (0.6; 0.9) |
| STFR (mg/l) | 1.2 (1.0; 1.3) | 1.3 (1.1; 1.5) | 1.2 (1.1; 1.3) |
| Hemoglobin (g/l) | 145 (136; 155) | 144 (134; 154) | 147 (137; 154) |
| MCV (fl) | 89.7 (87.1; 91.7) | 86.7 (85.1; 89.5) | 87.6 (86.0; 89.8) |
| Bone metabolism |  |  |  |
| Calcium (mmol/l) | 2.46 (2.42; 2.51) | 2.45 (2.41; 2.50) | 2.44 (2.38; 2.52) |
| Phosphorus (mmol/l) | 1.08 (1.01; 1.21) | 1.12 (0.97; 1.26) | 1.12 (1.06; 1.21) |
| uCa (mmol/l) | 0.6 (0.3; 1.3) | 0.80 (0.40 1.5) | 1.0 (0.6; 1.55) |
| uCCR (mg/mg) | 0.1 (0.1; 0.2) | 0.2 (0.1; 0.3) | 0.2 (0.1; 0.3) |
| uPCR (mg/mg) | 1.2 (0.7; 1.5) | 1.5 (0.8; 2.0) | 1.7 (1.3; 2.2) |
| PTH (pmol/l) | 3.3 (2.7; 4.1) | 3.0 (2.2; 3.4) | 3.0 (2.4; 4.0) |
| CTx (µg/l) | 0.4 (0.3; 0.6) | 0.4 (0.3; 0.5) | 0.4 (0.3; 0.6) |
| P1NP (µg/l) | 52 (40; 72) | 43 (37; 61) | 45 (33; 59) |
| Vitamin D (nmol/l) | 76 (65; 92) | 73 (59; 88) | 67 (52; 78) |
| Iodine and One-Carbon metabolism |  |  |  |
| UIC (µg/l) | 91 (46; 147) | 94 (42; 170) | 120 (81; 202) |
| uICR (µg/g) | 144 (67; 254) | 109 (66; 243) | 181 (111; 328) |
| Active B12 (pmol/l) | 90 (66; 124) | 75 (54; 106) | 84 (71; 109) |
| Homocysteine (µmol/l) | 14.2 (12.5; 17.1) | 15.8 (13.0; 21.5) | 15.2 (11.9; 18.0) |
| MMA (nmol/l) | 170 (135; 214) | 236 (161; 305) | 194 (160; 233) |

## Table 4 (continued) | Clinical variables in adults among dietary groups

|  | Vegan | Vegetarian | Omnivore |
|---|---|---|---|
| Folate (µg/l) | 12.6 (9.7; 16.1) | 12.9 (10.1; 15.4) | 9.2 (7.3; 11.6) |
| Others |  |  |  |
| Urea (mmol/l) | 4.0 (3.2; 4.5) | 4.4 (3.8; 5.1) | 4.8 (4.0; 5.7) |
| Creatinine (µmol/l) | 64 (56; 72) | 66 (58; 78) | 70 (59; 81) |
| Uric acid (µmol/l) | 300 (247; 347) | 290 (246; 350) | 306 (254; 363) |

Medians (25th; 75th percentile) of clinical variables in vegan (VN), vegetarian (VG), and omnivorous (OM) adults. *Active B12* Holotranscobalamine, *BMI* Body mass index, *CTx* beta cross laps, *FPG* Fasting plasma glucose, *C-HDL* HDL cholesterol, *DBP* diastolic blood pressure, *IGF-1* insulin-like growth factor 1, *C-LDL* LDL cholesterol, *MCV* Mean corpuscular volume, *MMA* Methylmalonic acid, *perc* percentile, *SBP* systolic blood pressure, *sTfR Index* Soluble transferrin receptor/log Ferritin Index, *STFR* Soluble transferrin receptor, *PTH* Parathormone, *P1NP* Procollagen type I aminoterminal propeptide, *UIC* Urine Iodine Concentration, *TC* total cholesterol, *TIBC* total iron binding capacity, *TG* Triglycerides, *uCa* urine calcium concentration, *uCCR* urinary calcium creatinine ratio, *uICR* urinary iodine creatinine ratio, *uPCR* urinary phosphate creatinine ratio. Source data are available online[21].

difference between out-of-sample areas under ROC curves of both models (AUC_gain/loss) (Table 7).

Generally, we were able to discriminate between VN and OM reliably. In adults, the out-of-sample AUC of the complete model reached 0.86, whereas it was only 0.54 in the baseline model, with a mean AUC gain of 0.33. In children, the mean AUCs of the baseline and complete models were similar, 0.54 and 0.85, respectively. The discrimination between VG vs. OM or VN vs. VG was poor in both adult and child cohorts.

We further wanted to know to what extent the difference between dietary groups is driven by the use of dietary supplements. Therefore, we fitted a *reduced model* that excluded clinical outcomes primarily affected by supplementation from the predictors (MMA, homocysteine, vitamin B12, vitamin D, s_iron, ferritin binding capacity, s_ferritin, s_transferin, transferrin saturation, soluble transferrin receptor, soluble transferrin receptor/ferritin ratio, urinary iodine, uCa, uCCR, uICR, uPCR). The performance of the reduced model was substantially diminished in children, indicating the strong influence of supplement use on the difference between vegans and omnivores in this age stratum. In contrast, the AUC_gain of *reduced* vs *baseline* model remained significant (0.79 vs 0.54, $p = 0.018$) in adults, suggesting the strong effect of the diet itself in this cohort.

## Discussion

Here, we present cross-sectional data from the Czech Republic, focusing on families in which all household members follow the same dietary pattern (VN, VG, and OM). The major findings of the study are:

(1) No significant differences in growth characteristics observed in children between dietary groups;

(2) Trend towards lower BMI and fat mass and lower total and LDL cholesterol and diastolic blood pressure in adult vegans;

(3) Lower serum concentrations of total and LDL cholesterol levels in vegan children when compared to omnivores;

(4) Lower urinary iodine and higher folate, vitamin B12, and vitamin D levels in vegan children when compared to omnivores;

(5) In children, age was the most influential covariate, whereas in adults, it was sex. Clustering within the family was found for many variables, the most prominent being height, HDL cholesterol, B12, PTH, uric acid, and vitamin D for children, and selenium and urinary iodine for adults.

(6) Dietary groups of VN vs OM were reliably discriminated in both adults and children. While in children this difference was largely driven by supplement-related biomarkers, in adults it persisted even after excluding supplementation effects, showing the influence of the diet itself.

Vegan diet, well-planned and properly supplemented with specific diet could be nutritionally adequate to support growth and development in vegan children[22–24]. Nevertheless, the evidence regarding the growth and development of children who follow the respective diets from birth is still

**Table 5 | Differences between diet groups across outcomes in children, based on a robust mixed-effects model (rLME)**

| | VN to OM | | VG to OM | | VN to VG | |
|---|---|---|---|---|---|---|
| | mean diff | *p*-val | mean diff | *p*-val | mean diff | *p*-val |
| Anthropometrics | | | | | | |
| Body weight (percentile) | -9.23 | 0.137 | −3.6 | 0.614 | -5.63 | 0.380 |
| Body height (percentile) | -7.97 | 0.231 | 0.86 | 0.911 | -8.83 | 0.197 |
| BMI (percentile) | −2.32 | 0.705 | −2.79 | 0.694 | 0.47 | 0.940 |
| Weight/height (percentile) | −3.17 | 0.575 | −3.66 | 0.572 | 0.49 | 0.933 |
| Glucose and lipid metabolism | | | | | | |
| FPG (mmol/l) | 0.01 | 0.920 | 0.05 | 0.648 | −0.04 | 0.700 |
| TC (mmol/l) | −0.32 | 0.037 | −0.18 | 0.308 | −0.14 | 0.409 |
| HDL-C (mmol/) | −0.01 | 0.883 | −0.06 | 0.500 | 0.05 | 0.545 |
| LDL-C (mmol/l) | −0.25 | 0.029 | 0.08 | 0.555 | −0.33 | 0.009 |
| log$_2$TG (mmol/l) | −0.01 | 0.957 | −0.31 | 0.088 | 0.3 | 0.071 |
| Biogenic elements (serum) | | | | | | |
| Magnesium (mmol/l) | 0.03 | 0.030 | 0.03 | 0.091 | 0 | 0.845 |
| log$_2$Selenium (mmol/l) | −0.08 | 0.239 | −0.01 | 0.898 | −0.07 | 0.345 |
| log$_2$Zinc (mmol/l) | −0.96 | 0.059 | −0.65 | 0.276 | −0.31 | 0.586 |
| Iron (µmol/l) | 1.81 | 0.189 | −0.77 | 0.625 | 2.58 | 0.087 |
| Iron metabolism | | | | | | |
| log$_2$Ferritin (µg/l) | −0.25 | 0.130 | −0.19 | 0.314 | −0.06 | 0.746 |
| Transferrin (g/l) | −0.17 | 0.024 | −0.07 | 0.407 | −0.1 | 0.239 |
| Transferrin saturation (%) | 3.82 | 0.052 | −0.03 | 0.989 | 3.85 | 0.074 |
| log$_2$sTfR Index | 0.03 | 0.695 | 0.08 | 0.447 | −0.04 | 0.659 |
| log$_2$STFR (mg/l) | −0.03 | 0.572 | 0.01 | 0.840 | −0.04 | 0.468 |
| Hemoglobin (g/l) | −0.44 | 0.810 | 1.14 | 0.591 | −1.58 | 0.434 |
| MCV (fl) | 2.97 | <0.001 | −0.13 | 0.892 | 3.1 | <0.001 |
| Bone health | | | | | | |
| Calcium (mmol/l) | −0.01 | 0.556 | 0.01 | 0.777 | −0.02 | 0.388 |
| Phosphorus (mmol/l) | −0.03 | 0.469 | 0 | 0.941 | −0.03 | 0.453 |
| log$_2$uCa (mmol/l) | −0.42 | 0.323 | 0.03 | 0.940 | −0.46 | 0.278 |
| log$_2$uCCR (mg/mg) | 0.02 | 0.948 | 0.19 | 0.660 | −0.17 | 0.660 |
| uPCR (mg/mg) | −2.12 | <0.001 | −0.63 | 0.160 | −1.48 | <0.001 |
| PTH (pmol/l) | 0.38 | 0.204 | −0.02 | 0.962 | 0.39 | 0.223 |
| log$_2$CTx (ug/l) | 0.06 | 0.464 | 0.08 | 0.345 | −0.03 | 0.715 |
| P1NP (ug/l) | 41.88 | 0.290 | 63.17 | 0.182 | -21.29 | 0.622 |
| log$_2$Vitamin D (nmol/l) | 0.28 | 0.006 | 0.26 | 0.033 | 0.02 | 0.864 |
| Iodine and One-Carbon metabolism | | | | | | |
| log$_2$UIC (µg/l) | -69.96 | 0.001 | -31.96 | 0.182 | -38 | 0.081 |
| log$_2$uICR (µg/µg) | −0.38 | 0.178 | −0.31 | 0.323 | −0.07 | 0.799 |
| log$_2$Active B12 (pmol/l) | 0.3 | 0.154 | −0.31 | 0.145 | 0.61 | <0.001 |
| log$_2$Homocysteine (µmol/l) | −0.3 | <0.001 | −0.11 | 0.268 | −0.19 | 0.053 |
| log$_2$MMA (nmol/l) | −0.79 | <0.001 | −0.26 | 0.141 | −0.53 | 0.001 |

**Table 5 (continued) | Differences between diet groups across outcomes in children, based on a robust mixed-effects model (rLME)**

| | VN to OM | | VG to OM | | VN to VG | |
|---|---|---|---|---|---|---|
| | mean diff | *p*-val | mean diff | *p*-val | mean diff | *p*-val |
| Folate (µg/l) | 4.3 | <0.001 | 4.81 | <0.001 | −0.51 | 0.620 |
| Growth factor and others | | | | | | |
| log$_2$IGF-1 (ug/l) | −0.16 | 0.288 | 0.01 | 0.951 | −0.17 | 0.303 |
| Urea (mmol/l) | −0.55 | 0.010 | −0.17 | 0.488 | −0.38 | 0.108 |
| log$_2$Creatinine (µmol/l) | −0.11 | 0.042 | −0.06 | 0.335 | −0.05 | 0.400 |
| Uric acid (µmol/l) | −2.38 | 0.831 | −4.94 | 0.709 | 2.56 | 0.831 |

The difference among groups was assessed using a robust mixed-effects model (rLME) with 'robustlmm' R package. *P*-values were derived from Wald tests of pairwise contrasts of estimated marginal means using *emmeans*. The models adjusted for potential confounders, including age, sex, and breastfeeding-related variables; when relevant, supplementation status or birth weight were also included. Family was included as a random intercept to account for within-family clustering. All *P*-values are two-sided; no adjustment for multiple comparisons was applied. For this analysis, all children were grouped together. *Active B12* Holotranscobalamine, *BMI* Body mass index, *CTx* beta cross laps, *FPG* Fasting plasma glucose, *C-HDL* HDL cholesterol, *DBP* diastolic blood pressure, *IGF-1* insulin-like growth factor 1, *C-LDL* LDL cholesterol, *MCV* Mean corpuscular volume, *MMA* Methylmalonic acid, *perc* percentile, *SBP* systolic blood pressure, *sTfR Index* Soluble transferrin receptor/log Ferritin Index, *STFR* Soluble transferrin receptor, *PTH* Parathormone, *P1NP* Procollagen type I aminoterminal propeptide, *UIC* Urine Iodine Concentration, *TC* total cholesterol, *TIBC* total iron binding capacity, *TG* Triglycerides, *uCa* urine calcium concentration, *uCCR* urinary calcium creatinine ratio, *uICR* urinary iodine creatinine ratio, *uPCR* urinary phosphate creatinine ratio. Source data are available online[21].

insufficient. There are signals that vegan children exhibit a lower BMI[25] and height[12] compared to their omnivorous counterparts. Lower caloric density and higher fiber content of plant-based diets, which often lead to lower net energy intake, were among the plausible explanations[26]. While higher fiber intake is generally considered a major benefit of plant-based diets[9,23,26], it may cause increased satiety and lead to inadequate total energy and protein intake in children, due to their smaller stomach volume. In addition, excess fiber may interfere with the absorption of fats and minerals and is also associated with higher intakes of substances that impair the absorption of some critical nutrients in a plant-based diet[27]. Despite these concerns, we found similar caloric intakes across age strata, though we confirmed that vegans and vegetarians generally had higher intakes of fiber.

Despite some differences in nutrient intake, we found no significant difference in the growth characteristics of children in adjusted models. Nevertheless, vegan children <3 years old were generally smaller and shorter when compared to other groups, which could be of clinical relevance. These findings corroborate Polish study outcomes performed on children aged 5–10 years[12] where vegan children were found to be shorter, but contrast the results of the Finnish study outcomes on preschool vegan children (median age 3.5 years), which showed no differences among groups[28]. In multi-variable analyses, we showed that the primary predictor of height in children was family background rather than dietary group. This finding may account for the discrepancies in the previously published results. In line, we also found that differences in IGF-1 were driven by family and age instead of diet.

Altogether, we found no indices of growth challenge in vegan and vegetarian children who followed the respective diet from birth, but a strong correlation within the family. Findings that corroborate previous evidence that genetic factors have a more significant influence on growth than environmental influences[29].

Cardiometabolic diseases are among the leading global health concerns, with diet being a significant environmental factor contributing to their rise[30–33]. Large cross-sectional and prospective cohort studies in Western countries such as Adventist Health studies, EPIC-Oxford, and UK Women´s Cohort, which included a significant proportion of vegan and vegetarian adults following plant-based diets, exhibited a lower LDL

## Table 6 | Differences between diet groups in adults, based on a robust mixed-effects model (rLME)

| | VN to OM | | VG to OM | | VN to VG | |
|---|---|---|---|---|---|---|
| | mean diff | *p*-val | mean diff | *p*-val | mean diff | *p*-val |
| **Anthropometrics** | | | | | | |
| BMI (kg/m$^2$) | −1.32 | 0.070 | −0.88 | 0.379 | −0.44 | 0.465 |
| Hand grip strength (kg) | 0.52 | 0.674 | 1.06 | 0.674 | −0.54 | 0.674 |
| Body fat (%) | −1.8 | 0.297 | −0.02 | 0.988 | −1.78 | 0.297 |
| log$_2$Body fat (kg) | −0.2 | 0.232 | −0.01 | 0.925 | −0.19 | 0.232 |
| log$_2$Free Fat Mass (kg) | −0.06 | 0.097 | −0.02 | 0.471 | −0.03 | 0.410 |
| **Blood pressure** | | | | | | |
| SBP (mm Hg) | −4.13 | 0.346 | −3.15 | 0.596 | −0.98 | 0.717 |
| DBP (mm Hg) | −2.32 | 0.295 | 1.96 | 0.295 | −4.28 | 0.032 |
| **Glucose and lipid metabolism** | | | | | | |
| FPG (mmol/l) | −0.09 | 0.711 | −0.07 | 0.749 | −0.01 | 0.879 |
| log$_2$TC (mmol/l) | −0.21 | <0.001 | −0.15 | 0.021 | −0.06 | 0.246 |
| C-HDL (mmol/) | −0.04 | 0.592 | −0.07 | 0.592 | 0.03 | 0.592 |
| C-LDL (mmol/l) | −0.5 | <0.001 | −0.32 | 0.041 | −0.18 | 0.142 |
| log$_2$TG (mmol/l) | −0.12 | 0.811 | −0.1 | 0.850 | −0.02 | 0.850 |
| **Biogenic elements (serum)** | | | | | | |
| Magnesium (mmol/l) | 0 | 0.889 | 0 | 0.889 | 0 | 0.889 |
| log$_2$Selenium (mmol/l) | −0.06 | 0.709 | −0.09 | 0.709 | 0.03 | 0.712 |
| log$_2$Zinc (mmol/l) | −0.22 | <0.001 | −0.13 | 0.041 | −0.09 | 0.076 |
| Iron (µmol/l) | 0.18 | 0.894 | −2.49 | 0.219 | 2.67 | 0.170 |
| **Iron metabolism** | | | | | | |
| log$_2$Ferritin (µg/l) | −0.74 | <0.001 | −0.55 | 0.007 | −0.2 | 0.235 |
| log$_2$Transferrin (g/l) | 0.03 | 0.719 | 0.03 | 0.719 | 0 | 0.911 |
| Transferrin saturation (%) | −0.36 | 0.862 | −4.2 | 0.157 | 3.84 | 0.157 |
| log$_2$sTfR Index | 0.25 | 0.006 | 0.3 | 0.006 | −0.05 | 0.554 |
| log$_2$STFR (mg/l) | 0.04 | 0.474 | 0.1 | 0.334 | −0.06 | 0.474 |
| Hemoglobin (g/l) | −1.09 | 0.996 | −1.1 | 0.996 | 0.01 | 0.996 |
| MCV (fl) | 1.66 | 0.014 | −0.82 | 0.251 | 2.48 | <0.001 |
| **Bone health** | | | | | | |
| Calcium (mmol/l) | 0 | 0.865 | 0.01 | 0.865 | 0 | 0.865 |
| Phosphorus (mmol/l) | −0.03 | 0.470 | 0 | 0.968 | −0.03 | 0.470 |
| log$_2$uCa (mmol/l) | −0.51 | 0.060 | −0.15 | 0.634 | −0.37 | 0.186 |
| log$_2$uCCR (mg/mg) | −0.16 | 0.362 | −0.01 | 0.970 | −0.16 | 0.398 |
| uPCR (mg/mg) | −0.62 | <0.001 | −0.35 | 0.021 | −0.28 | 0.042 |
| PTH (pmol/l) | 0.37 | 0.106 | −0.15 | 0.491 | 0.52 | 0.025 |
| log$_2$CTx (ug/l) | 0.05 | 0.719 | −0.1 | 0.719 | 0.15 | 0.719 |
| log$_2$P1NP (ug/l) | 0.14 | 0.515 | 0.01 | 0.950 | 0.13 | 0.515 |
| log$_2$Vitamin D (nmol/l) | 0.24 | 0.007 | 0.14 | 0.182 | 0.1 | 0.182 |
| **Iodine and One-Carbon metabolism** | | | | | | |
| log$_2$UIC (µg/l) | -28.58 | 0.401 | −27.65 | 0.431 | −0.92 | 0.962 |
| log$_2$uICR (µg/mg) | −0.29 | 0.360 | −0.67 | 0.074 | 0.37 | 0.252 |

## Table 6 (continued) | Differences between diet groups in adults, based on a robust mixed-effects model (rLME)

| | VN to OM | | VG to OM | | VN to VG | |
|---|---|---|---|---|---|---|
| | mean diff | *p*-val | mean diff | *p*-val | mean diff | *p*-val |
| log$_2$Active B12 (pmol/l) | −0.09 | 0.605 | −0.3 | 0.212 | 0.21 | 0.272 |
| log$_2$Homocysteine (µmol/l) | −0.04 | 0.538 | 0.11 | 0.354 | −0.15 | 0.140 |
| log$_2$MMA (nmol/l) | −0.13 | 0.144 | 0.25 | 0.036 | −0.38 | <0.001 |
| Folate (µg/l) | 3.27 | 0.001 | 3.27 | 0.002 | 0 | 0.997 |
| **Others** | | | | | | |
| log$_2$Urea (mmol/l) | −0.29 | <0.001 | −0.1 | 0.205 | −0.19 | 0.013 |
| log$_2$Creatinine (µmol/l) | −0.13 | 0.003 | −0.07 | 0.158 | −0.06 | 0.158 |
| Uric acid (µmol/l) | −5.57 | 0.819 | −8.03 | 0.819 | 2.46 | 0.819 |

The difference among groups was assessed using a robust mixed-effects model (rLME) with 'robustlmm' R package. P-values were derived from Wald tests of pairwise contrasts of estimated marginal means using *emmeans*. The models adjusted for potential confounders, including age and sex; supplementation status was included when relevant. Family was included as a random intercept to account for within-family clustering. All P-values are two-sided; no adjustment for multiple comparisons was applied. *Active B12* Holotranscobalamine, *BMI* Body mass index, *CTx* beta cross laps, *FPG* Fasting plasma glucose, *C-HDL* HDL cholesterol, *DBP* diastolic blood pressure, *IGF-1* insulin-like growth factor 1, *C-LDL* LDL cholesterol, *MCV* Mean corpuscular volume, *MMA* Methylmalonic acid, *perc* percentile, *SBP* systolic blood pressure, *sTfR* Index, Soluble transferrin receptor/log Ferritin Index, *STFR* Soluble transferrin receptor, *PTH* Parathormone, *P1NP* Procollagen type I aminoterminal propeptide, *UIC* Urine Iodine Concentration, *TC* total cholesterol, *TIBC* total iron binding capacity, *TG* Triglycerides, *uCa* urine calcium concentration, *uCCR* urinary calcium creatinine ratio, *uICR* urinary iodine creatinine ratio, *uPCR* urinary phosphate creatinine ratio. Source data are available online[21].

cholesterol, BMI, and blood pressure, that translated to lower prevalence of obesity and a reduced risk of ischemic heart disease and type 2 diabetes (T2D) compared to omnivores with similar backgrounds[34]. The long-term beneficial effects of a vegan diet have inspired its use for patients with metabolic syndrome and T2D. According to a meta-analysis of 11 short-term intervention trials, a healthy vegan diet led to better results in cardiometabolic parameters (BMI, total and LDL cholesterol, glycated hemoglobin) when compared to a healthy standard diet[35]. With the rising incidence of obesity across all age groups, the accumulation of cardiometabolic risk factors is shifting to younger populations, and the COVID-19 pandemic has even accelerated this trend[36]. Therefore, evaluating cardiovascular risk factors and screening for high blood pressure in childhood has been a matter of debate [36]. Only recently, the results of an extensive pediatric cohort study to evaluate cardiovascular risk, the International Childhood Cardiovascular Cohort Consortium, demonstrated strong associations of childhood risk factors with major cardiovascular events in midlife[37]. From this perspective, a reduction in risk-factor levels as early as in childhood may have the potential to lower the incidence of premature cardiovascular disease.

The difference in CVD risk-associated markers increased both with age ('children<3 years' <'children≥3 years' <'adults') and across dietary groups (VN < VG < OM). Both vegan children and adults had lower total and LDL cholesterol in adjusted models compared to omnivores. Adult VN had a lower BMI when compared to OM (p = 0.07). Though the long-term follow-up of people on respective diets is warranted, our data suggest that adopting plant-based diets as early as childhood may be among the counter-measures to the trend of increased CVD risk and premature death in adulthood.

There are indices that the bone health of vegans may be impaired, based on both cross-sectional[38,39], and prospective studies[14]. Deficiencies in critical nutrients for bone formation and mineralization, such as calcium and vitamin D, often precede a decline in bone density and fractures[14,40–42]. However, bone health may also be influenced by other nutrients such as zinc, vitamin B12, and omega-3 fatty acids that are less abundant or absent

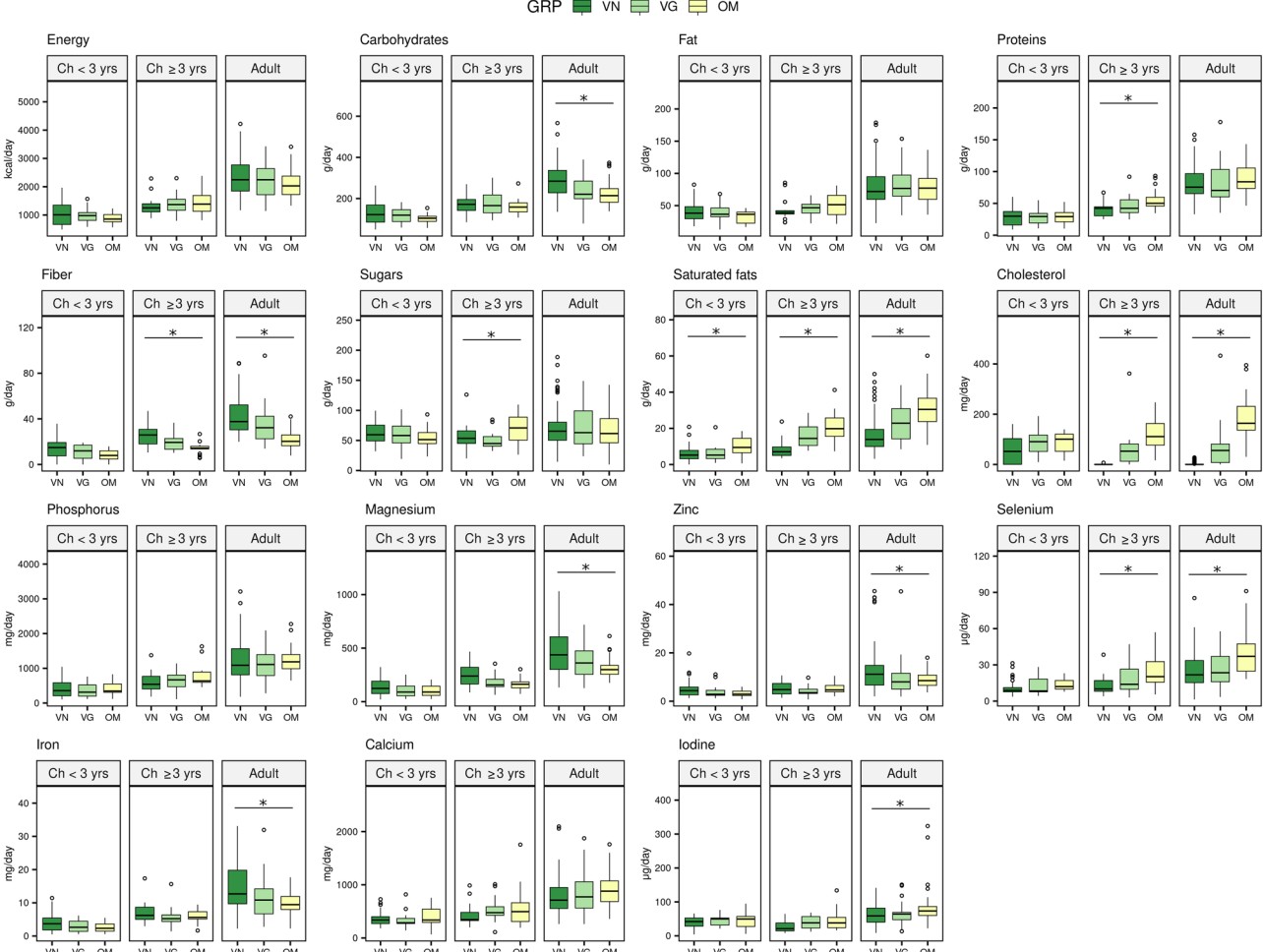

**Fig. 3 | Nutrient intake among dietary groups across age strata.** Vegan, vegetarian, and omnivore groups (VN, VG, and OM) age strata of children <3 and > 3 years, and adults. * *p* < 0.05 across the three groups evaluated using the Kruskal-Wallis test. GRP, group; Ch, children. adult: 38 (OM), 34 (VG), 82 (VN), from 19 (OM), 17 (VG) and 42 (VN) families; ch <3: 16 (OM), 12 (VG), 37 (VN), from 16 (OM), 12 (VG) and 36 (VN) families; ch >3: 17 (OM), 15 (VG), 16 (VN) from 11 (OM), 10 (VG) and 16 (VN) families. Source data are available online[21].

in plant-based diets[40,42]. It has been previously shown that both adult and child vegans have lower bone mineral density (BMD)[12]. However, it has been debated that the lower BMD reflects lower BMI in vegans[43]. Nevertheless, results from an extensive EPIC-Oxford cohort study indicated that these risks eventually led to a higher incidence of all-site fractures in vegans[14]. Accumulating evidence was reviewed in a meta-analysis, showing that both vegetarians and vegans have higher risks of fractures[42]. Higher bone turnover triggered by lower availability of minerals and increased action of PTH may precipitate the fracture, and findings of higher P1NP were already reported in vegans[44].

Despite the concerns of the vegan diet on bone health, we have not observed a higher reported prevalence of any bone fractures or fractures most commonly associated with osteoporosis (spine, hip, distal radius, and proximal humerus) in adult vegans. Even though we observed a higher average P1NP in vegan adults and children ≥ 3 years on descriptive analysis, the difference was insignificant when adjusted for confounders. Indeed, P1NP is dependent on age, and this needs to be adjusted for, namely, in children's groups. Higher vitamin D levels in vegans across all age strata were found, attributable to higher regular supplementation of vitamin D. Calcium intake was comparable among groups. The same is true for its bioavailability from plant sources[45,46], as we saw no differences between groups and across age in markers of calcium saturation (urinary calcium losses and PTH).

Consistent significant trends (VN < VG < OM) in lower phosphate losses in both children and adults were found. This is in line with sufficient saturation of the bone with calcium and probably reflects lower availability of phosphates from the plant proteins[47,48].

The interpretation of our results is limited by three factors. First, though we have analyzed calcium losses, we relied on single-spot urine measurements. Second, we have not assessed bone quality, so we cannot comment on any decline in bone mass. Third, the cross-sectional design did not allow for prospective analysis of incident fractures.

The findings do not support previous concerns regarding the adverse effects of vegan or vegetarian diets on bone health, provided that adequate calcium intake and vitamin D supplementation are ensured.

Iodine is perceived as a critical nutrient for vegan diets, as its content in plant sources is limited[49,50]. In the Czech Republic, salt is routinely fortified with iodine, which explains that compared to other European countries, iodine status in Czech school children and adults is considered sufficient (i.e. median standardized UIC > 100 μg/L)[51]. However, in the groups that lack major dietary sources of iodine, the deficiency may be an important safety concern. We and others have already shown that vegan children[9,15] as well as adults[52], are more likely to have lower iodine status.

In the current study, we found that in the children groups, iodine intakes were comparable among groups. On the other hand, both adult VN and VG had significantly lower iodine intake compared to OM.

**Table 7 | Accuracy metrics of predictive models based on clinical variables**

| | | AUC | | | AUC gain/loss | | | |
|---|---|---|---|---|---|---|---|---|
| | | baseline | complete | reduced | complete vs baseline | _p_val | reduced vs baseline | _p_val |
| all children | VN_OM | 0.54 [0.36, 0.69] | 0.86 [0.74, 0.95] | 0.65 [0.46, 0.8] | 0.33 [0.16, 0.51] | 0.002 | 0.12 [−0.09, 0.28] | 0.238 |
| | VG_OM | 0.57 [0.33, 0.78] | 0.59 [0.32, 0.85] | 0.42 [0.21, 0.64] | 0.02 [−0.28, 0.35] | 0.928 | −0.15 [−0.41, 0.11] | 0.242 |
| | VN_VG | 0.59 [0.39, 0.76] | 0.7 [0.52, 0.88] | 0.64 [0.47, 0.79] | 0.11 [−0.12, 0.37] | 0.337 | 0.05 [−0.16, 0.27] | 0.553 |
| adults | VN_OM | 0.54 [0.38, 0.71] | 0.85 [0.75, 0.94] | 0.79 [0.67, 0.9] | 0.31 [0.12, 0.51] | 0.002 | 0.24 [0.05, 0.42] | 0.018 |
| | VG_OM | 0.47 [0.3, 0.59] | 0.67 [0.49, 0.84] | 0.51 [0.33, 0.67] | 0.2 [−0.01, 0.42] | 0.062 | 0.04 [−0.18, 0.24] | 0.681 |
| | VN_VG | 0.55 [0.41, 0.73] | 0.66 [0.48, 0.8] | 0.66 [0.49, 0.79] | 0.11 [−0.12, 0.31] | 0.317 | 0.11 [−0.13, 0.29] | 0.321 |

The data are shown as the mean of 500 bootstrap resamplings and the 95% confidence interval (in brackets). Baseline model: Elastic Net logistic regression model using 'glmnet' R package, incorporating basic subject characteristics (age, sex, and, for children, breastfeeding status) as predictors. Complete model: Baseline model including all eligible clinical variables as predictors. Reduced model: Complete model excluding supplementation-dependent variables as predictors (methylmalonic acid, homocysteine, vitamin B12, vitamin D, iron, ferritin binding capacity, ferritin, transferrin, transferrin saturation, soluble transferrin receptor, soluble transferrin receptor/ferritin ration, urinary iodine, urinary calcium, urinary calcium to creatinine ratio, urinary iodine to creatinine ratio, urine phosphate to creatinine ratio). AUC gain/loss: AUC$_{complete model}$ − AUC$_{baseline or reduced model}$. _P_-values are two-sided and based on a nonparametric bootstrap test (500 resamples) comparing the out-of-sample AUC against 0.5 Sample sizes: children (OM: 44, VG: 37, VN: 61) and adults (OM: 50, VG: 45, VN: 92), recruited from 25 OM, 23 VG, and 47 VN independent families. Source data are available online[21].

Unexpectedly, the intakes were, on average, roughly half the daily recommended doses of 150 μg/day[53]. The dietary intake of iodine may have been underestimated due to insufficient coverage of iodine content in vegan-specific foods and regional-specific foods. Lower iodine intake translated to lower urinary iodine concentrations only in vegan children, where we also found the highest prevalence of potential iodine deficiency (i.e., UIC < 100 μg/L). Of note, all these children had TSH in the normal range, and none of them had TSH above the reference range. Though these findings corroborate with previous evidence[9,39], the results are limited by the measurement of single-spot urine. UIC, which is commonly used in clinical practice, may not fully capture iodine status due to inter-day variability in iodine excretion. Concerns about excessive iodine intake from seaweeds, which may be popular in the vegan population, were raised[54]. However, we have not identified any vegan subjects with excessive iodine intake in the current study.

Previous studies indicated that a vegan diet provides sufficient iron intake primarily sourced from plant-based foods in the form of non-heme iron[55,56]. However, non-heme iron absorption in vegan diets is susceptible to various inhibitors such as phytates and polyphenols, which collectively contribute to reduced iron absorption efficiency in vegan diets, despite adequate iron intake from plant sources[57]. Some studies on vegans have shown that low iron status did not correlate with lower iron intake[58–60].

In line with the evidence, we found comparable iron intake among children's groups or even higher iron intake in vegan adults, contrasting with significantly lower ferritin levels in adult vegans and vegetarians compared to omnivores. Of note, all ferritin values of the groups were on average within the reference range,s and the absolute differences were modest. Despite the lower ferritin levels in adult vegans, comparable hemoglobin levels were found among groups, findings corresponding with existing evidence[60].

In line with previous research[12,61], we found significantly higher corpuscular volumes of erythrocytes (MCV) in both children and adults, VN and V,G when compared to omnivores, with a clear trend of VG > VN > OM. Though it is well established that vitamin B12 deficiency relates to higher MCV[58] and low ferritin levels to lower MCV, we observed an opposite trend. Whether there is another diet-related mechanism beyond iron and B12 contributing to MCV is to be further studied. Though the differences were subtle and very likely below clinical relevance, MCV and Hb levels should not guide clinical decisions on iron or B12 deficiencies in vegans.

Families share dietary patterns, influencing the nutritional choices of all household members, especially children, during critical stages of growth and development [16]. We showed that family as a covariate influenced some group and age-related differences. Growth restriction is among the concerns in children on vegan diets[62]. Our findings indicated that both height and IGF-1 levels were associated with the family background, highlighting the importance of considering familial factors when assessing

anthropometric parameters in children. Furthermore, we found that in children, serum concentrations of PTH, vitamin D, and urinary iodine were related to the family background. In adults, family background impacted the ability to predict height, serum selenium and zinc levels, urinary iodine, and vitamins folate, B12, and D.

The observed differences among groups could be attributed to the influence of supplementation, as both the proportion of individuals taking vitamin B12 and D supplements and the levels of circulating biomarkers for these were higher in the vegan and vegetarian groups, respectively.

Data on associations of individuals' nutrient status and family risks are scarce, given the complexity of the research design, so further research is needed to replicate the results on a larger scale. Nevertheless, we conclude that family may be an important determinant of nutritional status, and dietary interventions should target the families and focus on parental education. A considerable number of 95 families were enrolled in the study. The family-based design is a key strength, an approach providing insight into the interplay between family eating habits and nutritional status. While there is ample evidence on the health risks and benefits of vegan diets among adults, children remain an under-represented population in nutritional research on the effects of vegan diets, particularly in studies evaluating the long-term effects of diets. We addressed the unmet need by enrolling children adhering to the respective diets from birth.

While there are strengths of the study, limitations need to be listed. The analysis is inherently descriptive, focusing on associations rather than causal effects. While causal inference exists on a continuum, our design does not allow for strong causal conclusions. There are errors inherent in the analysis of dietary intake. We relied on a 3-day prospective weighted record, which provided only a snapshot of a long-term intake; with its potential for reactivity, overall food intake may be underreported. Moreover, supplementation could be completely missed by individual records. We tried to overcome these by convenient enrollment across seasons and by qualitative (recall) assessment of the supplementation. Despite this approach, we were unable to produce any quantitative measure of micronutrient intake by supplementation, as the participants quite often could not recall exactly the manufacturer, nor quantify how often they used it.

To maintain sufficient statistical power in the mixed-effects analysis, the pediatric group was analyzed as a single group, which may have limited our ability to detect differences across distinct developmental stages, such as between toddlers and preschool-aged children. Lastly, a selection bias based on willingness to participate in such a study needs to be pointed out. Our participants were mainly well-educated and motivated families from urban areas, which may limit the generalizability of the outcomes. Future studies should focus on replicating these findings in prospective cohorts.

## Conclusion
Dietary habits strongly predict nutritional biomarkers, with familial factors contributing to interindividual variability. No significant differences are

**Article**

observed in growth characteristics across children's dietary groups. Age emerges as the most influential covariate among children, and sex among adults. Several variables cluster within families - including height, HDL cholesterol, vitamin B12, PTH, uric acid, and vitamin D in children, and selenium and urinary iodine in adults. While vegan diets offer cardiometabolic benefits, iodine deficiency remains prevalent in this group and represents a persisting nutritional concern.

## Data availability

All source data supporting the findings of this study are available online in the Zenodo repository https://doi.org/10.5281/zenodo.17250682[21].

## Code availability

The statistical code and detailed statistical methodology description are available online in an open GitHub repository: https://github.com/filip-tichanek/kompas_clinical[21].

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

## Acknowledgements

The study was supported by the Ministry of Health, Czech Republic, no. NU21-09-00362, the project LX22NPO5104, Funded by the European Union —Next Generation EU, and the Charles University grant support 260646/SVV/2023. The authors would like to express deep gratitude to the study participants whose contributions have made it possible to advance knowledge in the field.

## Author contributions

J.G., M.C., T.K., E.E. and P.D. were involved in the conceptualization, methodology, and data curation. F.T., P.P., T.K. and M.C. prepared the analytical plan and performed all statistical calculations. M.H., A.O., J.G., M.S., E.S., J.P. and D.H. conducted the clinical examinations. M.H., A.O., J.G. and M.C. drafted the manuscript. All authors were involved in writing, reviewing, and editing the final version of the manuscript.

## Competing interests

The authors declare no competing interests.

## Additional information

[1]Department of Internal Medicine, Kralovske Vinohrady University Hospital and Third Faculty of Medicine, Charles University, Prague, Czech Republic. [2]Department of Hygiene, Third Faculty of Medicine, Charles University, Prague, Czech Republic. [3]Department of Epidemiology, Third Faculty of Medicine, Charles University, Prague, Czech Republic. [4]Centre for Public Health Promotion, National Health Institute, Prague, Czech Republic. [5]Institute for Clinical and Experimental Medicine, Prague, Czech Republic. [6]Faculty of Science, Charles University, Prague, Czech Republic. [7]Department of Pediatrics, Kralovske Vinohrady University Hospital and Third Faculty of Medicine, Charles University, Prague, Czech Republic. [8]Department of Epidemiology, MedUni, Vienna, Austria. [9]These authors contributed equally: Marina Heniková, Anna Ouřadová. [10]These authors jointly supervised this work: Monika Cahová, Jan Gojda. ✉e-mail: jan.gojda@lf3.cuni.cz

