## [Transparent Peer Review file · Communications Medicine]

Dietary intake, nutritional status, and health outcomes among vegan, vegetarian, and omnivorous Czech families

Corresponding Author: Professor Jan Gojda

Version 0:

Reviewer comments:

Reviewer #1

(Remarks to the Author)

Reviewer: James P Goode, PhD

Thank you for the opportunity to review this manuscript. I have expertise in nutritional epidemiology, including plant-based diets and health outcomes. While I also have training in biostatistics, I would happily defer to a statistical reviewer with more expertise. The comprehensiveness of my comments speaks to the potential value of the underlying data. I commend the authors on their efforts. I look forward to future publications from the KOMPAS study.

Summary

This paper performed a cross-sectional analysis of baseline data from the KOMPAS study, a prospective cohort study of Czech families where all members of the household followed the same diet. While this is chiefly a descriptive study, certain elements are consistent with an analytical study. The main aim was to describe the dietary intakes, nutritional status, and health-related measures of three diet groups (vegan, lacto-ovo-vegetarian, omnivore) by three age groups (adults, infants/toddlers, pre-schoolers/schoolers). An additional aim was to explore the 'predictive potential' of diet group and family unit in relation to observed variation in diet and health-related measures.

Impressions

This paper presents novel data on an important topic. That is, children who have followed a vegan or vegetarian diet since birth in relation to diet and health-related outcomes within a unified family unit. Though a modestly sized cross-sectional study, the findings will still likely be of interest. Specific comments regarding the manuscript are outlined below for your consideration. While the minor comments are mostly related to clarity and reporting inconsistencies, the major comments are potentially more substantial.

--- Major comments ---

1) The length of the manuscript, large number of diet and health-related measures, and range of statistical analyses (with some being possibly underpowered, see comments #4 and #5) hinders readability and comprehensibility. This is compounded by there being nine comparison groups for most analyses (three diet groups by three age groups). This is a challenge to summarise succinctly and in an accessible manner, while also describing/discussing each aspect of the manuscript in sufficient detail. I also struggled to fully comprehend all aspects of the statistical analyses in relation to the fourth stated aim (specifically, the mixed-effects and elastic net models). The main text is over 6000 words. I wonder whether there is more than one paper's worth of data here. While it is certainly possible to present everything, a shorter, more focussed paper would likely be easier to read and more impactful.

2) The Methods section reported in the main manuscript is underdeveloped. While a study protocol is referenced, in addition to supplemental methods, key information that should ideally be in the main manuscript is missing. Examples include (1) how vegan, vegetarian and omnivorous diets were defined, (2) the sampling/recruitment strategy employed, (3) how missing

data were handled, (4) whether estimates of nutrient intake include supplementation, etc. This is not an exhaustive list. The STROBE-nut checklist may help with this: <https://www.equator-network.org/reporting-guidelines/strobe-nut/>. Also, the results of certain measures are presented but the associated methodology is not described (e.g., handgrip strength). In this manuscript, I would also consider placing the Methods section after the Introduction (and relocating any relevant information currently in the Introduction/Results to the Methods). That is, a more traditional IMRaD structure.

3) Tables that present descriptive data include p values under a test hypothesis of no difference (e.g., Tables 2 to 4). This practice is widespread and generally accepted, but STROBE reporting guidelines discourage the use of significance tests in descriptive tables (see item 14a in the full explanation/elaboration document): <https://pubmed.ncbi.nlm.nih.gov/17941715/>. While there may be a 'significant' difference between two groups, one must ultimately consider the magnitude and clinical significance of this difference. If significance testing is retained, an alpha level would need to be stated (e.g., 0.05 or 5%) and, if applicable, how this relates to the methods used for adjusting for multiple comparisons. Tables 2 to 4 also include p values from a covariate-adjusted mixed-effects model that combined all children, but I wonder whether this is necessary? The p values are very similar to those from the Mann-Whitney U tests. In addition, is there a need to adjust for potential confounders? This article may be of interest: <https://www.nature.com/articles/s41416-020-1019-z>. And is it necessary to include Supplemental Tables 11 to 13? They repeat data from Tables 2 to 4 and don't seem to add much.

4) Results (Evaluation of nutritional risks in marginal subgroups): Issues of whether to adjust for potential confounders aside, I'm wondering whether the sample size is sufficient to produce meaningful/stable estimates of different points in the distribution (20th, 50th, and 80th percentile) for use in quantile regression modelling. For example, in children aged >3 years, there were only 35 vegans, 13 vegetarians, and 21 omnivores, with an even smaller number for certain variables due to missingness.

5) Results (Clinical variables as diet predictors): Like my previous comment, the use of the receiver operating characteristic curve (as a measure of discrimination) may also be underpowered, particularly among children (aged <3 or >3 years). The 95% confidence intervals for the area under the curve vary widely from well under 0.5 (no discrimination) to about 0.8 to 0.9 (with 1 being the theoretical maximum). This lack of precision limits the ability to draw meaningful conclusions. Although, there may be a signal when comparing vegans to omnivores (base vs. full model), but the issue of precision remains.

6) Can a single spot urine be used to reliably estimate the prevalence of iodine deficiency? There is wide day-to-day variation in iodine intake and urinary iodine only represents recent intake. While this may be useful at the population level, is a single measure of urinary iodine a reliable way of screening individuals for iodine deficiency? See the following article: <https://pubmed.ncbi.nlm.nih.gov/19888863/>.

--- Minor comments ---

Title

7) L1-2: The title could be more specific/informative. For example, 'Dietary intake, nutritional status, and health outcomes among vegan, vegetarian, and omnivorous Czech families: a cross-sectional analysis'.

Abstract

8) L36: What are the main safety concerns? Nutrient deficiencies? Impaired growth and development during childhood? Giving an example or two may be helpful. For example, '...safety concerns such as nutrient deficiencies, particularly during critical growth periods...'

9) L38: To aid readability, I would avoid abbreviating vegan, vegetarian, and omnivore in the abstract.

10) L40: Remove 'vitamin' as it is redundant when combined with 'micronutrient'.

11) L42-43: Better total and LDL cholesterol in vegans compared to... who? Does this include adults as well?

12) L43-44: The claim that adult omnivores had a higher diastolic blood pressure and lower fat-free mass is not supported by Table 4. Also, compared to vegans or vegetarians, or both?

13) L44-46: Only adults had slightly higher PTH levels. Saying that higher bone turnover was 'related to PTH levels' implies a direct comparison. I would simply describe which variables tended to be higher/lower/similar. There is also a discrepancy in the units/values reported for P1NP between Tables 3 and 4.

14) Rather than simply describing variables are being higher or lower, try and convey the magnitude of the difference to the reader as well. A little or a lot? Consider throughout. Also, be mindful that some differences are described as 'significant' or 'non-significant' while other differences are described as higher/lower even though they were non-significant at the conventional threshold of <0.05 (e.g., adult omnivores, on average, had a higher body mass index than vegans/vegetarians, but the corresponding p value was 0.06. This was done on L43-44). This is a point about consistency. Although, the priority should be to convey the magnitude of the difference and the clinical significance rather than just the statistical significance.

15) L46: Given the high proportion of vegans who supplemented with vitamin D in this sample, why is it 'paradoxical' that vitamin D levels were higher in vegans? Also, compared to vegetarians or omnivores? And in which age groups?

16) L46-48: Given the study design, I would avoid saying that lower urinary iodine and iodine intake had an 'effect' on TSH levels. Also, vegetarians tended to have lower urinary iodine compared the omnivores as well, but only vegans are mentioned. It is important to note that overall iodine intake was low overall in this sample based on 3-day food record data. For example, median iodine intake was between 59 and 73 mcg/day in vegan, vegetarian, and omnivorous adults, but the recommended dietary allowance is usually about 150 mcg/day. Did estimates of iodine intake also consider iodised salt and supplementation? Also, did food composition databases include sufficiently complete information on the iodine content of relevant foods? I'm wondering whether iodine intake has been underestimated.

17) L42-48: The following abbreviations are undefined: LDL, BMI, PTH, TSH. Usually best to write out in full, even if considered standard abbreviations.

18) L45: Omit or spell out 'P1NP'.

19) L48-50: I think it's important to convey the extent/range of clustering here from the mixed-effects model. For example, only 57% of the variability in height among children can be attributed to differences between families/households. Knowing this may change the reader's interpretation. Also, do all these results (height, selenium, zinc, iodine, folate, vitamin B12, vitamin D) relate to adults or children, both, or a combination of the two?

20) L49: For consistency, use micronutrient symbols or write out in full (e.g., Se or selenium). I'd be inclined to avoid symbols where possible.

21) L50-51: Reevaluate use of 'impact' (i.e., strong causal language). Instead, associative language such as 'correlate', 'associate' or 'predict' would be more appropriate. I would also tone down the definiteness of this conclusion with more cautious language such as 'may' or 'potentially'. This paper might be of interest:
<https://academic.oup.com/aje/article/191/12/2084/6655746>.

22) L51-53: As this is mostly a descriptive study, the 'effect' of diet on health-related measures was not studied. Again, associative language would be more appropriate here. Also, what further research is needed in relation to iodine status and bone health? Larger samples? Different assessment methodologies? Longitudinal studies?

Introduction

23) L59-62: Rather than saying 'recent surveys', provide specific years if possible. In reference to the 25% who are 'reducing their intake of animal-based foods', is this in relation to flexitarians or occasional meat consumers? There should probably be a distinction between active reducers versus infrequent consumers of meat or animal-sourced foods in general. What does 'significantly more compared to previous years' mean? Try and convey the magnitude of the difference to the reader.

24) L66-69: Since all diets carry a risk of nutrient deficiencies, especially when self-directed and without guidance from a nutritional professional, it might be better to convey the idea that, on average, certain diets may be at a higher risk of nutrient deficiencies than others. The inclusion of selenium alongside the other listed nutrients of concern is not supported by the provided reference. Compared to omnivores, total energy and protein intake does tend to be lower (though sometimes similar) among vegan/vegetarian adults, but is there also a concern of undernutrition? This concern, however, may be more valid among young children. In general, be mindful of the year of data collection (and how the food environment has likely changed over time) and the issue of dietary measurement error, especially underreporting.

25) L76: Wording suggestion: '...underscoring the importance of... establishing healthy eating habits early in life.'

26) L79: Can the reader be referred to more information about the KOMPAS study? For example, the following pre-print:
<https://www.medrxiv.org/content/10.1101/2024.03.03.24303671v1>

27) It might be helpful to provide a high-level definition of the phrase 'plant-based diet' in the introduction. That is, a diet that prioritises the consumption of plant foods while moderating—without necessarily excluding—animal foods. However, definitions and usage can vary. That said, in certain situations, it may be preferable to retain well-known terms with specific meanings, such as vegetarian or vegan. Cited references often refer exclusively to a vegetarian or vegan diet. Be mindful when combining general statements about plant-based diets with references that are specific to vegetarian or vegan diets.

Methods

28) L593: Change name-date referencing style to numbered.

29) L596: For clarity, write out the month. October 2021 to October 2022?

30) L596-598: Check the exclusion criteria here is the same as Figure 1.

31) L599-600: Were medical histories based on participant self-report or were medical records provided/verified? Also, were medical histories restricted to a particular period (e.g., over the last 3 years)?

32) L600: Clarify that blood sampling was performed following a 12-h fast.

33) It's not explicitly stated how the 3 days of food record data were treated/combined for summary purposes. Averaged?

34) L605. For simplicity, this could be phrased as 'a 3-day weighed food record...'

35) L607: Write out 'FCDB' as only used once.

36) Figure 1: Write out or define 'I/E'. Two abbreviations used for year: 'yrs' and 'yo'. Unclear what is meant by the following: 'non-homogeneous family' and 'incomplete families'. Check the sample sizes reported in the bottom two 3x3 tables. 95 families and 329 individuals were enrolled. 70 individuals when then excluded (eight aged <3 years did not undergo blood sampling and 62 had incomplete food records). However, 321 individuals are then reported for the 'Nutrient intake analysis' and 290 for the 'Laboratory biomarkers analysis'. No need to include 'STROBE' in the title.

Supplemental Methods

37) L144: I was unable to locate the study protocol via the provided link. Consider proving a direct link to the protocol or a digital object identifier (or similar).

Results

38) Why are all results reported as median (interquartile range) by default? The general rule is to present mean (standard deviation) for continuous variables with a reasonably symmetric distribution and median (interquartile range) if highly skewed.

39) Additional information about study participants would be helpful. For example, demographics (e.g., socio-economic status), the number of families with more than one enrolled child, and the age range in each group (adults, infants/toddlers, and pre-schoolers/schoolers).

40) L93: If there were 95 families, would there not be 190 adults (rather than 187)?

41) L94: What is the reason for dividing children into two age groups? This is not clear to the reader. Also, an <= or >= symbol is missing. While information relevant to this is introduced in the second paragraph of the Results (L118--120), it would be helpful to mention this information earlier in the manuscript, including the importance of doing so.

42) L97: I'd remove the following sentence: 'Overall participants in the study were healthy.'

43) For consistency, keep the order of diet groups (VN, VG, and OM) the same when reporting results in parentheses.

44) Results (Description of the Study Groups): Discussing participant medical histories in such detail (e.g., reporting that one child had valve insufficiency) doesn't seem particularly informative given the sample size. Consider editing or removing. This will also help to reduce the length of the manuscript. Also, it is sometimes unclear whether adults or children are being discussed, including whether information is related to a specific table. Consider throughout.

45) L107: Should 'incidence' be 'prevalence'?

46) L135: Which variables were used to assess 'anthropometric and growth characteristics'? Body weight, height, body mass index, and weight to height ratio? For clarity, maybe omit 'growth' or state that growth characteristics were based on anthropometric measures.

47) L149-157: Where do these definitions of nutrient deficiency/depletion come from? Can suitable citations be provided? Consider throughout.

48) L187-189: I don't understand what is meant by '...a trend towards significant differences...'. Under the significance testing framework employed, it is either significant or not.

49) L223: Why are 'quartiles' mentioned here?

50) There is no in-text mention of Figure 3.

51) Figure 2: The red-yellow-blue colour scale is clear, but it is unclear what the black represents.

52) Results (Family clustering and covariates' importance): Given the interest in variables other than diet group/family unit, was age- and sex-standardisation considered for the presentation of summary data?

53) Figure 4 and Supplemental Tables 5 to 7 repeat data. This should be avoided where possible. Either use a figure or table to present data. The * to indicate significance in Figure 4 is also undefined in the caption.

54) L281: To discuss 'diet composition', nutrient intake would need to be adjusted for total energy intake. For example, as nutrient densities or nutrient intake standardised to a suitable reference amount (e.g., 2000 kcal/day for adults). This

manuscript instead presents absolute nutrient intakes. Unless diet composition is of interest, nutrient intakes should be discussed. It is also important to clarify whether estimates of nutrient intake include supplementation.

55) L286: It is incorrect to say 'no difference' when statistically non-significant. If needed, see this article: <https://www.nature.com/articles/d41586-019-00857-9>. A simple solution would be to instead say 'similar'. Consider throughout.

56) Figure 4: Why was the intake of vitamin B12, omega-3 fatty acids, and vitamin D not calculated as well? These are also nutrients of interest.

57) L328-331: Sentence seems unfinished due to '(?)' on line 330.

Discussion

58) L346-347: Saying 'cross-sectional study into dietary intake' is a bit vague. Instead, key elements of the study design could be briefly highlighted before stating the main findings. Possible aspects to highlight include... cross-sectional analysis of baseline data, Czech families where the household followed the same diet (vegan, vegetarian, omnivore), descriptive data on diet and health-related measures for adults and children, etc.

59) L349-350: There may have been no 'significant' differences, but were there any differences that could be of clinical importance? Vegan children aged <3 years, on average, appear to weigh less relative to vegetarians and omnivores, but have a similar body mass index (based on percentiles from reference growth charts).

60) L351-353: Anthropometric measures were similar among adults, but it's important to note that body mass index tended to be lower among vegans and vegetarians compared to omnivores. Average diastolic blood pressure among vegans was about 4 mmHg lower than vegetarians/omnivores... but is this difference meaningful?

61) L354-357: For major findings 3 and 4, are vegans being compared to vegetarians or omnivores?

62) L358-361: See comment #21 regarding the use of the word 'impact' in the context of this study. Also, it is not always clear what is meant by family clustering and covariate importance. Which is of most interest, the Akaike information criterion or intraclass correlation coefficient? A lay explanation and additional context for the reader would be helpful.

63) L362-363: Was there reliable discrimination between all diet group comparisons in adults? For example, in the full model, the AUC for vegetarian vs. omnivore was only 0.62 and the accompanying confidence interval was fairly wide (95% CI: 0.43 to 0.80).

64) L380-382: See comment #55 about the use of 'no difference' when comparing groups. A similar energy intake across diet categories... rather than age categories?

65) L383: See comment #54 about use of the phrase 'diet composition'.

66) L388: Should it be 'multivariate' or 'multivariable'?

67) L388-390: I could not locate the relevant results that support this statement. Was this found in children or adults?

68) 391-393: I struggled to follow/understand this sentence. Which diet groups are being compared? What does 'possibly lower' mean?

69) L395-396: I'm not sure what is meant by 'an important effect of family'. Can this be expanded upon?

70) L406-407: This sentence is unclear and seems incomplete.

71) L415-417: The provided reference (which is specific to blood pressure screening in US children/adolescents) does not support this statement. In general, the reason why screening for cardiovascular disease risk factors in childhood is not routinely done is not because cardiovascular events are rare in this age group.

72) L423-424: Why compare cardiovascular disease risk factors across age categories?

73) L424: Be consistent when referring to age groups. Say pre-schoolers/schoolers or children aged >3 years (or something similar). Consider throughout.

74) L428-429: This sentence is unclear. Simply state which variables, on average, were higher/lower/similar. The word 'relate' is the main issue as it implies a direct comparison rather than a listing of general characteristics.

75) L430-432: This statement is not fully supported by the presented findings. A longitudinal study design (where the same participants are followed over time) would be needed. Also, did participants in this sample follow a 'well-balanced' diet?

76) L444-445: This sentence is missing a reference.

- 77) L452-454: The calcium intake of adult vegans (which was similar to vegetarians) was not significantly different according to Supplemental Table 7.
- 78) L458: Was the 'prevalence' or 'incidence' of bone fractures investigated?
- 79) L460: Given that only four children had a history of bone fracture, I would only discuss adult bone fractures. Saying that there was a lower prevalence of bone fracture in all vegan age groups could mislead the reader.
- 80) L465: How was 'adequacy' established? Should this be vitamin D supplementation... or serum concentrations?
- 81) L467: See comment #78 about prevalence vs. incidence.
- 82) L467-469: I would rephrase this as... the importance of ensuring adequate calcium intake by consuming calcium-rich foods and/or foods fortified with calcium, and if necessary, supplementation.
- 83) L484-491: See comment #6 about the use of a single spot urine to screen for iodine deficiency at the individual level and comment #16 about the overall low intake of iodine in this sample.
- 84) L495-497: Why only emphasise supplementation? Does this include iodised salt and fortified foods? What about natural plant-based sources of iodine?
- 85) L508-510: Compared to who? Serum ferritin levels may have been lower, but were they within or below the normal range? Also, adult vegans and vegetarians had similar levels of serum ferritin.
- 86) L512: Avoid introducing new analyses/data into the Discussion. In this case, a correlation coefficient.
- 87) L515: Not always. For example, adult vegetarians had on average, the lowest MCV (not omnivores).
- 88) L525-534: I found most of this paragraph difficult to understand. I'm also confused as to why it would be important to assess the height of parents in relation to the growth of their child. The heritability of height within families is well recognised.
- 89) L535-537: Which micronutrients in particular? In adults or children?
- 90) L538-540: The meaning of this sentence is unclear.
- 91) L540-542: The nutritional status of who?
- 92) L545-553. This paragraph is mainly about the importance of this research and may be better placed in the Introduction. Instead, I would focus on the strength of the study design here.
- 93) L558-564: Think of causal inference as a sliding scale from low to high certainty as opposed to can and cannot. Also, this is mainly a descriptive study. Thus, the focus should be on describing characteristics/associations rather than attempting to analyse and explain causal effects. See the reference in comment #3 which discusses descriptive epidemiology in relation to confounders.
- 94) L568-569: Was it not possible/feasible to perform a quantitative assessment of nutrient intakes via supplementation?
- 95) L565-569: One of the main concerns with food records is the potential for reactivity (e.g., the tendency to change usual eating patterns to facilitate ease of recording). It is important to emphasise that a 3-day food record may not be representative of long-term dietary intake.
- 96) L571-572: Was information on educational attainment collected? This is the type of information that should be described in the Results or a table. And the information about geographical location should be reported in the Methods. It also hasn't been made clear that participants were from the Czech Republic. Is sampling bias a concern for internal validity or external validity (i.e., generalisability)?

Reviewer #2

(Remarks to the Author)

Dear authors, please read through all comments, end on improving the abstract. I hope you will see my 32 comments for text improvements, I have in a word file marked broad the comments (OBS) and marked red the lines in the merged file I read. It looks like these marks are not visible to you, but expect they are clear anyway, look for 1.OBS until 32.OBS.

The comments below from the reviewer are marked by OBS (numbered 1-32). Line numbers in the merged file are marked red

Abstract

1.OBS The Abstract is a bit hard to read as it does not describe what is compared when something is said to be lower or higher in or for a diet group, i.e. the information is lacking about „...as compared to which other group“, see in comments below.

35 Plant-based diets are growing in popularity because of their perceived environmental
36 and health benefits. However, they may be associated with safety risks, that may
37 cluster within families.

2.OBS: ref for growing in popularity? How? Among health staff? Advisors that give recommendations about food based
dietary guidelines? Among general population?

3.OBS: the safety risk, is it larger or smaller than the general OM diet?

37.....we conducted a cross-sectional study of 95 families (47

38 vegan [VN], 23 vegetarian [VG], and 25 omnivore [OM]), including 187 adults, 65

39 children >3 years, and 77 children

4.OBS: are the families clean as VN, VG or OM?

43..... OM had a

44 higher BMI, diastolic blood pressure, and lower fat-free mass in adults.

5.OBS higher than whom? VN or VG or both?

44.....Higher bone

45 turnover (P1NP) was found in older children and adult VN, where it was related to

46 higher PTH levels.

6.OBS higher than whom? both VG and OM or what?

46 Paradoxically, vitamin D levels were generally higher in VN. Lower

47 urinary iodine, associated with lower intake in VN was found across all age strata,

7.OBS vitamin D generally higher in VN than in whom? VG? Or OM? Or both?

8.OBS Lower urinary iodine in VN, as compared to?

51.....Although no

52 serious adverse effects of the diet were found, iodine status and bone health in

53 vegans warrant further research

9.OBS what about the other groups? any serious adverse effects?

10.OBS in general about the Abstract: Would it be more wise to report the values outside the reference limits for e.g. nutrient
status? And outside limits for recommended intake (too low, too high)?

Introduction

55 The global trend to reduce the environmental burden of food production and tackle the

56 obesity pandemic is being followed by a reduction in the consumption of foods of

57 animal origin. The growing trend towards plant-based diets is increasingly evident in

58 many regions from the Eastern European block including the Czech Republic.

11.OBS. What is meant? It is absolutely necessary to say e.g. recommendations/guidelines and use global and international
references. A trend can be something from a single moviestar or singer or whatever. Please use e.g. recent IPCC documents
and WHO? And country reports that have really focused on this you say is a global trend? Many countries are actually still
hesitating probably partly because of the strong influence of meat and dairy producers, which are the agricultural producers
receiving the majority of subsidies aimed for food production.

60....3% of Czech consumers identify themselves as vegan, 7% as

61 vegetarian, and a remarkable 25% as voluntarily reducing their intake of animal-based

62 foods, significantly more compared to previous years 1,2

12.OBS these values actually tell the whole story, 3 and 7 percentages, are 10% together i.e. a low % of the population. 25%
have finally diminished their intake of animal based foods – something that has been advised since the seventies or eighties
in the last century in many countries.

13.OBS Refs number 3 and 4 tell about positive health effects of plant-based diets, and refs 5 – 12 about negative effects,
this lacks some balance.

Results

Line 100-101

14.OBS two participants had hypertension compensated on the treatment 101 (VG=2, OM=1) ? (seems 3)

Line 103-104

15.OBS „87 subjects distributed 104 evenly across groups (OM=27, VG=23, VN=27)“ ?? (seems 77)

- 16.OBS Generally in this chapter, focus more on: Difference between groups in Clinical findings that might be associated
with the diet e.g. allergy !

In children the nutr.status of iodine, Vit-D, Folate, Vit-B12 are all of interest and surprisingly some VN have higher levels
than OM in B12, vit-D as well as in Folate, but apparently iodine warrants a further investigation. 17.OBS.Explain? probably
in discussion

In adults the status associated with the risk of coronary-heart disease, i.e. blood lipids and blood pressure, seem higher for
OM, however there seems risk of insufficient iron and zinc statuses.

Line 201

18.OBS The lowest 20%, and highest 80%

-Generally in this chapter: Is someone at probable risk?

Line 214

..... vegan diets may be associated with wider spread towards higher iodine levels“

19.OBS for discussion, include if this is an important option/fact for all diets in general, they can be bad and they can be
healthy so the choice of diet ingredients is a key (independent of type of OM, VG, VN). The next step in research and
guidelines should be more openly define the critical points in each diet

Line 247-253

Young children: age, birth weight, breastfeeding and diet influence the measured outcomes;

20.OBS this is relevant and expected – Discussion should involve that the age interval is large for this period in life incl the fastest growth (the first year) and a speedy development

Line 254-260

Children (<3 yrs): most important for outcomes were: Age, birth weight, supplementation and diet

21.OBS Further in this chapter about „Family clustering and covariates' importance“ The family clustering was important as explored for all children together (because of otherwise too few children in each group) in fact the reason can however be diet, and other factors in the environment. In adults sex and diet seemed most important

Line 328-332

„Generally, we were able to reliably discriminate between VN and OM in adults, with out-of-sample AUC 0.82 (95% CI: 0.69 to 0.92), whereas it was only 0.54 in the baseline model (not utilizing clinical characteristics), with a mean (?) AUC gain of 0.28 (0.08 to 0.49). The strongest predictors of VN diet are lower glycemia, total cholesterol, zinc, ferritin, and urea, and higher P1NP and folate.“

22.OBS What does the ? mean here?

Methods (this chapter was far back in the PDF docuemnt – but I read it and put it here as it is logical)

Line 600 „venous blood sampling after 12 hours“ 23.OBS meaning 12 hrs fasting??

Line 605 „weighted dietary record method“ 24.OBS what kind of scales were used??

Line 608-609 „For products not listed in any of the databases, the dietitian recorded nutrient content from the product packaging.“ 25.OBS How common was this for the different age groups? how many product? Can some intake values be too low because of missing values (analysis in the food tabel/database)?

Discussion

Line 360-361 „Family impacted height, micronutrient status (Se, Zn, urinary iodine), and vitamin levels (folate, B12, and D)“

26.OBS why? can it be an unsure methodological association between the diet and the biochemical measures of the nutrients statuses? Speculate: are the food table databases probably too low on some of these nutrients? Or are the biochem analysis of the status probably unsure... OBS there is too little on the status measurements in the Method chapter!

Line 365-396

27.OBS In the chapter about growth and anthropometrics in children – A very good chapter but add a sentence about genetics.

Line 398-....

Also the following chapters are good, on indicators of better cardiometabolic health in vegans which can be identified as early as preschool age; on bone health; and on iodine status.

Line 502-505

Non-heme iron absorption in vegan diets is susceptible to various inhibitors such as phytates, polyphenols, and calcium, which collectively contribute to reduced iron absorption efficiency in vegan diets, despite adequate iron intake from plant 28.OBS controversial to point out calcium here as it might be too low, as said before, for iron the meal composition might be important especially for those totally relying on non-heme iron.

Otherwise this chapter is ok

Line 521

29.OBS The chapter on family – consider former comments

Line 544-572

Chapter on strength and limitations: 30.OBS former comments on the text

Conclusion

Line 575-576

„To conclude, we described the differences among families with distinct eating habits in anthropometric measures, health, and nutritional“ intake and „status indices.“

31.OBS diet or nutrient intake is missing from this sentence, see added „intake and“

Line 581

32.OBS The positive effects of plant-based diets on public health taking the possible weaknesses into consideration, should be stated more clearly in the end, as well as mentioning the negative consequences of OM seen in this and former studies probably because of the habit not taking food based dietary guidelines seriously enough.

Answers accord. to editors Questions.

• What are the major claims of the paper?

ANS.: Major findings show that people / families on vegan diets were in some cases at risk considering bone health and iodine status. Additionally those on omnivorous diet were at higher cardio vascular risk. The latter one has to be mentioned in conclusion (but is not now) – and it is premature to only focus on possible risk of vegan diets. Several biomarkers were importantly associated to diet and/or nutrient intake. The family associations observed are interesting, but still they can be questioned, is it diet or genes or?– possible confounders in methods have to be mentioned more clearly as indicated in the comments to authors (marked OBS).

• Are the claims novel? If not, please identify the major papers that compromise novelty.

ANS.: This is an important study. And the findings are novel as the methods are very detailed – on intake, on growth of children, on biomarkers or nutrient status, on health measures and health history, on various age groups in the same family

• Will the paper be of interest to others in the field?

ANS: Yes definitely

• Will the paper influence thinking in the field?

ANS: Yes. Several improvements are needed on the writing as well as explanations though, these will broaden the text (increasing its importance for understanding research and their implications in the field).

• Are the claims convincing? If not, what further evidence is needed?

ANS.: Yes, but needs to be explained with the consequences of all the three different kinds of diets. The Q is always to

follow the best advice, where are the obstacles and how do/can we solve the problem.

- Are there other experiments that would strengthen the paper further? How much would they improve it, and how difficult are they likely to be?

ANS.: Only larger studies in the future. Not relevant for this study now.

- Are the claims appropriately discussed in the context of previous literature?

ANS.: Mostly. I have added suggestions, some small and some larger in the comments to authors.

- If the manuscript is unacceptable in its present form, does the study seem sufficiently promising that the authors should be encouraged to consider a resubmission in the future?

Yes. They can improve the paper considerably, and the work already done is definitely worth it.

Reviewer #3

(Remarks to the Author)

The manuscript provides an interesting perspective on plant-based dietary patterns within families; however, there are significant limits in the methods section and other sections. Enhancements in methodological rigor, clarity of statistical reporting, and focus on key outcomes are essential to improve its scientific quality. Below are the major points of concern:

Abstract

Specify the location or region where the study was conducted.

Rewrite the introduction to state the study aim clearly.

Briefly mention the statistical tests/models used, e.g., "We used mixed-effect models and ANOVA (?) to analyze the association between dietary patterns and health markers."

Highlight the most significant outcomes, such as differences in cardiometabolic health and iodine levels, while minimizing excessive detail. This will help in developing the method section.

Replace "significant impact" with more precise language and report statistical values if appropriate.

Use a structured abstract format (e.g., Background, Methods, Results, Conclusion) for clarity and coherence.

Introduction

The background of the study is clear but some sentences need to be rephrased for clarity.

Could you rephrase the sentence from line 65 to 69, for better readability? The phrase "they also carry potential risks" creates a smoother transition from benefits to risks.

Method

Please provide the method section before the result. This section is important for understanding the results.

The authors fail to specify what anthropometric measurements (e.g., height, weight, BMI, body composition) and health indicators were assessed and how these were measured. Details regarding the instruments used, their calibration, and the procedural protocols are essential for reproducibility and reliability.

While a 3-day weighted dietary record is mentioned, the authors do not justify the choice of this method or describe how seasonality, supplements, or missing data were accounted for. The lack of explanation undermines confidence in the dietary data's validity.

Although biomarkers like vitamin D and iodine were assessed, the manuscript does not provide a description of the laboratory methods, reference ranges, or quality control measures employed. This omission weakens the reliability of the results.

Participant demographics (e.g., socioeconomic status, ethnicity) and potential confounders like physical activity and education level are inadequately addressed, leaving gaps in the interpretation of results.

Although mixed models and quantile regression are mentioned, the manuscript lacks clarity on how covariates were selected and adjusted for in these analyses. Additionally, the description of predictive modeling lacks sufficient detail to assess its robustness.

Results

The Results section includes an overwhelming number of specific without prioritizing key outcomes. This dilutes the impact of the manuscript and makes it difficult for readers to discern the main message.

"Among the potential confounders influencing the link between diet and health outcomes are sociodemographic status, education level, and physical activity, none of which were accounted for in the study design. Given that this is a cross-sectional study utilizing cohort data, it is surprising and concerning that such critical confounding factors were neither collected nor adjusted for, as they could significantly impact the validity of the findings."

Version 1:

Reviewer comments:

Reviewer #1

(Remarks to the Author)

Reviewer: James P Goode, PhD

Manuscript under review: Dietary intake, nutritional status, and health outcomes among vegan, vegetarian, and omnivorous Czech families: a cross-sectional analysis (COMMSMED-24-0924A)

Revision round: 2

The authors have adequately addressed my main concerns, but reporting inconsistencies, omissions and ambiguities remain. Nonetheless, this version of the manuscript is a marked improvement over the original submission. I appreciate the

author's willingness to engage in scholarly discussion and the review process. As an aside, I applaud their decision to publicly share data and statistical code.

Lines numbers cited in this peer review report refer to the clean version of the submitted manuscript.

--- Graphical abstract ---

1. Update title.
2. Remove 'vitamin' from 'vitamin/micronutrient'. Micronutrient by itself is sufficient.

--- Abstract ---

3. L43-45: This sentence reads as though children were followed into adulthood when in fact these are separate groups. I suggest rewording to avoid confusion.
4. L47: It is unclear what is meant by 'across ages'. At this point, the reader is unaware that children were stratified into two age groups, or are you referring to children and adults?
5. L51: For consistency, say 'vitamin B12' rather than only 'B12'.
6. L54: It could be clearer that low iodine status is the concern (as opposed to excess).
List of abbreviations

7. L56-57: It's odd to define an abbreviation using another abbreviation (e.g., C-HDL = HDL cholesterol). I would write this out in full (i.e., HDL-C = high-density lipoprotein cholesterol).
8. L59: Why abbreviate group (GRP)? If only used once in a figure and not the main text, include the abbreviation in the legend of the figure.

--- Introduction ---

9. L80: Reference #2 is incomplete. Only a title and n.d. (no date) is listed in the reference list.
10. L96-97: This sentence is incomplete. The impact of plant-based diets... on what?
11. L104-106: Nutritional status is stated under point 1 and then the risk of micronutrient deficiencies is stated under point 2. Do these not overlap?

--- Materials and methods ---

12. L115: It is stated that participants gave 'written consent', but did participants give written informed consent? This is an important distinction.
13. L122: Can further information be provided about the study location or setting? A particular city, hospital or region in the Czech Republic?
14. L147-148: It is unclear what is meant by '3-day weighted dietitian-supervised dietary record method'. This was raised in my first peer review report (rebuttal #34). So, parents/carers completed a 3-day weighed food record that was then checked for completeness by a dietitian?
15. L155-157: Since this study sought to investigate the intake of key micronutrients between diet groups/families, the reason for not including the information provided by dietary supplements (despite being collected) should be noted (as explained in rebuttal #94).
16. L161: The reader would also benefit from knowing the reason for separating children into two age groups (i.e., <3 and ≥3 years). This was raised in my first peer review report (rebuttal #41).
17. L163-168: This section of text lacks clarity, especially point 2. Even after re-reading, it is still not entirely clear what is being investigated and how.
18. L186: ROC undefined.
19. L188: AUC undefined.
20. L197: FDR undefined.

--- Results ---

21. L205: I assume the > (more than) symbol should be >= (more than or equal to)? This was raised in my first peer review report (rebuttal #41). Throughout the manuscript, children are referred to as <3 or >3 years, so what about children aged 3 years?
22. L206-207: Here, there is no need to refer to a vegan diet (which has already been defined) as an 'exclusively plant-based diet'. Why switch to an undefined term?
23. Table 2-3 – Footnotes: Review and update the list of abbreviations. For example, systolic and diastolic blood pressure are defined but not mentioned in Table 2-3. Consider throughout the manuscript.
24. Table 5 – In L170-172, robust linear mixed-effect models with random effects (intercepts?) by family were adjusted for age, sex and variables related to breastfeeding, but in the footnote of Table 5, it states that robust linear mixed-effect models were 'adjusted for the effect of potential confounders (age, sex, diet, family, and other covariates).' There is a clear discrepancy here.
25. The inclusion of Supplemental Table 3 is helpful (as per rebuttal #39), but a key suggestion was to also mention the number of families with more than one enrolled child.
26. L215: In the Methods, children are referred to as < 3 or ≥ 3 years, but in the Results, the terms 'infants/toddlers' and 'pre-schoolers/schoolers' are introduced. There needs to be consistency in terminology. This was raised in my first peer review report (rebuttal #73).
27. L217-219: I do not see how the linear mixed-effects model quantifies the 'importance of covariates like age, sex, breastfeeding status, or family influence' and how this relates to Table 5-6.
28. L222-223: This sentence repeats information from the previous paragraph. Also, 'adults' is mentioned but this section relates to children.
29. L224-225: It is important to clarify which diet groups are being compared rather than just reporting results for 'children', particularly as children are sometimes separated into two age categories and sometimes combined. Also, state which results table is being referenced here.
30. L224-225: The authors were asked to clarify what is meant by growth (as per rebuttal #46), but how this has been actioned is unclear.
31. L225-226: What is the significance of weight/height being below the 3rd percentile? How percentiles were calculated should also be mentioned in the main text.
32. L227-228: Inconsistent use of abbreviations. Is it LDL cholesterol or C-LDL? Is it total cholesterol or TC? Consider throughout the manuscript.
33. L227-231: It is unclear which tables and age/diet groups are being reported here. This issue was raised in my first peer review report (rebuttal #44). This applies to the entire Results section as well.
34. L227-231: Is it necessary to report that four children had an LDL cholesterol measurement below the 'lower reference limit' (which has not been previously discussed or stated)?
35. In rebuttal #47, the authors state 'The iodine deficiency criteria used in the manuscript are based on the WHO document Urinary iodine concentrations for determining iodine status in populations. We have now added this reference. We recognize that the WHO median cut-offs for iodine deficiency should be used at the population level. In the revised version of the manuscript, we only comment on the median UIC in the subgroups.' However, no reference has been added and the number of children who had a urinary iodine concentration below WHO cut offs (<100 mcg/L and <20 mcg/L) is still reported in the Results (L237-240). Also, is it an issue that the WHO document is specifically referring to school-aged children (6 years or older) and adults, whereas most of the children in this study were aged <6 years?
36. L233-234: Where does this definition of mild vitamin D depletion come from? When citing cut offs and definitions in relation to nutrient status (which is done regularly), it is important to cite source documentation. Consider this throughout the manuscript.
37. L234-235: It is not clear to the reader why this finding is 'interesting'. The significance of this finding is not even mentioned in the Discussion.
38. L235: Typo. Should 'creatine' be 'creatinine'?
39. L236-237: Compared to other diet or age groups? Simply saying 'group' is a little vague because this study has both diet and age groups.
40. L254: What is the 'upper reference limit' for total and LDL cholesterol? To further emphasise comment #36, it is important to cite what the cut off is and where it comes from (i.e., source documentation).

41. L256: Here, it is incorrect to say that there were 'no differences'. This was raised in my first peer review report (rebuttal #55).
42. L261-262: Serum zinc levels were not significantly different between vegan and vegetarians according to Table 6. Use of the phrase 'clear, significant gradient' implies that more than two group means were compared simultaneously, which is not the case here.
43. L317-319: It is stated that 'only VN and VG children [< 3 years] had significantly lower intakes of saturated fats and cholesterol', but only saturated fat intake was lower (when compared to omnivores) according to Figure 3.
44. L319-321: The wording of this sentence could be much clearer. The phrase 'gradual shift' is the main issue as it lacks precision/clarity.
45. L323: Again, use the terminology defined in this manuscript for vegan and vegetarian diets rather than referring to them as 'plant-based diets.'
46. Figure 3 – The * remains undefined. This was raised in my first peer review report (rebuttal #53). The accompanying footnote also does not define 'GRP' or 'Ch'. Correct spelling of saturated fats (currently 'saturate fats'). To confirm, does the significance test compare the means of two specific diet groups (e.g., vegan vs. omnivore) or all three means (vegan vs. vegetarian vs. omnivore)?
47. L363-368: There are discrepancies between the values reported in this paragraph and Table 7. Double-check.
- Discussion ---
48. L383-384: For consistency, say vegan, vegetarian and omnivore or use abbreviations (VN, VG and OM). Do not switch between them. Either write out in full or abbreviate. Consider throughout the manuscript.
49. L393: This sentence lacks clarity and is incomplete: 'The most prominent covariate for children was age, and for adults, sex.'
50. L397-L400: The wording of the final key findings also lacks clarity.
51. L422-426: How do your results contrast with the Polish study? Vegan children were generally found to be shorter in both cases.
52. L460: The presentation of the age categories is difficult to follow here.
53. L462-463: See rebuttal #14 and #48 in relation to the phrase 'verge of significance'.
54. L475-476: This sentence reads as though it is referring to the present study (which is it not). Reword for clarity.
55. L477: BMD undefined.
56. L488-490: In the vegan group? It is also unclear what is meant by 'largely insignificant'.
57. L493-495: I do not follow the logic of this sentence. Would low calcium bioavailability from the diet increase faecal calcium loss, not urinary calcium loss?
58. L519-520: According to Supplemental Table 6, iodine intake was only significantly lower when compared to omnivores (not vegetarians as well).
59. L521: What is the recommended intake for iodine? And how much lower on average?
60. L537: Consider changing 'namely' to 'particularly'.
61. L558: For consistency, say 'vitamin B12' rather than just 'B12'. Consider throughout.
62. L576-578: The prevalence of allergies among adults was dependent on family? Where is this reported in the Results section?
63. L578-581: It is unclear what is meant by 'these nutrient'. Which nutrients?
64. L591-594: This sentence is incomplete. The effect of vegan diets... on what?
65. L602-603: Here, it is stated that 'random enrolment across seasons' was performed, however, this was not mentioned in the Methods.

--- Supplementary material ---

66. Update title.

Reviewer #2

(Remarks to the Author)

Reviewer #3

(Remarks to the Author)

Dear authors,

Congratulation for the comprehensive and thoughtful revisions made in response to the reviewer feedback. The manuscript is now substantially improved in terms of structure, methodological clarity, and scientific rigor. The expanded Methods section, clearer statistical reporting, and improved interpretative caution, particularly in replacing causal language with associative framing greatly strengthen the paper. The abstract is now more precise, and your discussion is particularly informative.

I believe this manuscript makes a valuable contribution to the literature on plant-based diets in family and pediatric contexts. That said, I encourage you to consider the following minor improvements during the final revision:

Abstract: While comparisons are now clearer, the phrasing of some findings remains dense. Simplifying or streamlining certain sentences (e.g., bone turnover and vitamin D findings) could improve clarity for general readers.

Discussion (Section 4.3 – Bone health): The section is thorough but could be more concise. Consider condensing repeated explanations about P1NP, vitamin D, and calcium to maintain focus without redundancy.

Discussion (Section 4.6 – Family clustering): Clarify whether the statement about allergy prevalence and family background reflects statistical findings or observational trends.

TEnsure consistent use and definition of technical terms (e.g., P1NP, MMA, sTfR). Also, harmonize phrases like “trend toward significance” and “no significant difference” for clarity and accuracy.

Some long sentences, especially in the Introduction and Methods, could benefit from minor editing to improve flow and readability for a broader scientific audience.

Overall, this is an important and well-executed study, and I appreciate the significant effort invested in strengthening it during revision.

Version 2:

Reviewer comments:

Reviewer #1

(Remarks to the Author)

I have no further comments on this submission. Given the length and complexity of the manuscript, the authors have done a commendable job in addressing the reviewers' comments. I wish them all the best in their future research endeavours.

James P. Goode, PhD

Reviewer #3

(Remarks to the Author)

Dear authors, thank you again for thoughtfully addressing all of the suggestions. Your clearer Methods section, consistent terminology and targeted revisions of the Results have materially strengthened the manuscript and improved its readability.

Rebuttal letter

Dear Editor, dear Reviewers,

On behalf of my co-authors and myself, it is my pleasure to submit our response to the reviewers' comments. We sincerely appreciate the time and effort the reviewers have taken to evaluate our manuscript. Their insightful feedback has helped us refine our work and improve the clarity and quality of our presentation.

In the attached letter, we provide detailed responses to each of the reviewers' comments point by point, outlining the revisions we have made to the manuscript accordingly. Thank you for your time and consideration. We sincerely appreciate the opportunity to revise and improve our manuscript in response to the reviewers' valuable input.

Allow me to add a more personal note: I am genuinely grateful for the reviewers' thoughtful and constructive feedback. Their comments have not only strengthened our manuscript but also contributed to a more precise and rigorous presentation of our findings. I sincerely appreciate the time and effort they have dedicated to providing valuable feedback. As someone who also strives to give thorough and constructive feedback to peers, I recognize the importance of this process for the scientific community, and I would like to specifically acknowledge and appreciate their contributions.

Best regards,

Jan Gojda, MD, PhD

On behalf of all the authors

Point-by-point responses to the reviewers

Note:

The references to lines in the rebuttal letter relate to the file: Related manuscript file marked up, where you can also track the changes.

Reviewer #1 (Remarks to the Author):

Reviewer: James P Goode, PhD

Thank you for the opportunity to review this manuscript. I have expertise in nutritional epidemiology, including plant-based diets and health outcomes. While I also have training in biostatistics, I would happily defer to a statistical reviewer with more expertise. The comprehensiveness of my comments speaks to the potential value of the underlying data. I commend the authors on their efforts. I look forward to future publications from the KOMPAS study.

We are very grateful for the detailed reading and analysis of our work. We are aware that the scope of the review is above standard, and it clearly helps us ensure that the final article is more understandable and has the potential for a greater impact.

--- Major comments ---

1/ The length of the manuscript, large number of diet and health-related measures, and range of statistical analyses (with some being possibly underpowered, see comments #4 and #5) hinders readability and comprehensibility. This is compounded by there being nine comparison groups for most analyses (three diet groups by three age groups). This is a challenge to summarise succinctly and in an accessible manner, while also describing/discussing each aspect of the manuscript in sufficient detail.

We fully recognize this point and agree with it. Both during the design phase and again when drafting the manuscript, we carefully considered several approaches for grouping the children and presenting the data. Specifically, we deliberated on how to handle the two age groups, which is somewhat arbitrary since age is a continuous variable. The decision to split the groups was based on two key factors: 1) a large proportion of breastfed children in the < 3 years group and 2) the way clinical paediatricians classify preschool children (toddlers and pre-schoolers). The first factor could impact the nutrient intake data (as breastfeeding intake was calculated based on total daily breastfeeding time), while the second could potentially reduce the appeal of the paper to the clinical community. As we are guided by the deliverables of national grant support, these considerations were not insignificant. Anyhow, in response to the reviewers' other comments, we have significantly restructured how we present our data. For the rLME models, we clarified that these child groups were merged, and we have minimized statements based on unadjusted comparisons. We also acknowledge that the readability of the text may be affected due to the large amount of data provided and have made efforts to improve the overall readability throughout the manuscript.

2/ I also struggled to fully comprehend all aspects of the statistical analyses in relation to the fourth stated aim (specifically, the mixed-effects and elastic net models).

Thank you for this comment. We agree that the description of the statistical analysis may have been too broad and thus difficult to fully comprehend. To address this, we: (i) in the summary tables, we now report only descriptive statistics by diet group and intentionally omit inferential results (P-values). Statistical inference is now presented only in the adjusted analyses based on linear mixed-effects models., (ii) fully omitted quantile regression from the revised analysis and manuscript, as it was no longer essential to the main conclusions, and (iii) provided a more detailed description in both the main manuscript and supplementary materials. Additionally, we included a link to an online statistical report with all relevant R codes.

3/ The main text is over 6000 words. I wonder whether there is more than one paper's worth of data here. While it is certainly possible to present everything, a shorter, more focused paper would likely be easier to read and more impactful.

We appreciate the reviewer's comments and acknowledge the considerations regarding the scope and impact of our manuscript. Before drafting the manuscript, we carefully discussed our publication strategy, and we finally decided to consolidate all cross-sectional data into a single paper. Here, we summarize the arguments supporting this decision.

1/ We believe that presenting the full dataset and results in a single publication enhances its scientific impact, providing a comprehensive overview. This approach ensures a coherent narrative and a contextual understanding of the data more likely than fragmenting the findings across multiple papers.

2/ There are quite a few analytes that were not of major interest and would not merit a sole publication. On the other hand, these results (though in the supplementary results) may be of interest for other research teams to work with. Therefore, while not central to the main findings, we have included them in the supplementary results to allow for future research.

3/ Lastly, our decision was influenced by practical aspects, i.e., the scope/timeframe of the grant funding and the public health relevance of the findings at a national level.

4/ The Methods section reported in the main manuscript is underdeveloped. While a study protocol is referenced, in addition to supplemental methods, key information that should ideally be in the main manuscript is missing. Examples include (1) how vegan, vegetarian and omnivorous diets were defined, (2) the sampling/recruitment strategy employed, (3) how missing data were handled, (4) whether estimates of nutrient intake include supplementation, etc. This is not an exhaustive list. The STROBE-nut checklist may help with this: <https://www.equator-network.org/reporting-guidelines/strobe-nut/>.

We agree on this point with the reviewer and acknowledge the importance of providing key methodological details in the main text to ensure clarity for the reader. Therefore, we have revised and expanded the main Methods section accordingly and added missing information (inclusion/exclusion criteria, sampling strategy, missing data handling, protocol, etc.). Additionally, we have cross-checked the manuscript against the STROBE checklist to ensure full alignment with its recommendations. Specifically to the points raised see (1) how vegan, vegetarian and omnivorous diets were defined (see lines 136-147), (2) the sampling/recruitment strategy employed (see lines 134-136), (3) how missing data were handled (see lines 189-191 and 196-197), (4) whether estimates of nutrient intake include supplementation (see lines 169-169).

We now provide a detailed explanation of our missing data approach for all analyses. Specifically, in the mixed-effects models, we used complete case analysis for the outcome, assuming that missingness was independent of the diet group. Additionally, we imputed missing values for one breastfeeding-related covariate (partial breastfeeding) using a simple regression-based approach based on age due to its strong association with this variable. For the elastic net models, we applied single stochastic imputation using predictive mean matching ('mice' R package) to handle missing predictor values. This ensures that missingness in predictors does not introduce bias in the predictive modeling. The predictive models were used only for the subjects with less than 25% missing values and features with less than 20% missing values. Other missing values were imputed with mean substitution.

We have clarified these details in both the main manuscript and supplementary materials and provided a link to an online statistical report with all relevant R code. See Materials and Methods section.

5) Also, the results of certain measures are presented but the associated methodology is not described (e.g., handgrip strength). In this manuscript, I would also consider placing the Methods section after the Introduction (and relocating any relevant information currently in the Introduction/Results to the Methods). That is, a more traditional IMRaD structure.

We acknowledge the comment, and we are in line with the reviewer that for the manuscript, the IMRaD structure yields more clarity. We have restructured the manuscript as suggested. Thank you for noticing the omission of the methods related to the results. This was an error. Both handgrip strength and bioimpedance analysis were performed during the clinical visit in adults. The methods section (in the manuscript as well as in the Suppl. File) was amended accordingly. See Materials and Methods section, lines 149-152.

3) Tables that present descriptive data include p values under a test hypothesis of no difference (e.g., Tables 2 to 4). This practice is widespread and generally accepted, but STROBE reporting guidelines discourage the use of significance tests in descriptive tables (see item 14a in the full explanation/elaboration document): <https://pubmed.ncbi.nlm.nih.gov/17941715/>. While there may be a 'significant' difference between two groups, one must ultimately consider the magnitude and clinical significance of this difference. If significance testing is retained, an alpha level would need to be stated (e.g., 0.05 or 5%) and, if applicable, how this relates to the methods used for adjusting for multiple comparisons. Tables 2 to 4 also include p values from a covariate-adjusted mixed-effects

model that combined all children, but I wonder whether this is necessary? The p values are very similar to those from the Mann–Whitney U tests. In addition, is there a need to adjust for potential confounders? This article may be of interest: <https://www.nature.com/articles/s41416-020-1019-z>. And is it necessary to include Supplemental Tables 11 to 13? They repeat data from Tables 2 to 4 and don't seem to add much.

Thank you for pointing this out. We have removed p-values from the descriptive summary tables and now focus on adjusted differences in the main text. To avoid over-adjustment and potential collider bias, we limited covariates to those we prespecified based on domain knowledge—factors known to influence clinical outcomes and shown to differ between the diet groups. This approach reduces the risk of data-driven selection while ensuring that meaningful confounders are properly addressed. We also considered the article you referenced and streamlined our presentation by removing redundant Supplemental Tables, as suggested.

Regarding statistical significance, we now clearly state that an alpha level of 0.05 ($P < 0.05$) was considered significant and that we primarily report unadjusted P-values to maximize sensitivity for any potential risk associated with a vegan/vegetarian diet. We consider it more serious to omit a true risk than to allow a few false positives; however, we explicitly note this rationale in both the Supplementary Materials and the main manuscript. Additionally, we have included FDR-corrected P-values in the online statistical report, along with confidence intervals for differences between diet groups (https://filip-tichanek.github.io/kompas_clinical/).

4) Results (Evaluation of nutritional risks in marginal subgroups): Issues of whether to adjust for potential confounders aside, I'm wondering whether the sample size is sufficient to produce meaningful/stable estimates of different points in the distribution (20th, 50th, and 80th percentile) for use in quantile regression modelling. For example, in children aged >3 years, there were only 35 vegans, 13 vegetarians, and 21 omnivores, with an even smaller number for certain variables due to missingness.

*Thank you for this important point. We agree that smaller subgroup sizes, particularly among older children, limit the robustness of quantile regression and confounder adjustment. Consequently, we have removed the quantile regression results from the manuscript. For both inference (mixed-effects models) and prediction (elastic net), we now combine all children into a single analysis, using log2-transformed age (along with other covariates) to account for differences related to age. This approach provides more stable estimates and avoids the limitations posed by small sample sizes in the children subgroup. In detail see the Methods section, **lines 181-202**.*

5) Results (Clinical variables as diet predictors): Like my previous comment, the use of the receiver operating characteristic curve (as a measure of discrimination) may also be underpowered, particularly among children (aged <3 or >3 years). The 95% confidence intervals for the area under the curve vary widely from well under 0.5 (no discrimination) to about 0.8 to 0.9 (with 1 being the theoretical maximum). This lack of precision limits the ability to draw meaningful conclusions. Although there may be a signal when comparing vegans to omnivores (base vs. full model), the issue of precision remains.

Thank you for this observation. To address the issue of limited power and wide confidence intervals, we have combined both children subgroups into a single analysis for prediction, ensuring a larger sample size. This adjustment has resulted in narrower confidence intervals for the area under the curve (AUC), improving the precision of our estimates while still capturing potential signals in diet group discrimination.

6) Can a single spot urine be used to reliably estimate the prevalence of iodine deficiency? There is wide day-to-day variation in iodine intake and urinary iodine only represents recent intake. While this may be useful at the population level, is a single measure of urinary iodine a reliable way of screening individuals for iodine deficiency? See the following article:

<https://pubmed.ncbi.nlm.nih.gov/19888863/>.

We agree with the reviewer on this point: spot urinary iodine concentration (UIC) in considerably small groups is not sufficiently reliable to estimate iodine status. Both day-to-day variations and hydration status may influence the reliability of UIC. We have already encountered this obstacle in our previous research by Svetnicka et al. (DOI: [10.1038/s41430-023-01312-9](https://doi.org/10.1038/s41430-023-01312-9)), where we made our own experience that it is borderline feasible to reliably perform 24-hour urine collection in preschool children. Therefore, we based our protocol on 1/ the WHO recommendation „Urinary iodine concentration for determining iodine status in people“.

https://iris.who.int/bitstream/handle/10665/85972/WHO_NMH_NHD_EPG_13.1_eng.pdf;jsessionid=8DD519777B3A8FE4E93B6E5592119431?sequence=1 Being aware that WHO protocol relates to populations, not individuals, 2/ and the fact that UIC is being widely used in clinical settings to identify persons at risk of iodine deficiency.

Moreover, the participants' hydration status may have influenced the results. To address this, we conducted an additional analysis of stored urine samples to determine urinary creatinine concentration and calculate the urine iodine-to-creatinine ratio. The Methods and Results sections have been updated accordingly. However, this additional analysis did not alter the overall study outcomes.

While the methodological limitations restrict the interpretative strength of our findings, we believe it remains valuable, as it suggests that some participants may have experienced periodically low iodine saturation. Given the current results, we refrained from drawing definitive clinical conclusions and instead considered this observation as a potential safety concern. To ensure clarity, we have refined our statements and revised the limitations section accordingly.

--- Minor comments ---

Title

7) L1-2: The title could be more specific/informative. For example, 'Dietary intake, nutritional status, and health outcomes among vegan, vegetarian, and omnivorous Czech families: a cross-sectional analysis'.

*We agree that stating the geographical location of the study in the title decreases reporting bias, and we are happy to do so. **See the Title.***

Abstract

8) L36: What are the main safety concerns? Nutrient deficiencies? Impaired growth and development during childhood? Giving an example or two may be helpful. For example, '...safety concerns such as nutrient deficiencies, particularly during critical growth periods...'

We have updated the section accordingly, see lines 31-33.

9) L38: To aid readability, I would avoid abbreviating vegan, vegetarian, and omnivore in the abstract.

We agree on this point and have omitted the abbreviations from the abstract. See lines 37-38.

10) L40: Remove 'vitamin' as it is redundant when combined with 'micronutrient'.

We agree on this point and have updated the manuscript accordingly. See line 34.

11) L42-43: Better total and LDL cholesterol in vegans compared to... who? Does this include adults as well?

For clarity reasons, we have rephrased the sentence. See lines 47-49.

12) L43-44: The claim that adult omnivores had a higher diastolic blood pressure and lower fat-free mass is not supported by Table 4. Also, compared to vegans or vegetarians, or both?

Thank you for spotting the error. We have observed the trends toward statistical significance in BMI and fat mass. For the sake of readability, we have omitted this statement from the abstract. See lines 47-48.

13) L44-46: Only adults had slightly higher PTH levels. Saying that higher bone turnover was 'related to PTH levels' implies a direct comparison. I would simply describe which variables tended to be higher/lower/similar. There is also a discrepancy in the units/values reported for P1NP between Tables 3 and 4.

*We agree that we overstated the findings. We have updated the section so that it is clear now that we report exactly what we found, without any implications of causality. On the other hand, we do not see any discrepancy in the units/values for P1NP between tables. **See lines 51-52.** Anyhow, we cross-checked the results in the table.*

14) Rather than simply describing variables are being higher or lower, try and convey the magnitude of the difference to the reader as well. A little or a lot? Consider throughout. Also, be mindful that some differences are described as 'significant' or 'non-significant' while other differences are described as higher/lower even though they were non-significant at the conventional threshold of <0.05 (e.g., adult omnivores, on average, had a higher body mass index than vegans/vegetarians, but the corresponding p value was 0.06. This was done on L43-44). This is a point about consistency. Although, the priority should be to convey the magnitude of the difference and the clinical significance rather than just the statistical significance.

*We acknowledge this reviewer's point. While it is important to distinguish between statistical vs. clinical significance and to give a measure of the magnitude, it is often difficult to provide a readable abstract while providing a balanced amount of outcomes. Anyhow, we have checked the abstract to make the reading consistent in terms of the significance of the results. **See the Results in the Abstract section, lines 46-59.***

15) L46: Given the high proportion of vegans who supplemented with vitamin D in this sample, why is it 'paradoxical' that vitamin D levels were higher in vegans? Also, compared to vegetarians or omnivores? And in which age groups?

*We agree that there is no paradox in the finding. Our statement related to the previously evidenced risks of vitamin D deficiency in vegans. We have omitted this word and rephrased the sentence to clearly state that across all age strata, the vegan groups had the highest vitamin D levels. **See line 52.***

16) L46-48: Given the study design, I would avoid saying that lower urinary iodine and iodine intake had an 'effect' on TSH levels. Also, vegetarians tended to have lower urinary iodine compared the omnivores as well, but only vegans are mentioned. It is important to note that overall iodine intake was low overall in this sample based on 3-day food record data. For example, median iodine intake was between 59 and 73 mcg/day in vegan, vegetarian, and omnivorous adults, but the recommended dietary allowance is usually about 150 mcg/day. Did estimates of iodine intake also consider iodised salt and supplementation? Also, did food composition databases include sufficiently complete information on the iodine content of relevant foods? I'm wondering whether iodine intake has been underestimated.

*A) Thank you for the insightful comment. We agree that 'having an effect' may imply causality, which is not what we found, and we have updated it accordingly in the Abstract, see **lines 54-55.***

B) Concerning the iodine intake. The majority of salts marketed in the Czech Republic are iodine-fortified. If not specified by the participants (declaring using non-iodized salt, such as Himalayan salt), all the salt (be it the added salt itself or the salt in the foods) was considered to be iodine-fortified. As specified in the methods section, supplementation was not included in the dietary estimates data. We primarily utilized the FRIDA food composition database (Denmark, EU), which provides validated iodine values for foods known to contain iodine. Roughly 75% of individual food items were analysed using the FRIDA database. However, for specific Czech food products (e.g., some dairy products, bakery, and traditional bread) and some vegan-specific foods, iodine content was unavailable (we relied on a limited Czech NutriCZ database or information from packaging with limited data). Moreover, 3 3-day diet record may also fail to capture the intakes of periodically ingested foods, which may be the case for continental regions such as the Czech Republic, where seaweeds and sea fish are not consumed regularly.

*We acknowledge, in agreement with the reviewer, that the actual intake levels may be higher than reported, and we stated that clearly in the manuscript (**lines 812-813**). Nevertheless, the observed values remain significantly below the recommended dietary allowance, underscoring the importance of iodine sufficiency monitoring in these populations.*

Of note, the Czech Republic has been considered a region with endemic iodine deficiency that has been successfully compensated for by fortifying salt since 1947. The Czech group for iodine surveillance within the State National Institute, which we collaborate with, is regularly following up on urinary iodine in different populations. In 2004, the experts declared that the iodine deficiency was solved <https://www.liebertpub.com/doi/abs/10.1089/105072504322783849?journalCode=thy>. Unfortunately, the iodine saturation has been declining since then https://hygiena.szu.cz/artkey/hyg-201801-0005_prevenca-jodoveho-deficitu-v-ceske-republice.php. It is, therefore, possible that the mean daily intake in the population is actually lower than the recommended doses.

Nevertheless, there is evidence that iodine intake among vegans may be considerably low DOI: <https://doi.org/10.1017/S000711452300051X> and doi: [10.3390/nu15051163](https://doi.org/10.3390/nu15051163) ; and we want to convey this safety concern to the readers.

17) L42-48: The following abbreviations are undefined: LDL, BMI, PTH, TSH. Usually best to write out in full, even if considered standard abbreviations.

As suggested, we wrote all the abbreviations in full. See lines 47-57.

18) L45: Omit or spell out 'P1NP'.

We omitted P1NP for redundancy. See line 50.

19) L48-50: I think it's important to convey the extent/range of clustering here from the mixed-effects model. For example, only 57% of the variability in height among children can be attributed to differences between families/households. Knowing this may change the reader's interpretation. Also, do all these results (height, selenium, zinc, iodine, folate, vitamin B12, vitamin D) relate to adults or children, both, or a combination of the two?

*We agree that the information in the abstract may be misleading. Given the limited amount of information we can provide in the abstract, we reformulated the section, stated the parameters where we found the family clustering, and specified the age groups. **See lines 56-59.** More specific results are now provided in the Results section.*

20) L49: For consistency, use micronutrient symbols or write out in full (e.g., Se or selenium). I'd be inclined to avoid symbols where possible.

*We agree and we have updated this throughout the text. **See line 58.***

21) L50-51: Reevaluate use of 'impact' (i.e., strong causal language). Instead, associative language such as 'correlate', 'associate' or 'predict' would be more appropriate. I would also tone down the definiteness of this conclusion with more cautious language such as 'may' or 'potentially'. This paper might be of interest: <https://academic.oup.com/aje/article/191/12/2084/6655746>.

*We agree on this point, it is not our intention to infer any causality. We have updated accordingly here and also throughout the text. **See line 60.***

22) L51-53: As this is mostly a descriptive study, the 'effect' of diet on health-related measures was not studied. Again, associative language would be more appropriate here. Also, what further research is needed in relation to iodine status and bone health? Larger samples? Different assessment methodologies? Longitudinal studies?

*We have revised the aforementioned sections to ensure clarity regarding the nature of our evidence. Larger sample sizes, particularly in pediatric populations, and prospective studies are warranted to strengthen the findings. Regarding dietary intake assessment, we acknowledge that no single method currently represents a gold standard in research. However, with advancements in artificial intelligence and smart wearables, future innovations may enable a more robust and precise characterization of dietary intake. **See line 64.***

Introduction

23) L59-62: Rather than saying 'recent surveys', provide specific years if possible. In reference to the 25% who are 'reducing their intake of animal-based foods', is this in relation to flexitarians or occasional meat consumers? There should probably be a distinction between active reducers versus infrequent consumers of meat or animal-sourced foods in general. What does 'significantly more compared to previous years' mean? Try and convey the magnitude of the difference to the reader.

*Thank you for the comment. We have updated the wording accordingly to present it in a clearer way. **See lines 86-89.***

24) L66-69: Since all diets carry a risk of nutrient deficiencies, especially when self-directed and without guidance from a nutritional professional, it might be better to convey the idea that, on average, certain diets may be at a higher risk of nutrient deficiencies than others. The inclusion of selenium alongside the other listed nutrients of concern is not supported by the provided reference. Compared to omnivores, total energy and protein intake does tend to be lower (though sometimes similar) among vegan/vegetarian adults, but is there also a concern of undernutrition? This concern, however, may be more valid among young children. In general, be mindful of the year of data collection (and how the food environment has likely changed over time) and the issue of dietary measurement error, especially underreporting.

*We agree on this point and we have updated the relevant section. We also updated the references for the evidence of selenium as a potential nutrient of concern. While nutrient inadequacy has been well established for the critical nutrients, it does not always translate to any relevant clinical syndromes. And when it comes to the prevalence of undernutrition, on average, the evidence does not support this risk for vegans but the prospective evidence for children is lacking. We have also excluded the 1988 Sanders reference as being outdated. We have now edited the paragraph. **See lines 92-101.***

25) L76: Wording suggestion: ‘...underscoring the importance of... establishing healthy eating habits early in life.’

*We have updated as suggested. **See line 106.***

26) L79: Can the reader be referred to more information about the KOMPAS study? For example, the following pre-print: <https://www.medrxiv.org/content/10.1101/2024.03.03.24303671v1>

*We have updated as suggested. **See line 112.***

27) It might be helpful to provide a high-level definition of the phrase ‘plant-based diet’ in the introduction. That is, a diet that prioritises the consumption of plant foods while moderating—without necessarily excluding—animal foods. However, definitions and usage can vary. That said, in certain situations, it may be preferable to retain well-known terms with specific meanings, such as vegetarian or vegan. Cited references often refer exclusively to a vegetarian or vegan diet. Be mindful when combining general statements about plant-based diets with references that are specific to vegetarian or vegan diets.

We agree with the reviewer at this point. While vegans and vegetarians are relatively well-established groups in the scientific literature, the term “plant-based diet” can have ambiguous interpretations. To ensure clarity and consistency, we have revised the text accordingly throughout the manuscript and use exclusively well-known terms: vegan and vegetarian.

Methods

28) L593: Change name-date referencing style to numbered.

Thank you for spotting the mistake, we have corrected it.

29) L596: For clarity, write out the month. October 2021 to October 2022?

*We have written out the months. **See line 136 and 244-245.***

30) L596-598: Check the exclusion criteria here is the same as Figure 1.

*We acknowledge the point. We adapted the Figure 1 according to the reviewer's suggestion, including a precise definition of inclusion and exclusion criteria. We also corrected the bottom 3x3 tables (comment 36). **See Figure 1.***

31) L599-600: Were medical histories based on participant self-report or were medical records provided/verified? Also, were medical histories restricted to a particular period (e.g., over the last 3 years)?

*The medical histories of the participants were obtained through self-reported data collected during a structured initial clinical visit. No independent medical records were reviewed for verification. Additionally, there were no time constraints regarding the events reported in the medical history (i.e. events from birth until the examination). **See line 147.***

32) L600: Clarify that blood sampling was performed following a 12-h fast.

*Clarified, thank you. **See line 153.***

33) It's not explicitly stated how the 3 days of food record data were treated/combined for summary purposes. Averaged?

*Thank you for the comment, we have specified as suggested, that the records were averaged to produce mean daily intakes. **See line 164.***

34) L605. For simplicity, this could be phrased as 'a 3-day weighed food record...'

*We have updated accordingly. **See lines 159-160.***

35) L607: Write out 'FCDB' as only used once.

We have updated accordingly. See line 164.

36) Figure 1: Write out or define 'I/E'. Two abbreviations used for year: 'yrs' and 'yo'. Unclear what is meant by the following: 'non-homogeneous family' and 'incomplete families'. Check the sample sizes reported in the bottom two 3x3 tables. 95 families and 329 individuals were enrolled. 70 individuals when then excluded (eight aged <3 years did not undergo blood sampling and 62 had incomplete food records). However, 321 individuals are then reported for the 'Nutrient intake analysis' and 290 for the 'Laboratory biomarkers analysis'. No need to include 'STROBE' in the title.

We acknowledge the point. We adapted the Figure 1 according to the reviewer's suggestion, including a precise definition of inclusion and exclusion criteria. We also corrected the bottom 3x3 tables (comment 30). We have also omitted the STROBE headings.

Supplemental Methods

37) L144: I was unable to locate the study protocol via the provided link. Consider providing a direct link to the protocol or a digital object identifier (or similar).

We are sorry for the wrong link, it has now been updated with the doi and a correct reference. See line 112 and 133.

Results

38) Why are all results reported as median (interquartile range) by default? The general rule is to present mean (standard deviation) for continuous variables with a reasonably symmetric distribution and median (interquartile range) if highly skewed.

We consistently use medians and interquartile ranges because many of our clinical characteristics show skewed distributions and/or contain outliers, which can make the mean and standard deviation less informative. Consequently, the median (IQR) provides a more robust measure of central tendency and spread in these cases. This is also consistent with expert recommendations (e.g. Frank Harrell's suggestion: <https://www.fharrell.com/post/rflow/>) and given the often skewed distribution of our variables with variously distant outliers, we chose to report all results as median (IQR). This, we believe, offers more robust reporting in the presence of outliers or non-normality.

39) Additional information about study participants would be helpful. For example, demographics (e.g., socio-economic status), the number of families with more than one enrolled child, and the age range in each group (adults, infants/toddlers, and pre-schoolers/schoolers).

Following the reviewer's suggestion, we have now included demographic information in supplementary tables. See Supp Table 3.

40) L93: If there were 95 families, would there not be 190 adults (rather than 187)?

*We thank the reviewer for this important comment. There was a drop-out of 3 adults AFTER enrolment, i.e. 3 adults from 3 independent families did not attend the clinical visit after enrolment. We have now included this information in **Figure 1 – Study design flowchart**.*

41) L94: What is the reason for dividing children into two age groups? This is not clear to the reader. Also, an <= or >= symbol is missing. While information relevant to this is introduced in the second paragraph of the Results (L118--120), it would be helpful to mention this information earlier in the manuscript, including the importance of doing so.

*As suggested, we have moved the analytical part of the results (second paragraph of the results) into the Methods section and updated it with the rationale for separating children into two age groups: Though age is a continuous variable we have separated children into two age groups for the descriptive analyses (toddlers vs pre-schoolers) as these groups differ in daily allowances, reference values and breastfeeding prevalence. Anyhow age was used as a continuous variable for the rLME models. **See also the response to the comment 1.***

42) L97: I'd remove the following sentence: 'Overall participants in the study were healthy.'

*We agree that this information is redundant in the context and we have removed it. **See lines 247-248.***

43) For consistency, keep the order of diet groups (VN, VG, and OM) the same when reporting results in parentheses.

*Thank you for this comment, we have it corrected accordingly. **See line 243.***

44) Results (Description of the Study Groups): Discussing participant medical histories in such detail (e.g., reporting that one child had valve insufficiency) doesn't seem particularly informative given the sample size. Consider editing or removing. This will also help to reduce the length of the manuscript. Also, it is sometimes unclear whether adults or children are being discussed, including whether information is related to a specific table. Consider throughout.

*We agree with the reviewer on this point and we have moved the whole section into the Supplementary material (**see Supp Table 1**). Anyhow, we believe it is important to show these data, namely for the reference of future prospective results that will arise from the follow-up.*

45) L107: Should 'incidence' be 'prevalence'?

*Yes, we are sorry for the mistake. We have corrected it. **See line 758.***

46) L135: Which variables were used to assess ‘anthropometric and growth characteristics’? Body weight, height, body mass index, and weight to height ratio? For clarity, maybe omit ‘growth’ or state that growth characteristics were based on anthropometric measures.

*We agree on this point, and we have checked the manuscript to correctly state that the growth in children was assessed based on anthropometric characteristics. See **Supp material – Supp Methods – Data Collection – Anthropometrics.***

47) L149-157: Where do these definitions of nutrient deficiency/depletion come from? Can suitable citations be provided? Consider throughout.

*We thank the reviewer for this comment. The iodine deficiency criteria used in the manuscript are based on the WHO document *Urinary iodine concentrations for determining iodine status in populations*. We have now added this reference. We recognize that the WHO median cut-offs for iodine deficiency should be used at the population level. In the revised version of the manuscript, we only comment on the median UIC in the subgroups. As the stratification of iodine deficiency into mild, moderate, and severe is only valid for school-aged children (> 6 years), we have omitted this more detailed stratification. See **lines 297-300.***

48) L187-189: I don’t understand what is meant by ‘...a trend towards significant differences...’. Under the significance testing framework employed, it is either significant or not.

*We agree with the reviewer on this point and we have updated the relevant statements throughout the Results section. See **line 346.***

49) L223: Why are ‘quartiles’ mentioned here?

*As we have decided, based on the reviewers’ comments, to exclude the analysis of marginal subgroups using the quintile regression, the comment is now irrelevant. For the excluded part see **lines 402-416***

50) There is no in-text mention of Figure 3.

*We have updated the relevant results section 3.4 with the missing reference in Figure 3. See **line 531.***

51) Figure 2: The red-yellow-blue color scale is clear, but it is unclear what the black represents.

*We acknowledge the comment, we have specified in the Legend of the Figure that the black color indicates that the given variable was not included in the given model. See **lines 1247-1248.***

52) Results (Family clustering and covariates' importance): Given the interest in variables other than diet group/family unit, were age- and sex-standardisation considered for the presentation of summary data?

Age and sex were accounted for in mixed models, so diet effect estimates estimated with these models are adjusted for both sex and age.

53) Figure 4 and Supplemental Tables 5 to 7 repeat data. This should be avoided where possible. Either use a figure or table to present data. The * to indicate significance in Figure 4 is also undefined in the caption.

*We acknowledge that Figure 4 and the Supplemental tables repeated the data. Anyhow, we believe that the descriptive tabular format allows for visualisation of the distribution of the variables, and as such, it could be of use for other researchers. We now show both the unadjusted descriptive tables and adjusted differences in the main text (see **Tables 2-6**)*

54) L281: To discuss 'diet composition', nutrient intake would need to be adjusted for total energy intake. For example, as nutrient densities or nutrient intake standardised to a suitable reference amount (e.g., 2000 kcal/day for adults). This manuscript instead presents absolute nutrient intakes. Unless diet composition is of interest, nutrient intakes should be discussed. It is also important to clarify whether estimates of nutrient intake include supplementation.

*We acknowledge this comment. Indeed, our focus throughout the manuscript is on absolute nutrient intakes rather than dietary composition. We have made the necessary corrections as suggested and have thoroughly reviewed the text to ensure consistency. **See line 534.***

55) L286: It is incorrect to say 'no difference' when statistically non-significant. If needed, see this article: <https://www.nature.com/articles/d41586-019-00857-9>. A simple solution would be to instead say 'similar'. Consider throughout.

*We agree we have not tested for the null hypothesis. We corrected the sentence here, as well as throughout the text, as suggested. **See lines 540-541.***

56) Figure 4: Why was the intake of vitamin B12, omega-3 fatty acids, and vitamin D not calculated as well? These are also nutrients of interest.

Thank you for the comment. We considered these parameters as being mainly determined by the supplementation, and therefore we did not include them in the dietary intake tables. Moreover, B12 and vitamin D intakes from the diet are null in vegans.

57) L328-331: Sentence seems unfinished due to '(?)' on line 330.

We have checked and corrected the relevant part.

Discussion

58) L346-347: Saying 'cross-sectional study into dietary intake' is a bit vague. Instead, key elements of the study design could be briefly highlighted before stating the main findings. Possible aspects to highlight include... cross-sectional analysis of baseline data, Czech families where the household followed the same diet (vegan, vegetarian, omnivore), descriptive data on diet and health-related measures for adults and children, etc.

*We acknowledge this valuable comment, key elements of the study design were included in the beginning of the discussion. See Discussion section, **lines 631-633**.*

59) L349-350: There may have been no 'significant' differences, but were there any differences that could be of clinical importance? Vegan children aged <3 years, on average, appear to weigh less relative to vegetarians and omnivores, but have a similar body mass index (based on percentiles from reference growth charts).

*We acknowledge that in the age group <3 years vegan children tended to be "smaller" (lower weight and height, same BMI). We have commented on it in the relevant section 4.1 of the Discussion, **see lines 679-680**.*

60) L351-353: Anthropometric measures were similar among adults, but it's important to note that body mass index tended to be lower among vegans and vegetarians compared to omnivores. Average diastolic blood pressure among vegans was about 4 mmHg lower than vegetarians/omnivores... but is this difference meaningful?

*We acknowledge this comment. As we now show our results based on adjusted models, the difference is not statistically significant for blood pressure, and we have removed it from the findings and mentioned only the trend in BMI. **See line 638**.*

61) L354-357: For major findings 3 and 4, are vegans being compared to vegetarians or omnivores?

*We specified the groups compared in the statement as suggested. **See lines 642 and 644**.*

62) L358-361: See comment #21 regarding the use of the word 'impact' in the context of this study. Also, it is not always clear what is meant by family clustering and covariate importance. Which is of most interest, the Akaike information criterion or intraclass correlation coefficient? A lay explanation

and additional context for the reader would be helpful.

Thank you for your comment. We have now provided a more detailed explanation in the statistics section. Covariate importance refers to the contribution of a variable or a group of related variables, such as breastfeeding, to the out-of-sample prediction of the outcome. This is assessed using the Akaike Information Criterion (AIC), where a drop in AIC after removing a variable suggests that the variable plays an important role in predicting the outcome.

Family clustering refers to the fact that individuals from the same family may share similar clinical characteristics, violating the assumption of independent observations. To account for this, we included a random effect for family in our models. Our primary interest was in AIC as a measure of model performance, but when the inclusion of the family random effect led to a substantial reduction in AIC, indicating that family structure was important for prediction, we also reported the intraclass correlation coefficient (ICC) to provide additional insight into the degree of clustering within families. This has now been clearly stated both in the main text (see Statistical analyses section) and further in the supplementary statistics section.

63) L362-363: Was there reliable discrimination between all diet group comparisons in adults? For example, in the full model, the AUC for vegetarian vs. omnivore was only 0.62 and the accompanying confidence interval was fairly wide (95% CI: 0.43 to 0.80).

*We agree that the reliable discrimination is only between adult OM vs VG. We have now specified it in the text. **See lines 655-656.***

64) L380-382: See comment #55 about the use of 'no difference' when comparing groups. A similar energy intake across diet categories... rather than age categories?

*We acknowledge the point, and we have reformulated the statement accordingly. **See lines 674-675.***

65) L383: See comment #54 about use of the phrase 'diet composition'.

*We acknowledge the point and we have reformulated the statement to clearly show what was actually measured. **See line 677.***

66) L388: Should it be 'multivariate' or 'multivariable'?

*We are grateful to the reviewer for spotting the mistake; in the text, there should be 'multivariable'. **See line 684.***

67) L388-390: I could not locate the relevant results that support this statement. Was this found in children or adults?

*The results to support the statement are summarized in Figure 2. We specified accordingly that the statement relates to children only. Now the sentence states: In multivariable analyses, we showed that the primary predictor of height in children was family background rather than dietary group. **See line 685.***

68) 391-393: I struggled to follow/understand this sentence. Which diet groups are being compared? What does 'possibly lower' mean?

*We acknowledge that the sentence is hard to follow and uses a vague "possible lower" term. We wanted to convey the findings that differences in IGF-1 seen in children above 3 years (non-significant) were not observed when adjusted for the family and age. We completely reformulated the sentence, so it states now: "differences in IGF-1 were driven by family and age instead of diet." **See lines 687-688.***

69) L395-396: I'm not sure what is meant by 'an important effect of family'. Can this be expanded upon?

*Again, we agree at this point, the statement was vague, so we reformulated it, and it now states: "but a strong correlation within the family". **See line 693-694.***

70) L406-407: This sentence is unclear and seems incomplete.

*We acknowledge that the sentence reads badly, and we have removed it. **See lines 706-707.***

71) L415-417: The provided reference (which is specific to blood pressure screening in US children/adolescents) does not support this statement. In general, the reason why screening for cardiovascular disease risk factors in childhood is not routinely done is not because cardiovascular events are rare in this age group.

*We acknowledge the concern raised, and we have reformulated the section so that it cannot mislead the reader on the debate on the rationale for screening of high blood pressure in children. **See lines: 718-719.***

72) L423-424: Why compare cardiovascular disease risk factors across age categories?

*We agree that the statement may be misleading. We have not formally compared the age strata head-to-head. The message we wanted to convey was that the gap in CVD risk-associated markers was widening with increasing age. **See lines 725-727.***

73) L424: Be consistent when referring to age groups. Say pre-schoolers/schoolers or children aged >3 years (or something similar). Consider throughout.

We acknowledge this inconsistency, and we have checked here and throughout the text to ensure the unified use of the names. See lines 727-731.

74) L428-429: This sentence is unclear. Simply state which variables, on average, were higher/lower/similar. The word 'relate' is the main issue as it implies a direct comparison rather than a listing of general characteristics.

We acknowledge the issue and we have restructured the sentence, to take into account adjusted models. See lines 727-734.

75) L430-432: This statement is not fully supported by the presented findings. A longitudinal study design (where the same participants are followed over time) would be needed. Also, did participants in this sample follow a 'well-balanced' diet?

We agree that we have not analyzed whether the diet actually was "well-balanced" and that the design was cross-sectional, precluding prospective outcomes. We have mitigated and reformulated the statement accordingly. See lines 735-738.

76) L444-445: This sentence is missing a reference.

The missing reference to the study by Tong et al., 2020, was added. Line 752.

77) L452-454: The calcium intake of adult vegans (which was similar to vegetarians) was not significantly different according to Supplemental Table 7.

We agree that we have found no significant difference in calcium intake among dietary groups and across ages. We related to the descriptive finding in adults, where the difference was at the level of $p=0.055$. Following the statistical comments above, we finally consider the intakes comparable and we have restructured the paragraph. While the intakes may be comparable, there may be a difference in resorption (lower availability of calcium from plant sources), so we decided to further measure urinary calcium/phosphate concentrations and their ratio to urinary creatinine. Nevertheless, we found no significant differences that would support this hypothesis.

The whole paragraph 4.3 of the Discussion has been restructured accordingly. See lines 740-793.

78) L458: Was the 'prevalence' or 'incidence' of bone fractures investigated?

Prevalence indeed, we corrected the mistake. See line 758.

79) L460: Given that only four children had a history of bone fracture, I would only discuss adult bone

fractures. Saying that there was a lower prevalence of bone fracture in all vegan age groups could mislead the reader.

*We agree on this point, and we have specified it in the text. **Line 760.***

80) L465: How was 'adequacy' established? Should this be vitamin D supplementation... or serum concentrations?

*We agree that the term adequacy is misleading in the context of the results and we have restructured the whole paragraph. **See lines 781-783.***

81) L467: See comment #78 about prevalence vs. incidence.

*We agree and we have corrected the mistake – rephrased the whole paragraph. **See line 781-783.***

82) L467-469: I would rephrase this as... the importance of ensuring adequate calcium intake by consuming calcium-rich foods and/or foods fortified with calcium, and if necessary, supplementation.

*We acknowledge that and we have rephrased accordingly. **See lines 781-783.***

83) L484-491: See comment #6 about the use of single spot urine to screen for iodine deficiency at the individual level and comment #16 about the overall low intake of iodine in this sample.

*We acknowledge this important point. The rationale is provided under the respective points above (response to comment 6). We have also incorporated the changes accordingly in the discussion section. Moreover, we have updated the Methods section in the Suppl. to specify that iodized salt and fortified foods were taken into consideration in measuring dietary intake. **Please see lines 812-813.***

84) L495-497: Why only emphasise supplementation? Does this include iodised salt and fortified foods? What about natural plant-based sources of iodine?

*We agree that supplementation should be the last option to consider once the dietary needs are not met by the regular diet intake and special emphasis should be placed on the use of iodized salt. We have updated the section accordingly, **see lines: 827-828.***

85) L508-510: Compared to who? Serum ferritin levels may have been lower, but were they within or below the normal range? Also, adult vegans and vegetarians had similar levels of serum ferritin.

We agree that the statement was misleading. We have specified that the differences were found between VN, VG vs OM adults, and they all were on average within the reference ranges. See lines 839-843.

86) L512: Avoid introducing new analyses/data into the Discussion. In this case, a correlation coefficient.

The section has been moved to the results section as suggested. See line 845.

87) L515: Not always. For example, adult vegetarians had on average, the lowest MCV (not omnivores).

We agree with the point and we have restructured the paragraph. We also added our assumption of the clinical relevance of the findings: MCV and Hb levels should not guide clinical decisions on iron or B12 deficiencies in the groups. See lines: 847-854.

88) L525-534: I found most of this paragraph difficult to understand. I'm also confused as to why it would be important to assess the height of parents in relation to the growth of their child. The heritability of height within families is well recognised.

We acknowledge this point. We tried to rephrase the paragraph to improve the readability as suggested. The point about the parental height and height of the children was specified to underscore the importance of this confounder for cross-sectional research in vegan vs. omnivore groups. See the 4.6. section of the Discussion.

89) L535-537: Which micronutrients in particular? In adults or children?

We have specified this in the respective sentences. See lines: 869-872.

90) L538-540: The meaning of this sentence is unclear.

We rephrased the sentence to improve readability. See lines 874-877.

91) L540-542: The nutritional status of who?

Thank you for the point, it was a bit unclear. We have specified that it is related to individuals. See line 878.

92) L545-553. This paragraph is mainly about the importance of this research and may be better placed in the Introduction. Instead, I would focus on the strength of the study design here.

We acknowledge that this part was better suited for the introduction and we have now restructured it as suggested. See lines: 886-892.

93) L558-564: Think of causal inference as a sliding scale from low to high certainty as opposed to can and cannot. Also, this is mainly a descriptive study. Thus, the focus should be on describing characteristics/associations rather than attempting to analyse and explain causal effects. See the reference in comment #3 which discusses descriptive epidemiology in relation to confounders.

We acknowledge the comment. We have rephrased the parts of the paragraph to improve the clarity. See lines: 898-911.

94) L568-569: Was it not possible/feasible to perform a quantitative assessment of nutrient intakes via supplementation?

Thank you for the question. Honestly, we have tried hard to quantitatively analyse the supplement use and unfortunately, we failed to produce any meaningful results. The use of supplements is very variable in terms of temporality and the compounds. While the participants remember that they use a compound (i.e. vitamin for instance) for some time during the year, they quite often cannot recall exactly the manufacturer, and they cannot quantify how often they use it. This irregular use of supplements was not captured by the diet records. We identified this obstacle early in the course of the study, and we had to make a decision to either let the participants confabulate this information or just stay on the firm ground of qualitative assessment. We chose the latter. We have the data on individual supplement use with doses and compounds, but unfortunately, these are largely limited by the recall bias. See lines 912-917.

95) L565-569: One of the main concerns with food records is the potential for reactivity (e.g., the tendency to change usual eating patterns to facilitate ease of recording). It is important to emphasise that a 3-day food record may not be representative of long-term dietary intake.

We cannot agree more with the reviewer on this point. Indeed, there is no optimal method to accurately assess dietary intake and all the methods currently used are prone to errors. The rationale behind the choice of the tool is mainly the fact that in vegans and vegetarians, there are no validated food frequency questionnaires that would take into account vegan-specific foods. But we are aware that underreporting is an important issue here, and we have now made it clear in the limitations section. Moreover, we tried to put more emphasis on the biomarkers defining the groups than on the dietary intake. See line 914.

96) L571-572: Was information on educational attainment collected? This is the type of information that should be described in the Results or a table. And the information about geographical location should be reported in the Methods. It also hasn't been made clear that participants were from the

Czech Republic. Is sampling bias a concern for internal validity or external validity (i.e., generalisability)?

*Thank you for the valuable comment. Indeed, we have collected the information on the education level and we have amended the analysis. **It is now placed in Suppl. Table 3.** As evident from the numbers, the majority of people across groups were from urban areas and were well-educated.*

Sampling bias is a concern of external validity as our selection was based on convenience sampling, i.e. willingness to participate, which may be higher among groups with higher health awareness. We have specified this in the respective section.

Reviewer #2 (Remarks to the Author):

Dear authors, please read through all comments, end on improving the abstract. I hope you will see my 32 comments for text improvements, I have in a word file marked broad the comments (OBS) and marked red the lines in the merged file I read. It looks like these marks are not visible to you, but expect they are clear anyway, look for 1.OBS until 32.OBS.

The comments below from the reviewer are marked by OBS (numbered 1-32). Line numbers in the merged file are marked red

Abstract

1/ OBS The Abstract is a bit hard to read as it does not describe what is compared when something is said to be lower or higher in or for a diet group, i.e. the information is lacking about „...as compared to which other group“, see in comments below.

*We acknowledge the point raised. We have reformulated the abstract and specified the groups compared. **See the Abstract.***

2/ OBS: ref for growing in popularity? How? Among health staff? Advisors that give recommendations about food based dietary guidelines? Among general population?

*We acknowledge the point and specified in the abstract that the growing popularity is among the general population. References are given in the **Introduction, lines 81-91.***

3/ OBS: the safety risk, is it larger or smaller than the general OM diet?

*We have omitted the ‘safety risk’ as these all are related to potential nutrient deficiencies. **See lines 93-97.***

37.....we conducted a cross-sectional study of 95 families (47
38 vegan [VN], 23 vegetarian [VG], and 25 omnivore [OM]), including 187 adults, 65
39 children >3 years, and 77 children
4/ OBS: are the families clean as VN, VG or OM?

Thank you for the comment. As specified in the Methods section, the enrolment was based on self-reported assignment that was further confirmed by diet records. Using this approach, our groups are clear as VN, VG, and OM. See lines 138-139.

43..... OM had a
44 higher BMI, diastolic blood pressure, and lower fat-free mass in adults.
5/OBS higher than whom? VN or VG or both?

We acknowledge these misleading statements, and we have now clarified them in the Abstract. See lines 47-50.

44.....Higher bone
45 turnover (P1NP) was found in older children and adult VN, where it was related to
46 higher PTH levels.
6/ OBS higher than whom? both VG and OM or what?

Based on the revised results, focusing on adjusted models (rLME) results, we reformulated respective parts of the Abstract, and again, we have specified the groups compared. See lines 51-54.

46 Paradoxically, vitamin D levels were generally higher in VN. Lower
47 urinary iodine, associated with lower intake in VN was found across all age strata,
7/OBS vitamin D generally higher in VN than in whom? VG? Or OM? Or both?

We acknowledge these misleading statements, and we have now clarified it in the Abstract.

8/ OBS Lower urinary iodine in VN, as compared to?

We acknowledge these misleading statements, and we have now clarified it in the Abstract. See lines 51-59.

51.....Although no
52 serious adverse effects of the diet were found, iodine status and bone health in
53 vegans warrant further research
9/ OBS what about the other groups? any serious adverse effects?

*We thank the reviewer for the comment. No serious adverse effect that could be attributable to any diet was found in our sample. We have restructured the Abstract. **See lines 61-62.***

10/ OBS in general about the Abstract: Would it be more wise to report the values outside the reference limits for e.g. nutrient status? And outside limits for recommended intake (too low, too high)?

We recognize this valid point. Indeed, central tendencies do not provide a complete picture of the potential risk for deficiencies, as a significant portion of the sample may still fall below the reference levels. While we discuss this in detail in the Results section, the limited space in the Abstract prevents us from specifying this information there.

Introduction

55 The global trend to reduce the environmental burden of food production and tackle the
56 obesity pandemic is being followed by a reduction in the consumption of foods of
57 animal origin. The growing trend towards plant-based diets is increasingly evident in
58 many regions from the Eastern European block including the Czech Republic.

11/ OBS. What is meant? It is absolutely necessary to say e.g. recommendations/guidelines and use global and international references. A trend can be something from a single moviestar or singer or whatever. Please use e.g. recent IPCC documents and WHO? And country reports that have really focused on this you say is a global trend? Many countries are actually still hesitating probably partly because of the strong influence of meat and dairy producers, which are the agricultural producers receiving the majority of subsidies aimed for food production.

We acknowledge the point and agree with the reviewer that it may be misleading to talk about “trends” without any relevant references; we have reformulated the sentence and added a relevant reference:

*The global trends to reduce the environmental burden of food production and tackle the pandemic of non-communicable diseases are being followed by tendencies both in consumers and policy-makers aimed at reducing the consumption of foods of animal origin [Fanzo et al., 2021]. **See lines 81-84.***

60....3% of Czech consumers identify themselves as vegan, 7% as
61 vegetarian, and a remarkable 25% as voluntarily reducing their intake of animal-based
62 foods, significantly more compared to previous years 1,2
12/ OBS these values actually tell the whole story, 3 and 7 percentages, are 10% together i.e. a low % of the population. 25% have finally diminished their intake of animal based foods – something that has been advised since the seventies or eighties in the last century in many countries.

We agree on this point. The Czech Republic lags behind the consumer trends in Western countries.

13/ OBS Refs number 3 and 4 tell about positive health effects of plant-based diets, and refs 5 – 12 about negative effects, this lacks some balance.

We agree on that point and we have balanced the references and included a recent umbrella review on meta-analysis of effects of vegan diets (Selinger et al., 2023) and a systematic review and meta-analyses on health aspects of vegan diets among children and adolescents (Koller et al., 2024). See lines 92-97.

Results

Line 100-101

14/ OBS two participants had hypertension compensated on the treatment 101 (VG=2, OM=1) ? (seems 3)

*We have corrected the mistake, we are grateful for your spotting it. Based on the comments of other reviewers, we have now moved the whole section on reported medical histories into a concise **Suppl. Table 1**.*

Line 103-104

15/ OBS „87 subjects distributed 104 evenly across groups (OM=27, VG=23, VN=27)“ ?? (seems 77)

*We acknowledge the point. As above, we have now moved the whole section on reported medical histories into a concise **Suppl. Table 1**.*

- 16/ OBS Generally in this chapter, focus more on: Difference between groups in Clinical findings that might be associated with the diet e.g. allergy !

In children the nutr.status of iodine, Vit-D, Folate, Vit-B12 are all of interest and surprisingly some VN have higher levels than OM in B12, vit-D as well as in Folate, but apparently iodine warrants a further investigation.

*We are grateful for the comment, and we have made the changes in the **Results section** accordingly.*

17/ OBS.Explain? probably in discussion

In adults the status associated with the risk of coronary-heart disease, i.e. blood lipids and blood pressure, seem higher for OM, however there seems risk of insufficient iron and zinc statuses.

Line 201

As we have decided, based on the reviewers' suggestions, to base our conclusions on adjusted models (rLME), we now have lower numbers of significant differences between groups. Concerning

*the differences in zinc and iron levels in adults, these are non-significant when adjusted for confounders. **Please see revised Table 4.***

18/ OBS The lowest 20%, and highest 80%
-Generally in this chapter: Is someone at probable risk?

We acknowledge the point. As we have decided, based on other reviewers' comments, to exclude the quintile regression from the results, this comment is now irrelevant.

Line 214

„... vegan diets may be associated with wider spread towards higher iodine levels“
19/OBS for discussion, include if this is an important option/fact for all diets in general, they can be bad and they can be healthy so the choice of diet ingredients is a key (independent of type of OM, VG, VN). The next step in research and guidelines should be more openly define the critical points in each diet

We agree with the reviewer on this point. As we have decided, based on other reviewers' comments, to exclude the quintile regression from the results, this comment is now irrelevant. Anyhow, we have substantially improved the discussion on the iodine intakes and saturation.

Line 247-253

Young children: age, birth weight, breastfeeding and diet influence the measured outcomes;
20.OBS this is relevant and expected – Discussion should involve that the age interval is large for this period in life incl the fastest growth (the first year) and a speedy development

We acknowledge this important point. We decided to state this clearly in the Limitations section as follows:

*“To maintain sufficient statistical power in the mixed-effects analysis, the pediatric group was analyzed as a single group, which may have limited our ability to detect differences across distinct developmental stages, such as between toddlers and preschool-aged children.” **Lines 918-921.***

Line 254-260

Children (<3 yrs): most important for outcomes were: Age, birth weight, supplementation and diet
21/ OBS Further in this chapter about „Family clustering and covariates' importance“ The family clustering was important as explored for all children together (because of otherwise too few children in each group) in fact the reason can however be diet, and other factors in the environment. In adults sex and diet seemed most important

We agree on this point. Based on the other reviewer's comments, we have decided not to draw any conclusion from the unadjusted descriptive results, and we now show all the major outcomes as based on the adjusted rLME models, where age was a continuous variable.

Line 328-332

„Generally, we were able to reliably discriminate between VN and OM in adults, with out-of-sample AUC 0.82 (95% CI: 0.69 to 0.92), whereas it was only 0.54 in the baseline model (not utilizing clinical characteristics), with a mean (?) AUC gain of 0.28 (0.08 to 0.49). The strongest predictors of VN diet are lower glycemia, total cholesterol, zinc, ferritin, and urea, and higher P1NP and folate.“

22/ OBS What does the ? mean here?

This was a typo, we deeply regret that we have not spotted it in the original version of the draft. We are thankful the reviewer pointed it out.

Methods (this chapter was far back in the PDF docuemnt – but I read it and put it here as it is logical)

Line 600 „venous blood sampling after 12 hours“

23.OBS meaning 12 hrs fasting??

*We acknowledge this point, and we have specified it in the text. **See line 153.***

Line 605 „weighted dietary record method“ 24.OBS what kind of scales were used??

*We used the kitchen measurement scales, which were not independently validated against any precision instrument. Anyhow, we provided all the participants with a unified brand of scales to ensure at least some consistency, if not 100% precision. It has been specified in the main Methods section. **See lines 160-161.***

Line 608-609 „For products not listed in any of the databases, the dietitian recorded nutrient content from the product packaging.“ 25.OBS How common was this for the different age groups? how many product? Can some intake values be too low because of missing values (analysis in the food tabel/database)?

*We are fully aware that underreporting is a significant concern in dietary records. This issue arises not only due to missing data from food composition databases but also because the method itself tends to be reactive, with participants often reporting lower intakes than the average long-term intakes. However, there is no optimal alternative, as Food Frequency Questionnaires (FFQs) and recalls are even less precise and lack validation in specific subpopulations. Regarding the proportion of manually imputed foods (e.g., composite or regionally specific foods), analyzing these items is a complex task, and we consider it beyond the scope of the current manuscript – we plan to analyze the food records in depth in the future. Nevertheless, in the data exported from the nutritional software, approximately 9% of the foods were manually imputed across groups. It is important to note that these items contain limited information on individual nutrients. We have highlighted the limitations of these methods in the Limitations section. **Lines 912-917.***

Discussion

Line 360-361 „Family impacted height, micronutrient status (Se, Zn, urinary iodine), and vitamin levels (folate, B12, and D)“

26.OBS why? can it be an unsure methodological association between the diet and the biochemical measures of the nutrients statuses? Speculate: are the food table databases probably too low on

some of these nutrients? Or are the biochem analysis of the status probably unsure... OBS there is too little on the status measurements in the Method chapter!

*We acknowledge the comment. The concentration biomarkers of nutritional status generally tend to be biased by resorption/distribution/clearance and provide only limited information. We tried to be more specific on that point (potential risk vs deficiency). On the other hand, the message we wanted to convey is that family (and shared eating) is an important confounder when comparing dietary groups. And that nutrient status may depend on the family. We have updated the Methods section (**lines 153-155**) and provided more details in **Supplementary Table 8**, where individual methods for laboratory analyses can be found.*

Line 365-396

27.OBS In the chapter about growth and anthropometrics in children – A very good chapter but add a sentence about genetics.

As suggested by the reviewer, we have updated the manuscript with a sentence:

*“Findings that corroborate previous evidence that genetic factors have a more significant influence on growth than environmental influences (Jelenkovic et al., Scientific Reports 2020)” **Lines 694-696.***

Line 398-....

Also the following chapters are good, on indicators of better cardiometabolic health in vegans which can be identified as early as preschool age; on bone health; and on iodine status.

We are grateful for the appreciation of the chapter, though we have slightly modified it according to our reviewers’ comments. Please see the revised version of the Discussion.

Line 502-505

Non-heme iron absorption in vegan diets is susceptible to various inhibitors such as phytates, polyphenols, and calcium, which collectively contribute to reduced iron absorption efficiency in vegan diets, despite adequate iron intake from plant

28.OBS controversial to point out calcium here as it might be too low, as said before, for iron the meal composition might be important especially for those totally relying on non-heme iron.

Otherwise this chapter is ok

*We acknowledge the comments and as suggested we omitted the calcium from the list, as in our specific sample, it may not be relevant. See the chapter 4.5 of the Discussion. **See lines 832-855.***

Line 521

29.OBS The chapter on family – consider former comments

We have considered and acknowledged the above-mentioned comments on the family.

Line 544-572

Chapter on strength and limitations: 30.OBS former comments on the text

We acknowledge the above-mentioned comments, and we have updated the respective section so to be clearer and more transparent on the limitations. See the 4.7 Strengths and limitations chapter of the Discussion.

Conclusion

Line 575-576

„To conclude, we described the differences among families with distinct eating habits in anthropometric measures, health, and nutritional“ intake and „status indices.“

31.OBS diet or nutrient intake is missing from this sentence, see added „intake and“

We acknowledge the point and we have restructured the Conclusion. See lines 927-934.

Line 581

32/ OBS The positive effects of plant-based diets on public health taking the possible weaknesses into consideration, should be stated more clearly in the end, as well as mentioning the negative consequences of OM seen in this and former studies probably because of the habit not taking food based dietary guidelines seriously enough.

We acknowledge the point raised by the reviewer. Our results provide only a description of the groups, therefore, the implication of positive effects is limited. As this was the baseline analysis of a cohort study, we plan to provide the research community with more results on health events of interest in the future. Anyhow, we have slightly reformulated the Conclusion section as suggested.

Answers accord. to editors Questions.

• What are the major claims of the paper?

ANS.: Major findings show that people / families on vegan diets were in some cases at risk considering bone health and iodine status. Additionally those on omnivorous diet were at higher cardio vascular risk. The latter one has to be mentioned in conclusion (but is not now) – and it is premature to only focus on possible risk of vegan diets. Several biomarkers were importantly associated to diet and/or nutrient intake. The family associations observed are interesting, but still they can be questioned, is it diet or genes or?– possible confounders in methods have to be mentioned more clearly as indicated in the comments to authors (marked OBS).

We are grateful to the reviewer for the comment. As suggested, we have updated the Conclusion section to include the statement about cardiovascular risks. We have also amended the Discussion to include the reference to genetic background and stressed potential confounders in the Limitations and throughout the text.

• Are the claims novel? If not, please identify the major papers that compromise novelty.

ANS.: This is an important study. And the findings are novel as the methods are very detailed – on intake, on growth of children, on biomarkers or nutrient status, on health measures and health history, on various age groups in the same family

• Will the paper be of interest to others in the field?

ANS: Yes definitely

• Will the paper influence thinking in the field?

ANS: Yes. Several improvements are needed on the writing as well as explanations though, these will

broaden the text (increasing its importance for understanding research and their implications in the field).

We have largely restructured the Results and the way we present them. We hope that in its current form, it will be more readable and informative for the readers. Please see also the Executive summary of the Rebuttal letter above.

- Are the claims convincing? If not, what further evidence is needed?

ANS.: Yes, but needs to be explained with the consequences of all the three different kinds of diets. The Q is always to follow the best advice, where are the obstacles and how do/can we solve the problem.

- Are there other experiments that would strengthen the paper further? How much would they improve it, and how difficult are they likely to be?

ANS.: Only larger studies in the future. Not relevant for this study now.

- Are the claims appropriately discussed in the context of previous literature?

ANS.: Mostly. I have added suggestions, some small and some larger in the comments to authors.

- If the manuscript is unacceptable in its present form, does the study seem sufficiently promising that the authors should be encouraged to consider a resubmission in the future?

Yes. They can improve the paper considerably, and the work already done is definitely worth it.

We are grateful for all these comments. We hope that we managed to deal with all the points raised, and we believe that in the present form, the Manuscript is more appealing to the readers.

Reviewer #3 (Remarks to the Author):

The manuscript provides an interesting perspective on plant-based dietary patterns within families; however, there are significant limits in the methods section and other sections. Enhancements in methodological rigor, clarity of statistical reporting, and focus on key outcomes are essential to improve its scientific quality. Below are the major points of concern:

Abstract

1/ Specify the location or region where the study was conducted.

*We acknowledge this point. The location was **specified in the Title of the manuscript** as well as in the respective Methods.*

2/ Rewrite the introduction to state the study aim clearly.

*We acknowledge that the introduction provided the Aims of the study insufficient and we have updated it accordingly. **See lines 114-120.***

3/ Briefly mention the statistical tests/models used, e.g., "We used mixed-effect models and ANOVA (?) to analyze the association between dietary patterns and health markers."

We acknowledge that the statistical approach was not sufficiently addressed. This aligns with feedback from other reviewers, and we have expanded and restructured the relevant sections of the Methods section accordingly. See lines 172-211.

4/ Highlight the most significant outcomes, such as differences in cardiometabolic health and iodine levels, while minimizing excessive detail. This will help in developing the method section. Replace "significant impact" with more precise language and report statistical values if appropriate.

We acknowledge that major outcomes are less clear, given the amount of results provided. We have restructured the Discussion and Conclusion to convey the main outcomes clearly. We have also replaced the words that imply any causal inference (such as mentioned 'impact') throughout the text. See lines 631-653.

5/ Use a structured abstract format (e.g., Background, Methods, Results, Conclusion) for clarity and coherence.

We have restructured the Abstract as suggested.

Introduction

6/ The background of the study is clear but some sentences need to be rephrased for clarity. Could you rephrase the sentence from lines 65 to 69, for better readability? The phrase "they also carry potential risks" creates a smoother transition from benefits to risks.

We acknowledge this point and we have rephrased the respective part of the Introduction. See lines 92-101.

Method

7/ Please provide the method section before the result. This section is important for understanding the results.

As suggested by the reviewer, we have moved the Methods section to align with IMRaD structure.

8/ The authors fail to specify what anthropometric measurements (e.g., height, weight, BMI, body composition) and health indicators were assessed and how these were measured. Details regarding the instruments used, their calibration, and the procedural protocols are essential for reproducibility and reliability.

We acknowledge this point and we have now amended the Methods section to provide all the details necessary. See lines 149-155. To avoid excessive length of the manuscript, we also placed more detailed information in Suppl. Methods.

9/ While a 3-day weighted dietary record is mentioned, the authors do not justify the choice of this method or describe how seasonality, supplements, or missing data were accounted for. The lack of explanation undermines confidence in the dietary data's validity.

Again, we acknowledge this point. The justification for the diet record has been added in the Methods and Limitations. A weighted dietary record is considered a golden standard in assessing dietary intake, though it has its limits. The method itself tends to be reactive, with participants often reporting lower intakes than the average long-term intakes; the reliability of the record depends on the food composition databases used and the way to deal with missing items (i.e. novel foods, UPFs etc.). However, there is no optimal alternative, as Food Frequency Questionnaires (FFQs) and recall methods have their own limits and are less valid in specific populations such as vegans. We have analyzed the number of manually imputed items (i.e., from food packaging), and it is roughly 9% in the whole sample, so we believe it has not contributed to a larger extent to underreporting. Anyhow, we have made it clear in the Limitations. See lines 912-917.

The concerns about how missing data were handled are now specified in the statistical part of the Methods – Statistical analyses.

10/ Although biomarkers like vitamin D and iodine were assessed, the manuscript does not provide a description of the laboratory methods, reference ranges, or quality control measures employed. This omission weakens the reliability of the results.

*We acknowledge this point, and we have amended the Methods section and added **Supp Table 8** with all the laboratory measures.*

11/ Participant demographics (e.g., socioeconomic status, ethnicity) and potential confounders like physical activity and education level are inadequately addressed, leaving gaps in the interpretation of results.

*We acknowledge these confounders, and indeed, we have collected them. We have amended the Results and Discussion sections resp. In brief, the majority of our sample comes from urban areas, is well-educated and there is no difference in self-reported habitual physical activity (as per the Baecke score). See also **Supplementary Table 3**.*

12/ Although mixed models and quantile regression are mentioned, the manuscript lacks clarity on how covariates were selected and adjusted for in these analyses. Additionally, the description of predictive modeling lacks sufficient detail to assess its robustness.

We agree on this point, which has been raised multiple times in the review. Now, the statistical section in the Methods provides all the details necessary to assure the robustness of the results. See lines 172-211.

Results

13/ The Results section includes an overwhelming number of specific without prioritizing key outcomes. This dilutes the impact of the manuscript and makes it difficult for readers to discern the main message.

*We are grateful for the comment that prioritizing key outcomes would improve the discussion's impact. We have restructured the whole **Results and Discussion section**, aiming to improve clarity as suggested, while also incorporating the suggestions from other reviewers.*

14/ "Among the potential confounders influencing the link between diet and health outcomes are sociodemographic status, education level, and physical activity, none of which were accounted for in the study design. Given that this is a cross-sectional study utilizing cohort data, it is surprising and concerning that such critical confounding factors were neither collected nor adjusted for, as they could significantly impact the validity of the findings."

*We acknowledge the comment, and we agree that these confounders are of importance. Indeed, we have collected data on education level, region, and physical activity. Unfortunately, we have not collected data on family incomes. We have now specified the potential confounders in **Supp Table 3** and updated the Discussion sections accordingly.*

Rebuttal letter

Dear Editor, dear Reviewers,

On behalf of my co-authors and myself, it is my pleasure to submit our second response to the reviewers' comments. Here again, we were very grateful for the time the reviewers engaged with the revised version of the manuscript.

We believe that the current version of the manuscript has improved and will be eligible for publication in Communications Medicine.

Best regards,

Jan Gojda, MD, PhD

On behalf of all the authors

Point-by-point responses to the reviewers

Note:

The references to lines in the rebuttal letter relate to the file: Related manuscript file marked up, where you can also track the changes.

Reviewers' comments:

Reviewer #1 (Remarks to the Author):

Reviewer: James P Goode, PhD

Manuscript under review: Dietary intake, nutritional status, and health outcomes among vegan, vegetarian, and omnivorous Czech families: a cross-sectional analysis (COMMSMED-24-0924A)

Revision round: 2

The authors have adequately addressed my main concerns, but reporting inconsistencies, omissions and ambiguities remain. Nonetheless, this version of the manuscript is a marked improvement over the original submission. I appreciate the author's willingness to engage in scholarly discussion and the review process. As an aside, I applaud their decision to publicly share data and statistical code.

Lines numbers cited in this peer review report refer to the clean version of the submitted manuscript.

--- Graphical abstract ---

1. Update title.

The title of the graphical abstract has been updated to match the current version of th Title.

2. Remove 'vitamin' from 'vitamin/micronutrient'. Micronutrients by itself is sufficient.

We agree and we have removed „vitamin“. See the Graphical abstract.

--- Abstract ---

3. L43-45: This sentence reads as though children were followed into adulthood when in fact these are separates groups. I suggest rewording to avoid confusion.

We agree with reviewer at this point and we have restructured the sentence accordingly. See lines 43-45.

4. L47: It is unclear what is meant by 'across ages'. At this point, the reader is unaware that children were stratified into two age groups, or are you referring to children and adults?

We agree and we have removed the part of the sentence to avoid confusion. See line 47.

5. L51: For consistency, say 'vitamin B12' rather than only 'B12'.

We have added it as suggested. See line 51.

6. L54: It could be clearer that low iodine status is the concern (as opposed to excess).

We specified that we related here to low iodine status. See line 55.

List of abbreviations

7. L56-57: It's odd to define an abbreviation using another abbreviation (e.g., C-HDL = HDL cholesterol). I would write this out in full (i.e., HDL-C = high-density lipoprotein cholesterol).

8. L59: Why abbreviate group (GRP)? If only used once in a figure and not the main text, include the abbreviation in the legend of the figure.

All the abbreviations were revised and corrected as suggested. See lines 57-58 and 60.

--- Introduction ---

9. L80: Reference #2 is incomplete. Only a title and n.d. (no date) is listed in the reference list.

We have updated the erroneously insufficient reference.

10. L96-97: This sentence is incomplete. The impact of plant-based diets... on what?

We have completed the sentence. Now it reads: „Investigating plant-based diets within family units, encompassing both children and parents, is integral to understanding their impact to health outcomes.“

11. L104-106: Nutritional status is stated under point 1 and then the risk of micronutrient deficiencies is stated under point 2. Do these not overlap?

Indeed, these may seem as overlapping aims. As per nutritional status, anthropometry and strength (hand-grip) were assessed as opposed to nutrient deficiencies, based on laboratory analyses. Anyhow, to avoid confusion in terms we have removed „nutritional status“ from the first aim. See line 105.

--- Materials and methods ---

12. L115: It is stated that participants gave ‘written consent’, but did participants give written informed consent? This is an important distinction.

We agree at this point, of course, informed consent was obtained from every single participant. In children, parental informed consent was sought. See line 116.

13. L122: Can further information be provided about the study location or setting? A particular city, hospital or region in the Czech Republic?

We have clarified the location as suggested. See line 123-124.

14. L147-148: It is unclear what is meant by ‘3-day weighted dietitian-supervised dietary record method’. This was raised in my first peer review report (rebuttal #34). So, parents/carers completed a 3-day weighed food record that was then checked for completeness by a dietitian?

We are grateful for this explanation, dietitian-supervised may not be a universally used term. We have rephrased it to describe the whole procedure as suggested. Now it states (see lines 154-157):

Participants were trained in the diet recording and after completion the records were checked and verified by a dietitian. For children diet was recorded by parents or carers, only unweighted estimates were generally available in nursery facilities.

15. L155-157: Since this study sought to investigate the intake of key micronutrients between diet groups/families, the reason for not including the information provided by dietary supplements (despite being collected) should be noted (as explained in rebuttal #94).

We agree on this point and we have updated the Limitations section to specify it. See lines 630-633.

„Despite this approach, we were unable to produce any quantitative measure of micronutrient intake by supplementation, as the participants quite often could not recall exactly the manufacturer, nor quantify how often they used it.“

16. L161: The reader would also benefit from knowing the reason for separating children into two age groups (i.e., <3 and ≥3 years). This was raised in my first peer review report (rebuttal #41).

We agree that this information is important for a reader and we have amended the section accordingly. See lines 165 and 168-171.

17. L163-168: This section of text lacks clarity, especially point 2. Even after re-reading, it is still not entirely clear what is being investigated and how.

We agree that the the section was not clear. We have restructured it improve the clarity. Now it reads: : 1/ do clinical outcomes vary significantly among different dietary groups, 2/ which factors (e.g., sex, age, breastfeeding status for children, or supplementation when applicable), are most significantly related to the clinical outcomes, besides the dietary group, 3/ are these factors correlated (“clustered”) within the same family, and 4/ can the clinical characteristics effectively discriminate between different diet groups? See lines 165-177.

18. L186: ROC undefined.

Definition added as suggested. Line 197

19. L188: AUC undefined.

Definition added as suggested. See line 198

20. L197: FDR undefined.

Definition added as suggested. See line 208

--- Results---

21. L205: I assume the > (more than) symbol should be >= (more than or equal to)? This was raised in my first peer review report (rebuttal #41). Throughout the manuscript, children are referred to as <3 or >3 years, so what about children aged 3 years?

We are gratefull to the reviewer to spot this inconsistency. Indeed, the children were separated to below 3 years and 3 and more years. We have checked and corrected this throughout the text. See line 216.

22. L206-207: Here, there is no need to refer to a vegan diet (which has already been defined) as an ‘exclusively plant-based diet’. Why switch to an undefined term?

We agree and we have omitted the reference to the exclusive plant-based diet. See line 218

23. Table 2-3 – Footnotes: Review and update the list of abbreviations. For example, systolic and diastolic blood pressure are defined but not mentioned in Table 2-3. Consider throughout the manuscript.

We are gratefull for the comment, we have updated the abbreviations and checked it throughout the manuscript.

24. Table 5 – In L170-172, robust linear mixed-effect models with random effects (intercepts?) by family were adjusted for age, sex and variables related to breastfeeding, but in the footnote of Table 5, it states that robust linear mixed-effect models were ‘adjusted for the effect of potential confounders (age, sex, diet, family, and other covariates).’ There is a clear discrepancy here.

Thank you for pointing out the inconsistency. We have revised the footnote of Table 5 to clarify that diet was the main fixed effect of interest, and that models were adjusted for age, sex, breastfeeding-related variables (in children), and other relevant covariates (e.g., supplementation status, birth weight when applicable). Family was included as a random intercept to account for within-family clustering. The description in the main text has been aligned accordingly. See lines 179-183

25. The inclusion of Supplemental Table 3 is helpful (as per rebuttal #39), but a key suggestion was to also mention the number of families with more than one enrolled child.

We updated the Methods section to include the exact numbers (see line 219):

„Numbers of families with more than one child were: VN=13, VG=9, and OM=15.“

26. L215: In the Methods, children are referred to as < 3 or ≥ 3 years, but in the Results, the terms ‘infants/toddlers’ and ‘pre-schoolers/schoolers’ are introduced. There needs to be consistency in terminology. This was raised in my first peer review report (rebuttal #73).

We agree that this raised concern about consistency in terminology and we have omitted this description here. See line 225.

27. L217-219: I do not see how the linear mixed-effects model quantifies the ‘importance of covariates like age, sex, breastfeeding status, or family influence’ and how this relates to Table 5-6.

Thank you for pointing this out. The term ‘importance’ referred to the contribution of individual factors (including the random intercept for family) to out-of-sample prediction, which was assessed using Akaike’s Information Criterion (AIC). While Tables 5–6 present results from the same models—specifically the adjusted differences between diet groups—the assessment of variable importance based on model comparisons is shown in Figure 2. The text has been revised to clarify this distinction. See lines 227-238.

28. L222-223: This sentence repeats information from the previous paragraph. Also, ‘adults’ is mentioned but this section relates to children.

We agree that this sentence is superfluous and have therefore omitted it from the final text.

29. L224-225: It is important to clarify which diet groups are being compared rather than just reporting results for ‘children’, particularly as children are sometimes separated into two age categories and sometimes combined. Also, state which results table is being referenced here.

We have specified the age groups in children in the main text (see lines 230-237).

30. L224-225: The authors were asked to clarify what is meant by growth (as per rebuttal #46), but how this has been actioned is unclear.

Thank you for the clarification. We have specified in the methods (see lines 142-144) that growth was assessed as conversion of height and weight into percentiles using standard percentile charts validated for use in the Czech Republic. We have corrected throughout the text when comparing groups in anthropometric characteristics only. See line 245.

31. L225-226: What is the significance of weight/height being below the 3rd percentile? How percentiles were calculated should also be mentioned in the main text.

We agree that this information was only in the Supplementary Methods and should have been mentioned in the main text. We have now updated the Methods, see lines 142-144.

32. L227-228: Inconsistent use of abbreviations. Is it LDL cholesterol or C-LDL? Is it total cholesterol or TC? Consider throughout the manuscript.

We have checked the inconsistencies in the used abbreviations. See lines 248-251.

33. L227-231: It is unclear which tables and age/diet groups are being reported here. This issue was raised in my first peer review report (rebuttal #44). This applies to the entire Results section as well.

We have specified in the text that the Results for children that we comment on are based on the adjusted outcomes from Table 5. See lines 245-246.

34. L227-231: Is it necessary to report that four children had an LDL cholesterol measurement below the 'lower reference limit' (which has not been previously discussed or stated)?

We agree that this information is redundant – given its low clinical relevance – and we have removed it. See lines 250-251

35. In rebuttal #47, the authors state 'The iodine deficiency criteria used in the manuscript are based on the WHO document Urinary iodine concentrations for determining iodine status in populations. We have now added this reference. We recognize that the WHO median cut-offs for iodine deficiency should be used at the population level. In the revised version of the manuscript, we only comment on the median UIC in the subgroups.' However, no reference has been added and the number of children who had a urinary iodine concentration below WHO cut offs (<100 mcg/L and <20 mcg/L) is still reported in the Results (L237-240). Also, is it an issue that the WHO document is specifically referring to school-aged children (6 years or older) and adults, whereas most of the children in this study were aged <6 years?

We agree with the reviewer and we are grateful for this insightful comment. Indeed, the criteria for stratification of iodine deficiency based on UIC is defined by the relevant document only for children ≥ 6 years. We have reformulated our statements to clarify this. See lines 257-260.

36. L233-234: Where does this definition of mild vitamin D depletion come from? When citing cut offs and definitions in relation to nutrient status (which is done regularly), it is important to cite source documentation. Consider this throughout the manuscript.

We agree yet again at this point. The reference of the depletion was based on our institutional laboratory cut-offs (cited in the Supp file). We have decided based on the comment to simply omit referencing to deficiencies and state only the actual findings.

37. L234-235: It is not clear to the reader why this finding is 'interesting'. The significance of this finding is not even mentioned in the Discussion.

We agree that highlighting this finding is not desirable and we have removed it. On the other, the relevance of low PCR is commented in the Discussion. See line 254 and 515-518 for the Discussion.

38. L235: Typo. Should 'creatine' be 'creatinine'?

We are sorry, this is a typo, now corrected. Thank you for spotting it.

39. L236-237: Compared to other diet or age groups? Simply saying 'group' is a little vague because this study has both diet and age groups.

We generally tried to use term group for the description of the dietary group and stratum for age groups. Anyhow we have made it clear in the text that here we relate to the dietary group. See line 257.

40. L254: What is the 'upper reference limit' for total and LDL cholesterol? To further emphasise comment #36, it is important to cite what the cut off is and where it comes from (i.e., source documentation).

*# When referring to the reference limits, these are specified by the methods used for particular laboratory analysis and are elaborated in detail in **Suppl. Table 8** Laboratory Analytic Methods. We feel this may be clear, given that this information is provided in the Methods section.*

41. L256: Here, it is incorrect to say that there were 'no differences'. This was raised in my first peer review report (rebuttal #55).

Thank you for the comment. We completely agree that we cannot state no difference with our analytical approach. We have reformulated it here and checked throughout the text. See line 257.

42. L261-262: Serum zinc levels were not significantly different between vegan and vegetarians according to Table 6. Use of the phrase 'clear, significant gradient' implies that more than two group means were compared simultaneously, which is not the case here.

We agree that this statement is not based on the findings, so we have reformulated it. See line 280-282

43. L317-319: It is stated that 'only VN and VG children [< 3 years] had significantly lower intakes of saturated fats and cholesterol', but only saturated fat intake was lower (when compared to omnivores) according to Figure 3.

We are grateful for spotting the mistake; indeed, the breast-fed vegan children had considerable cholesterol intakes. We have corrected the statement, see line 339.

44. L319-321: The wording of this sentence could be much clearer. The phrase 'gradual shift' is the main issue as it lacks precision/clarity.

We agree that this part of the results was not clear enough and we have restructured the whole paragraph. Now it reads (see lines 337-346):

*„The differences among groups were negligible in *children < 3 years old*. In this age group, the nutrient intake was similar across all groups, with the exception of saturated fats, where lower intakes were found in VN and VG when compared to OM. Similarly, in the age stratum of *children ≥ 3 years old*, the total energy, carbohydrate, and fat intake was comparable among groups. Both groups adhering to plant-based diets (VN and VG) had significantly higher intake of fiber and consumed less cholesterol and saturated fats (VN<<VG) compared with the OM group. The protein intake was lower in VN compared to OM. Micronutrient intake was comparable among all groups except selenium, whose intake was lower in both VN and VG.“*

45. L323: Again, use the terminology defined in this manuscript for vegan and vegetarian diets rather than referring to them as 'plant-based diets.'

We have corrected the term, see line 352.

46. Figure 3 – The * remains undefined. This was raised in my first peer review report (rebuttal #53). The accompanying footnote also does not define 'GRP' or 'Ch'. Correct spelling of saturated fats (currently 'saturate fats'). To confirm, does the significance test compare the means of two specific diet groups (e.g., vegan vs. omnivore) or all three means (vegan vs. vegetarian vs. omnivore)?

We agree that the figure lacks a sufficient legend and there was a typo in the graph heading. We have updated accordingly; the legend now reads:

*„Vegan, vegetarian, and omnivore groups (VN, VG, and OM) age strata of children < 3 and > 3 years, and adults. * Significant difference among the three groups evaluated using Kruskal-Wallis test. GRP, group; Ch, children.“*

47. L363-368: There are discrepancies between the values reported in this paragraph and Table 7. Double-check.

We have double-checked and corrected the paragraph as suggested. See lines 363-368.

--- Discussion ---

48. L383-384: For consistency, say vegan, vegetarian and omnivore or use abbreviations (VN, VG and OM). Do not switch between them. Either write out in full or abbreviate. Consider throughout the manuscript.

We have corrected the section as suggested. See line 404.

49. L393: This sentence lacks clarity and is incomplete: 'The most prominent covariate for children was age, and for adults, sex.'

We have corrected the phrase, now it reads (see line 413-414) :

„In children, age was the most influential covariate, whereas in adults, it was sex.“

50. L397-L400: The wording of the final key findings also lacks clarity.

We have corrected the phrase to improve the clarity, now it reads (see line 418-424) :

„Dietary groups of VN vs OM were reliably discriminated in both adults and children. While in children this difference was largely driven by supplement-related biomarkers, in adults it persisted even after excluding supplementation effects, showing the influence of the diet itself.“

51. L422-426: How do your results contrast with the Polish study? Vegan children were generally found to be shorter in both cases.

We agree that the statement was misleading and have corrected the phrase; now it reads (see line 446-450):

„These findings corroborate Polish study outcomes performed on children aged 5–10 years where vegan children were found to be shorter, but contrast the results of the Finnish study outcomes on preschool vegan children (median age 3.5 years), which showed no differences among groups.“

52. L460: The presentation of the age categories is difficult to follow here.

Thank you for the comment, we have rephrased the whole sentence, now it reads (see lines 483-484):

The difference in CVD risk-associated increased both with age ('children<3 years' < 'children≥3 years' < 'adults') and across dietary groups (VN < VG < OM).

53. L462-463: See rebuttal #14 and #48 in relation to the phrase 'verge of significance'.

We agree again at this point, and we have corrected it to state what we found (see line 487):

„Adult VN had a lower BMI when compared to OM (p=0.07).“

54. L475-476: This sentence reads as though it is referring to the present study (which is it not). Reword for clarity.

We have rephrased to improve the clarity. See line 500.

55. L477: BMD undefined.

Corrected as suggested. See line 501.

56. L488-490: In the vegan group? It is also unclear what is meant by 'largely insignificant'.

We agree that under our statistical approach, the results are either significant or not. We have rephrased accordingly and added a missing „vegan“ word. See lines 513-515.

57. L493-495: I do not follow the logic of this sentence. Would low calcium bioavailability from the diet increase faecal calcium loss, not urinary calcium loss?

Thank you for the comment. To clarify it, we considered urinary calcium losses (CCR) as markers of body saturation with calcium. Lower calcium intake, or lower bioavailability, would be associated with lower calcium clearance (CCR). Anyhow, we have slightly rephrased the sentence to improve the clarity. See lines 519-520

58. L519-520: According to Supplemental Table 6, iodine intake was only significantly lower when compared to omnivores (not vegetarians as well).

Thank you for spotting the mistake. We have corrected it. See lines 544-545

59. L521: What is the recommended intake for iodine? And how much lower on average?

Thank you for the comment, the reference values are found in the referenced document. We have updated the whole sentence as suggested (see lines 545-546):

„Unexpectedly, the intakes were, on average, roughly half the daily recommended doses of 150µg/day.“

60. L537: Consider changing ‘namely’ to ‘particularly’.

We have changed it accordingly. See line 563.

61. L558: For consistently, say ‘vitamin B12’ rather than just ‘B12’. Consider throughout.

We have corrected this and checked throughout. See line 583.

62. L576-578: The prevalence of allergies among adults was dependent on family? Where is this reported in the Results section?

We have removed the sentence entirely. The statement was based on previous versions of analyses.

63. L578-581: It is unclear what is meant by ‘these nutrient’. Which nutrients?

We have specified in the text, that now reads (see lines 604-605):

„The observed differences among groups could be attributed to the influence of supplementation, as both the proportion of individuals taking vitamin B12 and D supplements and the levels of circulating biomarkers for these were higher in the vegan and vegetarian groups, respectively.“

64. L591-594: This sentence is incomplete. The effect of vegan diets... on what?

We have specified in the text, that we meant health risks and benefits (see line 617).

65. L602-603: Here, it is stated that ‘random enrolment across seasons’ was performed, however, this was not mentioned in the Methods.

We are grateful for this comment. The statement of random enrollment was misleading. We have conveniently enrolled over the enrollment period (as per the Methods). We have corrected this (see line 628).

--- Supplementary material ---

66. Update title.

Thank you for the suggestion, we have updated it. See Suppl. File.

Reviewer #3 (Remarks to the Author):

Dear authors,

Congratulation for the comprehensive and thoughtful revisions made in response to the reviewer feedback. The manuscript is now substantially improved in terms of structure, methodological clarity, and scientific rigor. The expanded Methods section, clearer statistical reporting, and improved interpretative caution, particularly in replacing causal language with associative framing greatly strengthen the paper. The abstract is now more precise, and your discussion is particularly informative.

I believe this manuscript makes a valuable contribution to the literature on plant-based diets in family and pediatric contexts. That said, I encourage you to consider the following minor improvements during the final revision:

1/ Abstract: While comparisons are now clearer, the phrasing of some findings remains dense. Simplifying or streamlining certain sentences (e.g., bone turnover and vitamin D findings) could improve clarity for general readers.

Thank you for the suggestion, we have restructured the Abstract to make it more comprehensible. See the Abstract.

2/ Discussion (Section 4.3 – Bone health): The section is thorough but could be more concise. Consider condensing repeated explanations about P1NP, vitamin D, and calcium to maintain focus without redundancy.

Thank you for the suggestion, we have made some changes based also on the comments of the other reviewer. We believe that some detailed focus needs to be maintained as the bone health in vegans is highly debated in the literature. See lines 497-534.

3/ Discussion (Section 4.6 – Family clustering): Clarify whether the statement about allergy prevalence and family background reflects statistical findings or observational trends.

We are grateful for this comment. The statement was based on previous analyses (i.e. descriptive finding) and we have not incorporated it in the adjusted models. We decided (also based on the other reviewer's suggestion to remove the sentence from the section.

4/ Ensure consistent use and definition of technical terms (e.g., P1NP, MMA, sTfR). Also, harmonize phrases like “trend toward significance” and “no significant difference” for clarity and accuracy.

Thank you for the very valid comment. We have checked and corrected it throughout the text.

5/ Some long sentences, especially in the Introduction and Methods, could benefit from minor editing to improve flow and readability for a broader scientific audience.

We appreciate this comment and we re-read and restructured hard-to-read sentences.

Overall, this is an important and well-executed study, and I appreciate the significant effort invested in strengthening it during revision.

We appreciate the time and effort the reviewer took to help us improve the Manuscript.

The comments below from the reviewer are marked by **OBS (numbered 1-32)**. **Line numbers in the merged file are marked red**

Abstract

1.OBS The Abstract is a bit hard to read as it does not describe what is compared when something is said to be lower or higher in or for a diet group, i.e. the information is lacking about „...as compared to which other group“, see in comments below.

35 Plant-based diets are growing in popularity because of their perceived environmental
36 and health benefits. However, they may be associated with safety risks, that may
37 cluster within families.

2.OBS: ref for growing in popularity? How? Among health staff? Advisors that give recommendations about food based dietary guidelines? Among general population?

3.OBS: the safety risk, is it larger or smaller than the general OM diet?

37.....we conducted a cross-sectional study of 95 families (47

38 vegan [VN], 23 vegetarian [VG], and 25 omnivore [OM]), including 187 adults, 65

39 children >3 years, and 77 children

4.OBS: are the families clean as VN, VG or OM?

43..... OM had a

44 higher BMI, diastolic blood pressure, and lower fat-free mass in adults.

5.OBS higher than whom? VN or VG or both?

44.....Higher bone

45 turnover (P1NP) was found in older children and adult VN, where it was related to

46 higher PTH levels.

6.OBS higher than whom? both VG and OM or what?

46 Paradoxically, vitamin D levels were generally higher in VN. Lower

47 urinary iodine, associated with lower intake in VN was found across all age strata,

7.OBS vitamin D generally higher in VN than in whom? VG? Or OM? Or both?

8.OBS Lower urinary iodine in VN, as compared to?

51.....Although no

52 serious adverse effects of the diet were found, iodine status and bone health in

53 vegans warrant further research

9.OBS what about the other groups? any serious adverse effects?

10.OBS in general about the Abstract: Would it be more wise to report the values outside the reference limits for e.g. nutrient status? And outside limits for recommended intake (too low, too high)?

Introduction

55 The global trend to reduce the environmental burden of food production and tackle the

56 obesity pandemic is being followed by a reduction in the consumption of foods of

57 animal origin. The growing trend towards plant-based diets is increasingly evident in

58 many regions from the Eastern European block including the Czech Republic.

11.OBS. What is meant? It is absolutely necessary to say e.g. recommendations/guidelines and use global and international references. A trend can be something from a single moviestar or singer or whatever. Please use e.g. recent IPCC documents and WHO? And country reports that have really focused on this you say is a global trend? Many countries are actually still hesitating probably partly because of the strong influence of meat and dairy producers, which are the agricultural producers receiving the majority of subsidies aimed for food production.

60...3% of Czech consumers identify themselves as vegan, 7% as

61 vegetarian, and a remarkable 25% as voluntarily reducing their intake of animal-based

62 foods, significantly more compared to previous years ^{1,2}

12.OBS these values actually tell the whole story, 3 and 7 percentages, are 10% together i.e. a low % of the population. 25% have finally diminished their intake of animal based foods – something that has been advised since the seventies or eighties in the last century in many countries.

13.OBS Refs number 3 and 4 tell about positive health effects of plant-based diets, and refs 5 – 12 about negative effects, this lacks some balance.

Results

Line 100-101

14.OBS two participants had hypertension compensated on the treatment 101 (VG=2, OM=1) ? (seems 3)

Line 103-104

15.OBS „87 subjects distributed 104 evenly across groups (OM=27, VG=23, VN=27)“ ?? (seems 77)

- **16.OBS** Generally in this chapter, focus more on: Difference between groups in Clinical findings that might be associated with the diet e.g. allergy !

In children the nutr.status of iodine, Vit-D, Folate, Vit-B12 are all of interest and surprisingly some VN have higher levels than OM in B12, vit-D as well as in Folate, but apparently iodine warrants a further investigation. **17.OBS.** Explain? probably in discussion

In adults the status associated with the risk of coronary-heart disease, i.e. blood lipids and blood pressure, seem higher for OM, however there seems risk of insufficient iron and zinc statuses.

Line 201

18.OBS The lowest 20%, and highest 80%

-Generally in this chapter: Is someone at probable risk?

Line 214

„... vegan diets may be associated with wider spread towards higher iodine levels“

19.OBS for discussion, include if this is an important option/fact for all diets in general, they can be bad and they can be healthy so the choice of diet ingredients is a key (independent of type of OM, VG, VN). The next step in research and guidelines should be more openly define the critical points in each diet

Line 247-253

Young children: age, birth weight, breastfeeding and diet influence the measured outcomes;

20.OBS this is relevant and expected – Discussion should involve that the age interval is large for this period in life incl the fastest growth (the first year) and a speedy development

Line 254-260

Children (<3 yrs): most important for outcomes were: Age, birth weight, supplementation and diet

21.OBS Further in this chapter about „Family clustering and covariates' importance“ The family clustering was important as explored for all children together (because of otherwise too few children in each group) in fact the reason can however be diet, and other factors in the environment. In adults sex and diet seemed most important

Line 328-332

„Generally, we were able to reliably discriminate between VN and OM in adults, with out-of-sample AUC 0.82 (95% CI: 0.69 to 0.92), whereas it was only 0.54 in the baseline model (not utilizing clinical characteristics), with a mean (?) AUC gain of 0.28 (0.08 to 0.49). The strongest predictors of VN diet are lower glycemia, total cholesterol, zinc, ferritin, and urea, and higher P1NP and folate.“

22.OBS What does the ? mean here?

Methods (this chapter was far back in the PDF document – but I read it and put it here as it is logical)

Line 600 „venous blood sampling after 12 hours“ **23.OBS** meaning 12 hrs fasting??

Line 605 „weighted dietary record method“ **24.OBS** what kind of scales were used??

Line 608-609 „For products not listed in any of the databases, the dietitian recorded nutrient content from the product packaging.“ **25.OBS** How common was this for the different age groups? how many product? Can some intake values be too low because of missing values (analysis in the food label/database)?

Discussion

Line 360-361 „Family impacted height, micronutrient status (Se, Zn, urinary iodine), and vitamin levels (folate, B12, and D)“

26.OBS why? can it be an unsure methodological association between the diet and the biochemical measures of the nutrients statuses? Speculate: are the food table databases probably too low on some of these nutrients? Or are the biochem analysis of the status probably unsure... **OBS** there is too little on the status measurements in the Method chapter!

Line 365-396

27.OBS In the chapter about growth and anthropometrics in children – A very good chapter but add a sentence about genetics.

Line 398-....

Also the following chapters are good, on indicators of better cardiometabolic health in vegans which can be identified as early as preschool age; on bone health; and on iodine status.

Line 502-505

Non-heme iron absorption in vegan diets is susceptible to various inhibitors such as phytates, polyphenols, and *calcium*, which collectively contribute to reduced iron absorption efficiency in vegan diets, despite adequate iron intake from plant

28.OBS controversial to point out calcium here as it might be too low, as said before, for iron the meal composition might be important especially for those totally relying on non-heme iron.

Otherwise this chapter is ok

Line 521

29.OBS The chapter on family – consider former comments

Line 544-572

Chapter on strength and limitations: **30.OBS** former comments on the text

Conclusion

Line 575-576

„To conclude, we described the differences among families with distinct eating habits in anthropometric measures, health, and nutritional“ **intake and** „status indices.“

31.OBS diet or nutrient intake is missing from this sentence, see added „intake and“

Line 581

32.OBS The positive effects of plant-based diets on public health taking the possible weaknesses into consideration, should be stated more clearly in the end, as well as mentioning the negative consequences of OM seen in this and former studies probably because of the habit not taking food based dietary guidelines seriously enough.